# DataComp-LM: In search of the next generation of training sets for language models

**Jeffrey Li*[1,2]  Alex Fang*[1,2]  Georgios Smyrnis*[4]  Maor Ivgi*[5]**
**Matt Jordan[4]  Samir Gadre[3,6]  Hritik Bansal[8]  Etash Guha[1,15]  Sedrick Keh[3]  Kushal Arora[3]**
**Saurabh Garg[13]  Rui Xin[1]  Niklas Muennighoff[22]  Reinhard Heckel[12]  Jean Mercat[3]  Mayee Chen[7]  Suchin Gururangan[1]  Mitchell Wortsman[1]  Alon Albalak[19,20]  Yonatan Bitton[14]**
**Marianna Nezhurina[9,10]  Amro Abbas[23]  Cheng-Yu Hsieh[1]  Dhruba Ghosh[1]  Josh Gardner[1]**
**Maciej Kilian[17]  Hanlin Zhang[18]  Rulin Shao[1]  Sarah Pratt[1]  Sunny Sanyal[4]  Gabriel Ilharco[1]**
**Giannis Daras[4]  Kalyani Marathe[1]  Aaron Gokaslan[16]  Jieyu Zhang[1]  Khyathi Chandu[11]**
**Thao Nguyen[1]  Igor Vasiljevic[3]  Sham Kakade[18]  Shuran Song[6,7]  Sujay Sanghavi[4]  Fartash Faghri[2]  Sewoong Oh[1]  Luke Zettlemoyer[1]  Kyle Lo[11]  Alaaeldin El-Nouby[2]  Hadi Pouransari[2]  Alexander Toshev[2]  Stephanie Wang[1]  Dirk Groeneveld[11]  Luca Soldaini[11]  Pang Wei Koh[1]**
**Jenia Jitsev[9,10]  Thomas Kollar[3]  Alexandros G. Dimakis[4,21]**
**Yair Carmon[5]  Achal Dave†[3]  Ludwig Schmidt†[1,7]  Vaishaal Shankar†[2]**

[1]University of Washington, [2]Apple, [3]Toyota Research Institute, [4]UT Austin,
[5]Tel Aviv University, [6]Columbia University, [7]Stanford, [8]UCLA, [9]JSC, [10]LAION, [11]AI2,
[12]TUM, [13]CMU, [14]Hebrew University, [15]SambaNova, [16]Cornell, [17]USC, [18]Harvard,
[19]UCSB, [20]SynthLabs, [21]Bespokelabs.AI, [22]Contextual AI, [23]DatologyAI

contact@datacomp.ai

## Abstract

We introduce DataComp for Language Models (DCLM), a testbed for controlled dataset experiments with the goal of improving language models. As part of DCLM, we provide a standardized corpus of 240T tokens extracted from Common Crawl, effective pretraining recipes based on the OpenLM framework, and a broad suite of 53 downstream evaluations. Participants in the DCLM benchmark can experiment with data curation strategies such as deduplication, filtering, and data mixing at model scales ranging from 412M to 7B parameters. As a baseline for DCLM, we conduct extensive experiments and find that model-based filtering is key to assembling a high-quality training set. The resulting dataset, DCLM-BASELINE, enables training a 7B parameter language model from scratch to 64% 5-shot accuracy on MMLU with 2.6T training tokens. Compared to MAP-Neo, the previous state-of-the-art in open-data language models, DCLM-BASELINE represents a 6.6 percentage point improvement on MMLU while being trained with 40% less compute. Our baseline model is also comparable to Mistral-7B-v0.3 and Llama 3 8B on MMLU (63% & 66%), and performs similarly on an average of 53 natural language understanding tasks while being trained with $6.6\times$ less compute than Llama 3 8B. Our results highlight the importance of dataset design for training language models and offer a starting point for further research on data curation. We release the DCLM benchmark, framework, models, and datasets at https://datacomp.ai/dclm.

---

\* Shared first author; † Shared last author

38th Conference on Neural Information Processing Systems (NeurIPS 2024) Track on Datasets and Benchmarks.

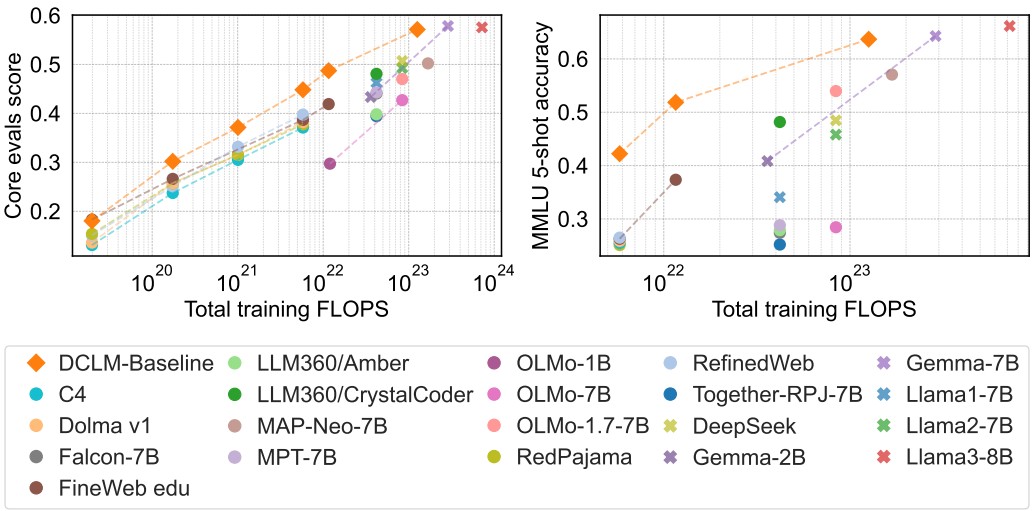

Figure 1: **Improving training sets leads to better models that are cheaper to train.** Using DataComp-LM, we develop a high-quality dataset, DCLM-BASELINE, which we use to train models with state-of-the-art trade-offs between compute and performance. We compare on both *(left)* a CORE set of tasks and on *(right)* MMLU 5-shot. Specifically DCLM-BASELINE (orange) shows favorable performance relative to both close-source models (crosses) and other open-source datasets and models (circles). Models in this figure are from [4, 10, 22, 46, 73, 103, 103, 106, 127, 136, 156, 160, 162, 166–168, 195]. Table 33 provides a table version of this figure.

# 1 Introduction

Large training datasets are an important driver of progress in the recent language modeling (LM) revolution [63, 69, 90, 129, 137, 156, 161, 178]. As the cost of training state-of-the-art language models continues to grow, researchers increasingly focus not only on *scaling* but also on *improving* training datasets that enable efficient generalization on a wide range of downstream tasks. Indeed, there is a growing number of proposals for filtering data, removing (near-) duplicates, finding new data sources, weighting data points, generating synthetic data, and so on [2, 8, 75, 94, 97, 102, 119, 183].

A key challenge in this emerging research area is a lack of controlled comparisons. While the aforementioned proposals generally use the same evaluation datasets, researchers often compare models that are trained with different architectures, compute, or hyperparameters. Hence, it is often unclear what data curation strategies work best: Are the results of training set A better than training set B because training set A is truly better, or because the model trained on A was combined with a better architecture, learning rate schedule, or more compute? Disentangling the many factors influencing the quality of a language model is crucial to understanding which data curation strategies work best and ultimately building better language models.

Beyond the lack of standardized benchmarks, another challenge for research on training data is that details about training sets are becoming increasingly rare, even for open weight models such as the Llama, Mistral, or Gemma models [83, 160, 167]. For all of these models, the training sets are not publicly available, and the corresponding model documentation only provides a coarse description of the respective training data, if any at all. As a result, it is currently unclear what ingredients constitute a state-of-the-art training set for langauge models.

To address these challenges, we introduce **DataComp for Language Models (DCLM)**, the first large-scale benchmark for language model training data curation. In DCLM, researchers propose new training sets and data curation algorithms and then evaluate their datasets by training LMs with a *fixed* training recipe on their data. By measuring the performance of the resulting model on downstream tasks, researchers can quantify the strengths and weaknesses of the corresponding training set.

To enable DCLM, we contribute a comprehensive experimental testbed. A key component is **DCLM-POOL**, a corpus of 240 trillion tokens derived from Common Crawl [45]. DCLM-POOL is the largest public corpus for language model training and forms the cornerstone of the DCLM filtering track,

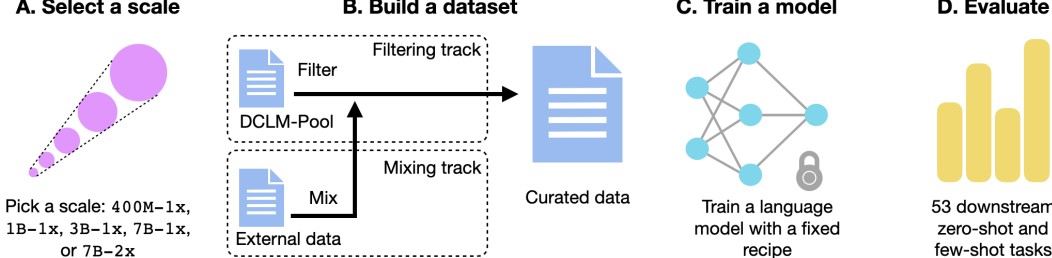

Figure 2: **The DCLM workflow.** *(A)* A participant first chooses a scale, where larger scales reflect more training tokens or model parameters. *(B)* A participant then filters a pool of data (filtering track) or mixes data of their own (mixing track) to create a dataset. *(C)* Using the curated dataset, a participant trains a language model, with standardized training code and scale-specific hyperparameters, which is then *(D)* evaluated on 53 downstream tasks to judge dataset quality.

where participants aim to curate the best possible training set out of DCLM-POOL. In addition, we provide open-source software for processing large datasets with several filtering approaches.

The high cost of training language models makes it necessary to understand the performance of training recipes across different compute and data scales. Hence, our third contribution is an investigation of **scaling trends for dataset design**. We find that models as small as 400M parameters can still provide signal on which training sets perform better at larger scales. Based on our experiments, we organize DCLM into five compute scales spanning a range of about $600\times$ in compute from 400M parameter models to over-trained 7B models. This multi-scale design makes DCLM accessible to researchers with varying compute budgets.

As a starting point for DCLM, we conduct **416 baseline experiments** with different training sets and compute scales. Our experiments identify model-based filtering as a key component for effective data curation. We also show that details of the filtering model can have a large impact on performance, ranging from 35% to 44% accuracy on MMLU 5-shot [77] at the 7B parameter scale (280B training tokens). Interestingly, a simple bigram classifier, combined with a carefully selected set of positive and negative examples, performs best among the classifiers we experimented with. In addition, we find that human quality judgments have only limited value in identifying high-quality training data.

Finally, we combine our results into **DCLM-BASELINE, a new state-of-the-art public training set** for language models. When training a 7B parameter language model on 2.6 trillion tokens using DCLM-BASELINE, the resulting model achieves 64% on MMLU, which is state-of-the-art among open-data models and close to models such as Mistral-7B-v0.3 (63%) or Llama 3 8B (66%) that are trained with up to $6.6\times$ more compute (Llama 3 8B). Compared to Llama 2 7B, training a 7B parameter model on 280B tokens from DCLM-BASELINE achieves 5 pp higher MMLU while being trained with $7\times$ less compute. As our 7B model uses a standard decoder-only Transformer [133, 167, 171], our results also highlight that a systematic approach to data curation is key to training performant language models.

We publicly release our DCLM framework, models, and training sets at `https://datacomp.ai/dclm` to enable other researchers to participate in DCLM and to strengthen the empirical foundations for data-centric research on language models.

## 2 Related work

We summarize closely related work in this section and provide additional related work in Appendix B.

**Data curation for language models.** To collect large datasets for training LMs [31], researchers typically resort to web crawls, which can contain undesirable content that can be improved via curation. Most data curation efforts focus on methods for improving model performance [31, 127, 134, 137, 156, 176], including filtering by language [47, 92, 137, 186], heuristic-based filtering [37, 63, 127, 134, 156], quality filtering [53, 105, 145, 176, 184], data deduplication [3, 94] and mixing [6, 154, 183]. While prior work examines a limited set of filters, we conduct the largest public investigation of data curation, resulting in a strong DCLM-BASELINE dataset.

Table 1: **DCLM competition scales.** DCLM contains five competition scales, enabling research in varying compute regimes. Each scale specifies the model size ('Model parameters', $N$), the number of tokens seen during training ('Train tokens', $D$), and the size of the original pool that can be used for filtering ('Pool size'). We provide an estimate of the compute required for training ('Train FLOPs'$= 6ND$) and GPU hours ('Train H100 hours') using the OpenLM training framework [76].

| Scale | Model parameters | Train tokens | Train FLOPs | Train H100 hours | Pool size |
|---|---|---|---|---|---|
| 400M-1x | 412M | 8.2B | 2.0e19 | 26 | 469B |
| 1B-1x | 1.4B | 28.8B | 2.4e20 | 240 | 1.64T |
| 3B-1x | 2.8B | 55.9B | 9.4e20 | 740 | 3.18T |
| 7B-1x | 6.9B | 138B | 5.7e21 | 3,700 | 7.85T |
| 7B-2x | 6.9B | 276B | 1.1e22 | 7,300 | 15.7T |

**Open-source datasets.** As the scale of LMs has increased over the past years [4, 39, 79, 121, 134, 160, 167, 168], the community has curated larger datasets to match. Early works include the C4 dataset with 160 billion (B) tokens and The Pile [63] with 300B tokens. More recently, RefinedWeb [127] contains 600B tokens, Dolma [156] 3 trillion (T) tokens, FineWeb 15T tokens [128], and RedPajama-v2 30T tokens [46]. There are also large domain-specific datasets, such as the code-focused StackV2 with 900B tokens [107], as well as high-quality filtered subsets such as FineWeb-Edu [106] with 1.3T tokens. We include performance comparisons with various datasets in Figure 1 and examine FineWeb's LightEval evaluation framework more closely in Appendix G. We release the largest pool of raw text data to date with 240T web-crawled tokens. We also release DCLM-BASELINE, a high-quality dataset from our pool that yields better models than prior datasets.

**Data-centric benchmarks.** Past work on benchmarking data improvements includes dataset distillation [50], curriculum learning [143], and transfer learning [5, 32]. In DataComp [61] and DataPerf [112], participants iterate on a dataset with a fixed model and training recipe for vision, vision-language, and speech tasks. For LMs, the Data-Juicer [36] effort includes benchmarks for cleaning and mixing *fine-tuning* data while the BabyLM challenge *Loose* track [173] focuses on efficient development of 125M to 220M parameter LMs pretrained on 10M to 100M tokens. With a 200T token pool and 7B models, DCLM is the largest data-centric benchmark for language models.

## 3 The DataComp for language models (DCLM) benchmark

This section describes the main components of DCLM. We start with DCLM-POOL, the raw text corpus underlying our benchmark (Section 3.1). We then develop the DCLM workflow, visualized in Figure 2: selecting a competition scale (Section 3.2), curating a dataset by filtering DCLM-POOL and potentially mixing in other sources (Section 3.3), training a model with fixed hyperparameters (Section 3.4), and evaluating the model to score the dataset (Section 3.5).

### 3.1 DCLM-POOL

DCLM-POOL is an unfiltered web-text corpus comprised of all Common Crawl [45] data prior to 2023. Based on Section 4.2, we re-extract text from HTML using `resiliparse` [20, 21] instead of using Common Crawl's pre-extracted text. DCLM-POOL contains 200B documents (370TB after gzip compression), resulting in 240T GPT-NeoX [24] tokens. See Appendix E for additional details.

**Decontamination.** Test set samples often contaminate language model training sets [52, 55, 187]; however, the effect of such samples on downstream performance remains largely unclear [94, 121, 156]. To allow researchers to better understand contamination, we release decontamination tooling instead of decontaminating DCLM-POOL directly. First, as used in Section 4.6, we implement our own decontamination process for two popular tasks, MMLU and Hellaswag [78, 192]. Second, we allow participants to examine their datasets for overlap with *all* of our test sets based on Lee et al. [94]. We ask all submissions to disclose a decontamination report and avoid using highly-contaminated data. For the highest scoring submissions, we plan to specifically evaluate them for contamination.

## 3.2 Competition scales: Supporting participants with different compute constraints

To ensure DCLM is accessible to researchers with different compute constraints and to facilitate the study of scaling trends, we create different competition scales spanning three orders of compute magnitude (Table 1).Each scale (i.e., 400M-1x, 1B-1x, 3B-1x, 7B-1x, and 7B-2x) specifies the number of model parameters (e.g., 7B) and a *Chinchilla multiplier* (e.g., 1x). The number of training tokens for each scale is $20 \times$ number of parameters $\times$ Chinchilla multiplier so that a multiplier of 1x corresponds to a compute allocation that Hoffmann et al. [79] found near-optimal.

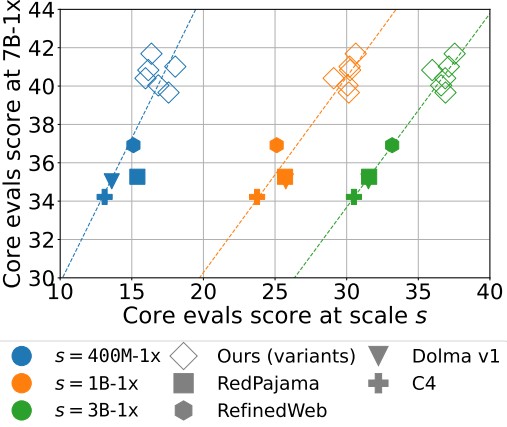

Figure 3: **Datasets rank consistently across competition scales in DCLM.** This makes it possible to iterate on data curation at small scales.

A potential pitfall in our multi-scale design is that the ranking of data curation methods may change when increasing the compute scale. To better understand this concern, in Figure 3, we plot the performance of 10 methods at the 7B-1x scale as a function of their performance at smaller scales. We find high rank correlation between the results for smaller scales (400M-1x, 1B-1x, 3B-1x) and those for the larger 7B-1x scale (Pearson's $r = 0.838$, $r = 0.956$, $r = 0.982$ respectively), suggesting better curation strategies at smaller scales transfer to larger scales. For more competition scale ablations, including experiments suggesting dataset improvements are largely orthogonal to training hyperparameters, see Appendix H.

## 3.3 Benchmark tracks: Filtering and mixing

After choosing a scale, participants choose one of two tracks. (i) In the *filtering track*, participants propose algorithms to select training data from a candidate pool. We start with five pools, one for each scale in Table 1, which are random document subsets of DCLM-POOL. We restrict initial pool sizes by scale to encourage scalable filtering strategies and reflect realistic data download and storage constraints. (ii) In the *mixing track*, a submission may combine documents from potentially many sources. For instance, participants can synthesize documents from DCLM-POOL, a custom crawl, Stack Overflow, and Wikipedia. Appendix C provides detailed rules for each track, and Appendix D describes our extensible open-source tooling for executing filtering and mixing operations.

## 3.4 Training

To isolate the effect of dataset interventions, we fix a training recipe at each scale. Based on prior ablations on model architectures and training [4, 31, 39, 62, 79, 93, 133, 167, 168, 180], we adopt a decoder-only Transformer (e.g., GPT-2, Llama) [133, 167, 171], implemented in OpenLM [76]. We also provide unified data processing utilities. Appendix F contains additional training details.

## 3.5 Evaluation

Our full evaluation suite, based on LLM-Foundry [115], contains 53 downstream tasks suitable for base model evaluation (i.e., without finetuning): from question answering to open-ended generation formats, considering varied domains like coding, text-book knowledge, and common-sense reasoning. To evaluate data curation algorithms, we focus on three main performance metrics. First, we consider *MMLU 5-shot accuracy* [78], which is widely used to compare state-of-the-art models like GPT-4 [121] and Llama 3 70B [4]. Second, we propose CORE *centered accuracy*, computed over a subset of 22 tasks (e.g., HellaSwag [192] and ARC-E [43]) that provide a low-variance signal even at small scales, linearly rescaling the accuracy per task so that 0 corresponds to random guessing and 1 corresponds to perfect accuracy. Finally, we report EXTENDED *centered accuracy*, which averages the centered performance for all of our 53 tasks. For more metric details, see Appendix G.

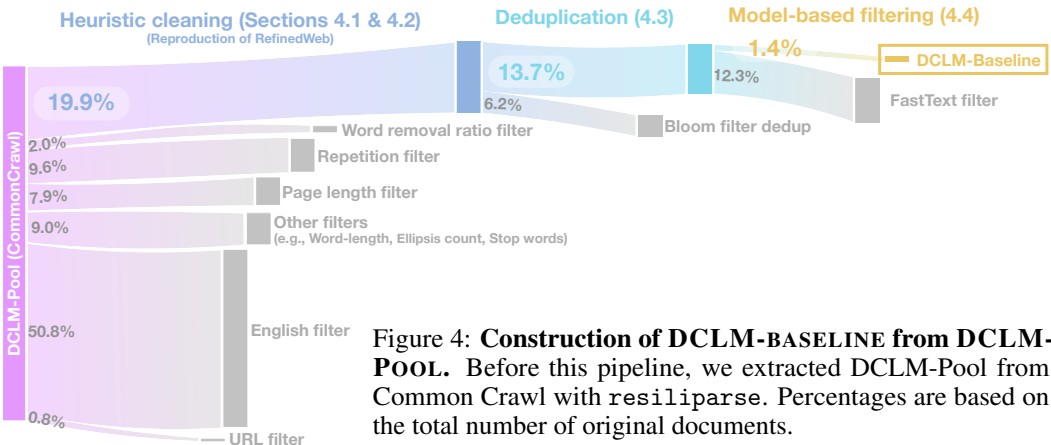

Figure 4: **Construction of DCLM-BASELINE from DCLM-POOL.** Before this pipeline, we extracted DCLM-Pool from Common Crawl with `resiliparse`. Percentages are based on the total number of original documents.

# 4 Building high-quality training datasets with DCLM

We now show how the DCLM workflow can lead to high-quality datasets and quantify the effect of data curation methods. This section describes the process of converting Common Crawl into our dataset, DCLM-BASELINE, as shown in Figure 4. We provide ablation experiments for each step along the way. We first evaluate open-source datasets as a starting point (Section 4.1). Next, we experiment with alternatives for several key phases of dataset construction: text extraction (Section 4.2), deduplication (Section 4.3), and model-based filtering (Section 4.4). We then experiment with mixing in high-quality sources (Section 4.5) and provide a contamination analysis (Section 4.6). In Section 5, we scale up this approach to train a 7B model for 2T tokens.

## 4.1 Evaluating existing training datasets

We begin by evaluating several well-known open-source datasets (`C4` [52, 136], `RefinedWeb` [127], `RedPajama` [166], and `Dolma-V1` [156]) in Table 2. While all four datasets use various heuristic filters and data cleaning steps, we find that `RefinedWeb` performs the best on our CORE and EXTENDED metrics at the 7B-1x scale. `RefinedWeb` applies the following filtering pipeline: Common Crawl text extraction, heuristic selection rules (e.g., to remove spam), and deduplication of repeated content. Interestingly, `RefinedWeb` is solely filtered from Common Crawl, unlike `RedPajama` and `Dolma-V1`, which additionally mix in curated, "high-quality" sources like Wikipedia. The comparison suggests the relative strength of filtering, which we explore later in our experiments.

> **Takeaway:** For DCLM-BASELINE and other experiments, we adopt `RefinedWeb`'s heuristic filters.

## 4.2 Text extraction

Text extraction is a crucial early processing step that pulls content from raw HTML. To understand the effect of this step, we compare three extraction approaches: `resiliparse`, `trafilatura` (used

Table 2: **Comparison to existing datasets** (7B-1x scale). Despite not mixing high-quality sources (unlike `Dolma-V1` and `RedPajama`), `RefinedWeb` performs best.

| Dataset | CORE | EXTENDED |
|---|---|---|
| C4 | 34.2 | 18.0 |
| Dolma-V1 | 35.0 | 18.4 |
| RedPajama | 35.3 | 18.2 |
| RefinedWeb | **36.9** | **19.8** |

Table 3: **Comparison of text extractors** (`1B-1x` scale). We apply three approaches for text extraction from HTML, process their output using the `RefinedWeb` heuristic filters, and evaluate the models trained on the resulting datasets. We find stricter extractors such as `resiliparse` and `trafilatura` are superior to WET files provided by Common Crawl.

| Text Extraction | CORE | EXTENDED |
|---|---|---|
| resiliparse | 24.1 | **13.4** |
| trafilatura | **24.5** | 12.5 |
| WET files | 20.7 | 12.2 |

Table 4: **Quality filtering comparison** (`1B-1x` scale). We evaluate various choices for model-based quality filters. Training a `fastText` classifier for filtering performs best.

| Filter | CORE | EXTENDED |
|---|---|---|
| RefinedWeb reproduction | 27.5 | 14.6 |
| Top 20% by Pagerank | 26.1 | 12.9 |
| SemDedup [1] | 27.1 | 13.8 |
| Classifier on BGE features [182] | 27.2 | 14.0 |
| AskLLM [145] | 28.6 | 14.3 |
| Perplexity filtering | 29.0 | 15.0 |
| Top-k average logits | 29.2 | 14.7 |
| fastText [87] OH-2.5 +ELI5 | **30.2** | **15.4** |

by `RefinedWeb`), and the Common Crawl-provided WET files that contain pre-extracted text. We then apply `RefinedWeb`'s heuristic quality filters to each of the text extractions. In Table 3, we find both `resiliparse` and `trafilatura` improve CORE by at least 2.5 points over the WET extraction. This is significant because most open source datasets, including `C4`, `RedPajama`, and `Dolma-V1`, use WET files, which could partially explain their worse performance in Table 2. While `resiliparse` and `trafilatura` have similar downstream performance, `resiliparse` is $8\times$ faster to run and hence more practical for large-scale processing. For more analysis, see Appendix K.

> **Takeaway:** For DCLM-POOL and the remaining experiments, we use `resiliparse` to extract text.

### 4.3 Deduplication

Web-crawled datasets often contain many duplicate or near-duplicate data strings. Removing these duplicates serves the dual purpose of improving performance by reducing memorization [35, 94] and increasing data diversity. For deduplication, we explore MinHash [29], as part of a suffix array pipeline [94, 127], and near-duplicate Bloom filtering, which modifies an exact document and paragraph deduplication scheme [156]. We find that both approaches provide comparable downstream performance: within 0.2 CORE percentage points at the `7B-2x` scale. However, our modified Bloom filter scales more easily to datasets surpassing 10TB. We provide additional analysis in Appendix L.

> **Takeaway:** We use a Bloom filter for DCLM-BASELINE and MinHash for other experiments.

### 4.4 Model-based quality filtering

Recent literature [28, 59, 156] indicates that using learnable models as quality filters leads to downstream improvements. In this section, we investigate model-based filtering.

**Comparing model-based filtering approaches.** We compare many strategies: 1) *PageRank score filtering* to retain documents based on how likely they are to be linked to other documents, 2) *Semantic Deduplication (SemDedup)* to remove documents with similar informational content [1], 3) *linear classifiers* fit on pre-trained BGE text embeddings [182], 4) *AskLLM* that prompts an LM to see if a

document is helpful [145], 5) *Perplexity filtering* where we retain low perplexity sequences following CCNet [176], 6) *Top-k average logits* where we average the top-$k$ model logits over all words in a document to score how confident a model is that the correct words are within $k$ reasonable choices, and 7) `fastText` [87] binary classifiers to distinguish data quality. For training classifiers, we train on $\sim$ 400k documents split equally between positive and negative classes. We experiment with different options for positive data and fix negative data as a random sample from a version of our `RefinedWeb` reproduction. For the perplexity filtering and the top-k average logits strategies, we utilize a 154M parameter causal Transformer trained on a mix of English Wikipedia, the books subset of `RedPajama-v1`, and peS2o [155, 166]. We compare the aforementioned approaches in Table 4 and find that `fastText`-based filtering outperforms all other approaches. We next aim to understand how `fastText` training recipes affect its effectiveness as a data filtering network [59].

**Text classifier ablations.** To better understand the limits of `fastText`, we train several variants, exploring different choices for the reference data (i.e., the examples given positive labels), feature space, and filtering threshold, as shown in Table 5. For reference positive data, we considered commonly used sources like Wikipedia [63], OpenWebText2 [63], and `RedPajama-books` [166], following the reference data used for GPT-3 [31]. We also try a novel approach, using instruction-formatted data, drawing examples from OpenHermes 2.5 [163] (OH-2.5) and high-scoring posts from the `r/ExplainLikeImFive` (ELI5) subreddit. Overall, we find, when controlling for other hyperparameters, the `fastText` OH-2.5 +ELI5 approach gives a 3.5 percentage point lift on CORE compared to the other more conventional choices. It is natural to ask whether using OH-2.5 data for filtering could preclude additional gains from instruction-tuning. In Appendix Q, we show this is not the case, further suggesting the strength and compatibility of this approach with modern finetuning paradigms. Finally, we observe that using a fairly strict threshold, which keeps the top-10% of examples, helps over more permissive top-15% and top-20% thresholds. We further study the unintuitive behavior of dataset filtering and its connection to human judgment in Appendix N.

Table 5: `fastText` **ablations** (7B-1x scale). We ablate choices for the positive data (top) and threshold (bottom). 'Dataset' is the positive set, while the negatives are randomly sampled our `RefinedWeb` reproduction. 'Threshold' is the percentile used for filtering based on `fastText` scores. "GPT-3 Approx" refers to a mix of Wikipedia, OpenWebText2, and RPJ Books, as in [31].

| Dataset | Threshold | CORE | MMLU | EXTENDED |
|---|---|---|---|---|
| OH-2.5 + ELI5 | 10% | **41.0** | **29.2** | 21.4 |
| Wikipedia | 10% | 35.7 | 27.0 | 19.1 |
| OpenWebText2 | 10% | 34.7 | 25.0 | 18.7 |
| GPT-3 Approx | 10% | 37.5 | 24.4 | 20.0 |
| OH-2.5 + ELI5 | 15% | 39.8 | 27.2 | **21.5** |
| OH-2.5 + ELI5 | 20% | 38.7 | 24.2 | 20.3 |

> **Takeaway:** For DCLM-BASELINE and the remaining experiments, we use `fastText` OH-2.5 + ELI5 classifier score to keep the top 10% of documents. The result of this filtering is DCLM-BASELINE.

### 4.5 Dataset mixing

Often, Common Crawl (CC) is combined with other data sources that are considered high-quality [63, 70, 166, 168] (e.g., Wikipedia, StackExchange, and peS2o [155]). Since DCLM participants can include additional data sources in our mixing track, we examined the potential benefits of adding high-quality sources to training sets derived from Common Crawl only. We compare a model trained on 100% filtered CC data to models trained with the mixing proportion from Llama 1 and `RedPajama`: 67% CC, and 33% from Wikipedia, Books, Stack exchange, arXiv, and Github. For the CC component, we consider different variants: a subset of our DCLM-BASELINE, `RedPajama`'s CC portion, `RefinedWeb`, and `C4`. The results in Table 6 show that mixing improves performance for the lower-performing CC subsets (`C4`, `RedPajama`-CC, and `RefinedWeb`). In the case of DCLM-BASELINE however, mixing actually hurts performance on average, which suggests it can be counterproductive given performant filtering. For additional mixing results, see Appendix M.

Table 6: **Mixing high-quality sources with subsets of CommonCrawl** (1B-1x scale). We evaluate the impact of mixing high-quality sources ('RPJ extras') with various datasets derived from CommonCrawl, using the mixing ratios from Llama/RPJ. Numbers in parentheses indicate the gain or loss in performance due to mixing compared to using only the base dataset.

| | CORE | | EXTENDED | |
| Dataset | Base | w/ RPJ extras | Base | w/ RPJ extras |
| --- | --- | --- | --- | --- |
| C4 | 23.7 | 25.9 (+2.2) | 12.5 | 13.3 (+0.8) |
| RPJ CC only | 24.0 | 25.7 (+1.7) | 12.1 | 13.5 (+1.4) |
| RefinedWeb | 25.1 | 26.5 (+1.4) | 12.9 | 13.1 (+0.2) |
| DCLM-BASELINE | 31.1 | 29.9 (−1.2) | 16.0 | 15.0 (−1.0) |

## 4.6 Decontamination

Here, we examine whether contamination of our pretraining data with evaluation data influences our results for DCLM-BASELINE. We focus on MMLU and Hellaswag as our evaluation sets of choice, given their popularity as metrics for language model performance at the 7B scale.

As an experiment, we attempt to remove examples from these two sets that exist in DCLM-BASELINE. For both, our strategy is to flag training documents that contain the question text along with one of the corresponding answer options. For these flagged examples, we then remove all matched question and option strings. In order to improve recall for MMLU, which contains some long passage-based questions, we opt to detect only the last sentence from each question, reducing the chance of missing questions due to formatting differences. Based on inspection, this still incurs many false positives. We then train a 7B-2x model with our DCLM-BASELINE without the detected overlaps. As seen in Table 7, this does not lead to decreases in model performance, so our performance gains on these two tasks are not likely to be caused by increased presence of their test examples in our dataset.

Table 7: **MMLU and Hellaswag overlap removal** (7B-2x scale). We remove overlaps detected with MMLU and Hellaswag, in cases where a question and one of its options are detected. We compare models trained before and after this decontamination step, and see that performance does not fall.

| Dataset | $E$ = MMLU | $E$ = Hellaswag |
| --- | --- | --- |
| DCLM-BASELINE | 51.8 | 77.9 |
| DCLM-BASELINE ($E$ removed) | 52.7 | 78.4 |

We also apply the above removal strategy for MMLU on Dolma-V1.7 [156] and FineWeb-Edu [106]. The results can be seen in Table 25 in Appendix O, from which we observe that DCLM-BASELINE has roughly similar contamination stats as these other high performing datasets. We also provide further analysis that extends to our entire evaluation suite in Appendix O.

# 5 Scaling up DCLM-BASELINE to the trillion token scale

Here, we test if datasets that perform well on the DCLM benchmark also maintain their strength with an order of magnitude more compute. To ensure our trained model is broadly useful, including for math and coding tasks, we combine our 3.8T DCLM-BASELINE with the StarCoder [96] and ProofPile2 [14] datasets to arrive at a 4.1T token dataset. We train a 7B model for 2.5T tokens on this dataset with the same hyperparameters as our largest competition scale except for two separate cool-downs phase for the 200B and 270B tokens on a modified distribution that was 70% DCLM-BASELINE with a tighter fastText threshold, and 30% math datasets (see Appendix Q). We then take a "model soup" of these two separate cool-downs[179]. Finally, we adopt the continual pretraining methodology from Pouransari et al. [132] for 100B tokens on the same distribution to increase the context length from 2048 to 8192 (see Appendix Q.2).

In Table 8, we show that our model outperforms all 7B models trained on public training sets and approaches closed-data models trained for more tokens such as Llama-8B, Mistral-7B, and Gemma-

7B. Additionally, in Appendix P, we show that our model achieves strong instruction-tuning (IT) performance. After instruction tuning on publicly available IT datasets, our model maintains most of its benchmark performance and achieves an AlpacaEval2.0 LC Win-rate of 16.6, which outperforms Gemma-Instruct (10.4), while approaching the strong performance of Mistral-v0.2-7B (17.1) and Llama3-Instruct (22.9). Finally, in Appendix Q.3, we show results from training a 1B model on 4.3T tokens from DCLM-BASELINE, StarCoder and ProofPile2 combined, resulting in a strong, small model that outperforms prior small models including Gemma-2B and Qwen2-1.5B.

Table 8: **State-of-the-art comparison** (beyond 7B–2x scale). We compare our final model with other 7–8B parameter models. DCLM-BASELINE yields a model that outperforms models trained on open datasets and is competitive with models trained on private datasets.

| Model | Params | Tokens | Open dataset? | CORE | MMLU | EXTENDED |
|---|---|---|---|---|---|---|
| **Open weights, closed datasets** | | | | | | |
| Llama2 | 7B | 2T | ✗ | 49.2 | 45.8 | 34.1 |
| DeepSeek | 7B | 2T | ✗ | 50.7 | 48.5 | 35.3 |
| Mistral-0.3 | 7B | ? | ✗ | 57.0 | 62.7 | 45.1 |
| QWEN-2 | 7B | ? | ✗ | 57.5 | **71.9** | 50.5 |
| Llama3 | 8B | 15T | ✗ | 57.6 | 66.2 | 46.3 |
| Gemma | 8B | 6T | ✗ | 57.8 | 64.3 | 44.6 |
| Phi-3 | 7B | ? | ✗ | **61.0** | 69.9 | **57.9** |
| **Open weights, open datasets** | | | | | | |
| Falcon | 7B | 1T | ✓ | 44.1 | 27.4 | 25.1 |
| OLMo-1.7 | 7B | 2.1T | ✓ | 47.0 | 54.0 | 34.2 |
| MAP-Neo | 7B | 4.5T | ✓ | **50.2** | **57.1** | **40.4** |
| **Models we trained** | | | | | | |
| FineWeb edu | 7B | 0.14T | ✓ | 38.7 | 26.3 | 22.1 |
| FineWeb edu | 7B | 0.28T | ✓ | 41.9 | 37.3 | 24.5 |
| DCLM-BASELINE | 7B | 0.14T | ✓ | 44.1 | 38.3 | 25.0 |
| DCLM-BASELINE | 7B | 0.28T | ✓ | 48.9 | 50.8 | 31.8 |
| DCLM-BASELINE + StarCoder + ProofPile2 | 7B | 2.6T | ✓ | **57.1** | **63.7** | **45.4** |

## 6   Conclusion and limitations

We introduced the DCLM testbed and demonstrated how it leads to new state-of-the-art training sets. Our exploration of the dataset design space is only the beginning and has clear limitations. Due to compute constraints, we could only ablate design dimensions individually and could not test all approaches at larger scales nor train models beyond 7B parameters. We also could not sufficiently explore run-to-run variation. Moreover, there are many variations of DCLM-BASELINE that we did not explore, such as alternatives to sharded deduplication and using differently trained filtering models. We also conducted most of our experiments with only one tokenizer (GPT-NeoX), and other tokenizers may perform better on multilingual tasks or math. Still, we hope that this paper is a starting point for further research on data curation that pushes the state-of-the-art beyond DCLM-BASELINE.

While models trained on DCLM-BASELINE are competitive on common language understanding tasks, they currently do not perform as well on code and math. We view this as a consequence of our focus on language understanding in the first version of DCLM, and not an inherent limitation of our benchmark or dataset. Prior work has shown that adding specific training data and post training methods for code and math can substantially improve models on those domains [14, 96, 175, 194, 199]; combining DCLM-BASELINE with these domain-specific training sets and extending DCLM to cover code and math are interesting future directions. Other important dimensions to expand DCLM along are fairness, multilinguality, and safety. We include some analysis in Appendix S and hope that our open-source testbed can strengthen data-centric research in these directions as well.

**Acknowledgements.** We would like to thank Lilith Bat-Leah, Loubna Ben Allal, Samy Bengio, Mia Chiquier, Adrien Gaidon, Lizzy Grant, Tom Gunter, Awni Hannun, Jonathan Hayase, Mike Lewis, Percy Liang, Ian Magnusson, Yifan Mai, Sewon Min, David Mizrahi, Praveen Paritosh, Guilherme Penedo, Kyle Richardson, Weijia Shi, Karanjeet Singh, Joshua Susskind, Oyvind Tafjord, Carl Vondrick, and Elle Wohlmuth, for helpful feedback at various stages of the project. We would like to thank Mike Garrison and Romil Shah for help with compute and infrastructure.

This research was supported by Allen Institute for AI, Open Philanthropy, Institute for Foundations of Machine Learning (IFML), AFOSR MURI grant FA9550-22-1-0380, Israeli Science Foundation (ISF) grant no. 2486/21, Singapore AI Visiting Professorship Programme AIVP-2024-001, Alon Fellowship, Adelis foundation, Israeli Council for Higher Education, Onassis Foundation - Scholarship ID: F ZS 056-1/2022-2023, NSF Grants AF 1901292, CNS 2148141, Tripods CCF 1934932, IFML CCF 2019844,, research gifts by Western Digital, Amazon, WNCG IAP, UT Austin Machine Learning Lab (MLL), Cisco and the Stanly P. Finch Centennial Professorship in Engineering, NSF Graduate Research Fellowship. MN acknowledges funding by the Federal Ministry of Education and Research of Germany under grant no. 01IS22094B WestAI - AI Service Center West.

We gratefully acknowledge compute budget granted by Gauss Centre for Supercomputing e.V. and by the John von Neumann Institute for Computing (NIC) on the supercomputers JUWELS Booster and JURECA at Jülich Supercomputing Centre (JSC)

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

# Appendix

## Table of Contents

# A Contributions

All authors are listed alphabetically by last name.

## Data processing

**DCLM-BASELINE**

Khyathi Chandu, Alex Fang, Saurabh Garg, Thao Nguyen, Vaishaal Shankar

**Decontamination**

Achal Dave, Saurabh Garg, Jeffrey Li, Vaishaal Shankar, Georgios Smyrnis

**Deduplication**

Achal Dave, Alex Fang, Jeffrey Li, Matt Jordan, Vaishaal Shankar

**Extra data sources**

Yonatan Bitton, Mayee Chen, Giannis Daras, Achal Dave, Alex Fang, Joshua Gardner, Maciej Kilian, Jeffrey Li, Niklas Muennighoff, Marianna Nezhurina, Vaishaal Shankar, Hanlin Zhang

## Baseline datasets

**Pre-processing existing datasets**

Achal Dave, Alex Fang, Samir Gadre, Reinhard Heckel, Sedrick Keh, Marianna Nezhurina, Vaishaal Shankar, Georgios Smyrnis

**Reproductions**

Amro Abbas, Hritik Bansal, Yonatan Bitton, Yair Carmon, Khyathi Chandu, Alex Fang, Dhruba Ghosh, Cheng-Yu Hsieh, Maor Ivgi, Matt Jordan, Sedrick Keh, Jeffrey Li, Kyle Lo, Luca Soldaini, Hanlin Zhang, Jieyu Zhang

## Data curation

**Filtering**

Hritik Bansal, Alex Fang, Saurabh Garg, Maor Ivgi, Matt Jordan, Jeffrey Li, Marianna Nezhurina, Vaishaal Shankar

**Rewriting**

Saurabh Garg, Maor Ivgi, Niklas Muennighoff

**Mixing**

Achal Dave, Jeffrey Li, Georgios Smyrnis

## Evaluations

**Evaluation framework**

Alon Albalak, Kushal Arora, Hritik Bansal, Achal Dave, Maor Ivgi, Sedrick Keh, Vaishaal Shankar, Rulin Shao, Rui Xin

**Evaluation runs**

Kushal Arora, Achal Dave, Alex Fang, Jeffrey Li, Sedrick Keh, Vaishaal Shankar

**Evaluation metrics**

Achal Dave, Alex Fang, Jeffrey Li, Vaishaal Shankar

## Training

**Training code**

Achal Dave, Samir Gadre, Suchin Gururangan, Kalyani Marathe, Jean Mercat, Hadi Pouransari, Sunny Sanyal, Georgios Smyrnis, Igor Vasiljevic, Mitchell Wortsman

**Data preprocessing**

Alex Fang, Matt Jordan, Vaishaal Shankar, Georgios Smyrnis

**Training runs**

Kushal Arora, Achal Dave, Alex Fang, Samir Gadre, Jenia Jitsev, Sedrick Keh, Jeffrey Li, Marianna Nezhurina, Vaishaal Shankar, Georgios Smyrnis

**Trillion-token models**

Alex Fang, Jeffrey Li, Hadi Pouransari, Vaishaal Shankar

**Instruction tuning**

Kushal Arora, Hritik Bansal, Aaron Gokaslan, Etash Guha, Niklas Muennighoff

## Competition setup

**Competition design**

Yair Carmon, Achal Dave, Alex Fang, Samir Gadre, Reinhard Heckel, Jeffrey Li, Ludwig Schmidt, Vaishaal Shankar

**Competition tooling**

Gabriel Ilharco, Maor Ivgi, Jeffrey Li, Sarah Pratt

**Paper writing**

Alon Albalak, Hritik Bansal, Yair Carmon, Achal Dave, Alexandros G. Dimakis, Alex Fang, Samir Gadre, Etash Guha, Reinhard Heckel, Maor Ivgi, Sedrick Keh, Jeffrey Li, Niklas Muennighoff, Sarah Pratt, Ludwig Schmidt, Vaishaal Shankar, Georgios Smyrnis

## Leadership and advising

**Advising**

Alaa El-Nouby, Fartash Faghri, Dirk Groeneveld, Reinhard Heckel, Jenia Jitsev, Sham Kakade, Pang Wei Koh, Thomas Kollar, Kyle Lo, Niklas Muennighoff, Sewoong Oh, Sujay Sanghavi, Luca Soldaini, Shuran Song, Alexander Toshev, Stephanie Wang, Luke Zettlemoyer

**Leadership**

Yair Carmon, Achal Dave, Alexandros G. Dimakis, Ludwig Schmidt, Vaishaal Shankar

**Project coordination**

Achal Dave, Ludwig Schmidt, Vaishaal Shankar

## B  Additional related work

Data curation methods have been proposed that can be grouped into two categories: methods that aim to enhance performance, and those with non-performance related goals. Performance-oriented methods include language detection, heuristics-based filtering, quality filtering, data deduplication, data mixing, and synthetic data. Non-performance-oriented filters include the removal of copyrighted text, toxic content, personally identifiable information (PII), opt-out, and evaluation data.

**Language detection.**  Language detection methods most often rely on a `fastText` classifier that has been trained to identify 157 languages [71, 127, 156], but past methods have also utilized other classifiers including a naive Bayes classifier [137]. When collecting multilingual datasets, another curation option is to filter web pages based on country domains or by selecting URLs that are correlated with data from certain languages [109, 130, 153].

**Heuristics.** It is widely known that web-scraped text contains high quantities of boilerplate HTML, error messages, stock tickers, and other undesirable data for training language models, much of which can be detected and removed by heuristic-based systems. The exact heuristics used by each data curation method vary but can be largely grouped into five categories: item count, repetition count, existence, ratio, and statistics. For example, Rae et al. [134] remove any lines that contain at least two of the following stop words: *the, be, to, of, and, that, have, with*, defining an item count heuristic. An example of a statistic might be removing documents that have a mean line length greater than 100 characters [37].

**Quality filtering.** Filtering for "high-quality" data (data which was written by humans and has likely gone through an editing process [8]) is a common step for data curation pipelines. The most commonly used method for quality filtering is to train a binary classifier on data from a perceived high-quality dataset (e.g. Wikipedia) and a perceived low-quality dataset (e.g. unfiltered web text) and filter out data where the classifier assigns sufficiently low scores [31, 53, 63]. A less commonly used method is to train a language model on the high-quality dataset and calculate perplexity on the data to be filtered, where high perplexity scores suggest that the data is lower quality [119, 176].

Recently, works have proposed the use of pretrained language models to identify and curate high-quality data through prompting for various dimensions of perceived quality [106, 145, 177]. Ankner et al. [12] even find that it's possible to use a small pretrained model (125M parameters) to prune training data for models as large as 3B. MiniPile [88] demonstrated that a 1M document subset of the Pile selected by clustering and removing low-quality clusters, can lead to small LMs that maintain performance on GLUE, while significantly reducing the scale of training data. RHO-1 [98] has a similar goal to quality filtering, but rather than filtering data out of the dataset, they propose Selective Language Modeling, an objective function that selectively masks the loss of tokens that are predicted to be low quality.

**Deduplication.** Deduplication has proven to be a beneficial step in almost every data curation pipeline. The methods used vary in complexity, including deduplication based on URLs, hashing, string metrics, and using model-based representations. URL deduplication has long been in use for deduplicating web snapshots [3]. Commonly used hash-based deduplication methods include Bloom filters [25, 156], suffix array-based methods [94, 127], and MinHash-based methods [30] such as MinHashLSH [31]. Model-based methods include SemDeDup [1] which embeds each point in a dataset, clusters data points together, and removes data points within clusters that are too similar, and D4 [165] which further applies the SSL prototypes method from Sorscher et al. [157] and removes the most prototypical example from each cluster.

**Data mixing.** When the training dataset is composed of data from multiple domains or sources (e.g. web text, Wikipedia, and books), then an additional challenge for data curation is to determine what percent of the final dataset comes from each source, known as data mixing. Methods for data mixing include using heuristics (such as human judgment) [63, 167], or empirically determining the best domain weights according to some downstream evaluation [53, 134]. More principled approaches have been proposed that are based on Group DRO [183], multi-armed bandits [7], and information theory [6]. Further methods have been proposed building off of these principled approaches, including DoGE [58], Skill-it [38], and ShearedLlama [181], each bringing some improvements. Thudi & Maddison [164] develop MixMax, a provably optimal method under a concave objective, which improves upon Group DRO-based alternatives but has not been proven at scales typical of language modeling. Ge et al. [64] propose BiMix, a unified scaling law that simultaneously models the behaviors of data quantity and mixing weights, using only small models as proxies to calculate the scaling laws.

**Synthetic data.** With the improvements in the ability of language models to accurately model text distributions, the generation of synthetic data has become an additional avenue for data curation. Notable methods for pretraining include the Phi models [2, 75], which generate synthetic textbook data from the GPT series of models, as well as WRAP [110] which uses a similar method to the Phi models, but demonstrates that the synthetic data generation pipeline is feasible with much smaller models (1.8B and 7B parameters). Beyond generating synthetic data for pretraining, Singh et al. [152] propose ReST$^{EM}$, a method for generating synthetic data for math and coding benchmarks, which uses binary feedback (eg. whether the code gives the correct output) to repeatedly filter self-generated

data. Similarly, Zelikman et al. [191] propose STaR, which bootstraps a dataset of rationales for commonsense question answering and mathematics datasets.

**Non-performance related methods** have been designed for a variety of purposes, including to remove copyrighted content [104, 150, 151], toxic speech [137, 170], private information [9, 107], opt-out data [107, 118] or to decontaminate data to avoid benchmark leakage [188]. While these methods are less relevant to this work, they are nonetheless important in real-world data curation pipelines.

## C   Benchmark rules

This section provides detailed guidelines for submissions in the two DCLM tracks.

### C.1   General rules

The following applies to both the filtering and mixing tracks.

1. Submissions should include documentation detailing their key components.

2. The dataset underlying a submission to the leaderboard, or fully working code to reproduce it, should be freely available to encourage reproducibility of submissions. Submissions that do not satisfy this requirements may still be accepted, but we will mark them as such in the leaderboard.

3. Tokenization must be performed with our provided script that tokenizes the data and performs a global shuffle.

4. Submissions cannot make any changes to the training or evaluation code.

5. Use of evaluation data (test data from our evaluation tasks) for purposes other than evaluation and decontamination is forbidden.

### C.2   Filtering track

The defining characteristic of entries in the filtering track is that they form the dataset by applying a processing pipeline on the subset of DCLM-POOL corresponding to the chosen compute scale (see Table 1) without including any external data. The rationale behind this requirement is twofold. First, the size and quality of initial data for filtering affects both the processing cost and the quality of the processed dataset. By fixing the initial dataset we level the playing field and allow comparison to focus on core curation techniques. Second, we wish to encourage the development of methods potentially relevant even at frontier-model scale. Using the 7B-2x pool (containing roughly 16T tokens) for the 400M-1x compute scale (requiring roughly 8B tokens for training) would allow filtering strategies that keep less than 0.1% of the data and cannot scale to generating a trillion-token dataset.

As we wish to encourage creative and performant submissions, our requirement for using only DCLM-POOL comes with the following qualifications:

1. **Modifying HTML extraction.** We create DCLM-POOL by extracting text from Common Crawl archives using `resiliparse`, which eases the computational burden on participants who may not have resources to extract text themselves. However, we additionally specify the Common Crawl archives for each pool to allow experimentation with text extraction. Participants may either start with our parsed DCLM-POOL data or work directly with the relevant Common Crawl WARC archives.

2. **Using models trained on external data.** We allow the DCLM-POOL processing pipeline to leverage models for quality filtering, paraphrasing, etc. These models may be trained on external data with the exception of evaluation data as per the general guidelines. We will not accept submissions abusing this allowance to introduce external data via a backdoor, e.g., by "paraphrasing" documents from DCLM-POOL into memorized data.

### C.3 Mixing track

In the mixing track, participants are free to use any data source, provided it meets the general guidelines by being freely available and not including evaluation data. Submissions to the mixing track should clearly document their data sources, the weight given to each source, and the ratio of tokens used for training (fixed for every benchmark scale) to the overall custom pool size.

## D Tooling

**Download.**    For the construction of our pool, we download WARC files from Common Crawl, and process them via `resiliparse`, we do this by streaming data directly from S3 to EC2 using the Ray data processing framework. This is the starting point for our data processing pipeline. For the dataset released to participants, we release various sizes of DCLM-POOL, that we make available for download. For details on the data, see Appendix E.

**Processing.**    Given raw pool of text, it is often useful to define a processing pipeline to clean, modify and filter it. We provide a robust framework to do that at scale, by sharding the pool and processing it in parallel. Namely, to process a pool one needs to define a sequence of *Mappers*, each taking a single document with its associated metadata as input, and output a list of documents. Our mappers include:

1. **Filters** which either retain or discard the input document according to some filtering criteria such as having a maximum or minimum length.
2. **Enrichers** which always return a list of documents with the page content as is, adding additional information to the metadata, such as detected language or number of tokens.
3. **Modifiers** change the content of the text itself, and can also split the document to create several new documents. This is useful for example, as a participant may design a function to remove padding white-space.

In particular, we implement all mappers used in RefinedWeb (which includes those from Gopher as a subset) and C4 along with many new ones, and allow users to integrate custom mappers into their pipeline. Additionally, while mappers allow for document-level processing, in some cases it may also be necessary to execute corpora-level operations. For instance, a user may wish to deduplicate spans that appear in several documents. Our tooling also supports global functions that depend on all documents.

**Contamination Analysis.**    We use the tools provided by Lee et al. [94] as a base and adapt them to evaluate the contamination of our training set with the evaluation sets. As done in Touvron et al. [168], we measure the number of tokens that appear in the same consecutive sequence of at least 10 tokens, between a training sample and an evaluation sample. With this number, we calculate how many tokens on average per evaluation sample are "contaminated", appearing both in the training and the evaluation data.

**Tokenization and shuffling.**    Once documents have been mapped, filtered, or globally processed, we provide standardized code to tokenize and shuffle data. The output of this code is a trainable dataset artifact. For tokenization, our code uses the GPT-NeoX [24] tokenizer. Our tokenization code adopts Ray [1] for cluster management and scales from a single node setups for small datasets to multiple nodes for larger ones. After tokenizing, we perform a global shuffle of our dataset.

**Training Setup.**    We base our training code on OpenLM [76], and provide configuration files for each of our scales. We also provide scripts that train models using each configuration, and produce `json` files that describe a trained model in detail. For further training details, see Appendix F.

**Evaluation.**    We base our evaluation pipeline on the evaluation tasks provided by LLM-foundry [116]. Using one of the aforementioned model `json` files as input, our tools evaluate the associated checkpoint on all of our tasks. A new `json` file is then produced, including the evaluation results in each task, as well as aggregate metrics. This `json` file can then be submitted via a pull request to submit the results to our leaderboard.

---

[1] https://github.com/ray-project/ray

**Reproducibility.** All of our results, including data processing, model training, evaluations, and plots included in this paper, are reproducible using our open-source framework and the recipes in https://datacomp.ai/dclm. We list compute requirements for our code in Appendix R. We provide a list of all 416 experiments at https://github.com/mlfoundations/dclm/blob/main/assets/DCLM_model_database.csv.

# E DCLM-POOL

DCLM-POOL was collected by taking all 5.1M Common Crawl WARC dumps from 2013 to 2022 (inclusive) and extracting text from the HTML using the resiliparse framework. We opted to omit 2023 and above to prevent large amounts of language model generated text from polluting our datasets and to provide a hold out for future use. The entirety of DCLM-POOL is hosted by Common Crawl[2] while the competition subsets are released HuggingFace with CC-BY-4 license. We release DCLM-POOL as a set of .jsonl files similar to Dolma-V1 and RedPajama. We provide the fields that are in the .jsonl in Table 9. The entire pool is 5.1M gzip compressed .jsonl files which take up 340TB in total on disk. The use of this dataset is also subject to CommonCrawl's Terms of Use: https://commoncrawl.org/terms-of-use.

Common Crawl respects robots.txt, and thus our pool does so as well, giving content creators a mechanism to opt out of Common Crawl and DCLM-POOL. Since DCLM-POOL is a large subset of Common Crawl it will contain some PII data, however Common Crawl does honor deletion requests and periodically redacts dumps. We designed DCLM-POOL to maintain a one-to-one mapping between raw Common Crawl WARC files and DCLM-POOL .jsonl files, allowing us to update DCLM-POOL based on redactions.

We note that Common Crawl includes raw data as collected from the web without filtering. While some of our pools, such as DCLM-BASELINE, underwent some filtering of malicious URLs, none have had any special treatment for PII and sensitive content to preserve representativeness of the raw data. For a more complete discussion on PII and consent regarding our pools, see Appendix U.

Table 9: Metadata provided in DCLM-POOL data.

| Label | Additional notes |
|---|---|
| metadata.Content-Length | Length of the content. |
| metadata.Content-Type | Type of the content. |
| metadata.WARC-Block-Digest | Digest for data integrity. |
| metadata.WARC-Concurrent-To | Related WARC record. |
| metadata.WARC-Date | Date of the WARC record. |
| metadata.WARC-IP-Address | IP address of the source. |
| metadata.WARC-Identified-Payload-Type | Identified payload type. |
| metadata.WARC-Payload-Digest | Payload digest for integrity. |
| metadata.WARC-Record-ID | Unique ID of the WARC record. |
| metadata.WARC-Target-URI | Target URI of the record. |
| metadata.WARC-Type | Type of WARC record. |
| metadata.WARC-Warcinfo-ID | Related warcinfo record ID. |
| text | Text content. |
| url | URL of the source. |
| warcinfo | Information about the WARC file. |

---

[2]https://data.commoncrawl.org/contrib/datacomp/index.html

Table 10: **Main models and hyperparameters used in our investigation.** For each scale, we list the number of layers $n_{\text{layers}}$, number of attention heads $n_{\text{heads}}$, model width $d_{\text{model}}$, and width per attention head $d_{\text{head}}$. Batch sizes are global and in units of sequences. Each sequence has 2,048 tokens.

| Scale | $n_{\text{layers}}$ | $n_{\text{heads}}$ | $d_{\text{model}}$ | $d_{\text{head}}$ | Warmup | Learning rate | Weight decay | z-loss | Batch size |
|---|---|---|---|---|---|---|---|---|---|
| 400M-1x | 24 | 8 | 1,024 | 128 | 2,000 | 3$e$-3 | 0.033 | 1$e$-4 | 512 |
| 1B-1x | 24 | 16 | 2,048 | 128 | 5,000 | 3$e$-3 | 0.033 | 1$e$-4 | 256 |
| 3B-1x | 32 | 32 | 2,560 | 128 | 5,000 | 3$e$-3 | 0.033 | 1$e$-4 | 256 |
| 7B-1x, 7B-2x | 32 | 32 | 4,096 | 128 | 5,000 | 2$e$-3 | 0.05 | 5$e$-6 | 2,048 |

# F    Training details

**Overview.**    Our training setup follows closely that of Wortsman et al. [180] and Gadre et al. [62]. Specifically, we build our training infrastructure using the OpenLM [76], which supports decoder-only, pre-normalization Transformers [171], following an architecture inspired by GPT-2 [133] and Llama [167]. OpenLM is a PyTorch [13, 124] code-base that targets FSDP modules for distributed training [197].

**Architecture details.**    We utilize LayerNorm [15] without bias parameters for all normalization, qk-LayerNorm [51] on queries and keys for training stability, SwiGLU [149] multilayer perceptrons (MLPs), and a depth-scaled initialization scheme following Zhang et al. [193]. Our sequence length, during pretraining is 2048. We pack multiple sequences into batches to fill the entire context, with an EOS token to split documents. We allow causal attention to attend across documents; we experimented with masking attention across documents but early experiments indicated little impact on downstream performance.

**Training sets and tokenization.**    Since the focus of our paper is dataset development, we train on over 270 data distributions, mostly filtered from Common Crawl. For the majority of our experiments we use GPT-NeoX [24] for tokenization, which yields a vocabulary size of 50k.

**Optimization details.**    As metioned in the main body, we train with a standard next-token prediction objective. Following Chowdhery et al. [39], we employ z-loss to encourage output logit magnitudes to remain in a numerically stable range.

**Hyperparameters.**    We detail the hyperparameters for our models in Table 10. For the 400M-1x and 1B-1x, we follow hyperparameters from [62], which were tuned to optimize perplexity on a validation set containing tokens from recent arXiv papers, the OpenLM codebase itself, and news articles. For the 1B-1x scale, we also investigated alternative hyperparameters in Table 12, and find the hyperparameters from [62] perform best. For the 7B-1x and 7B-2x, we used a higher learning rate, and a lower weight decay, guided by the hyperparameter sweep in Table 11. We use a cooldown of 3$e$-5 for all experiments. For Table 2, we trained with a lower learning rate following [62] as these experiments were performed before our sweep. Specifically, we used a learning rate of 3$e$-4 and weight decay of 0.33.

Table 11: **Learning rate and weight decay sweep** (7B-1x scale). We evaluated the impact of learning rate and weight decay on an earlier iteration of DCLM-BASELINE. Based on this sweep, we specify the settings for Table 10 for the 7B-1x and 7B-2x scales.

| LR | WD | CORE |
|---|---|---|
| 1$e$-03 | 0.1 | 44.1 |
| 2$e$-03 | 0.05 | **44.8** |
| 3$e$-03 | 0.033 | 44.7 |
| 1$e$-02 | 0.01 | 43.8 |

# G  Evaluation details

Below we outline the tasks we used to evaluate our models in LLM Foundry. We also examine the LightEval [60] evaluation pipeline used in the FineWeb-Edu [106] evaluations.

## G.1  Evaluation Tasks

We divide our evaluations into two high-level categories: CORE (22 tasks) and EXTENDED (53 tasks). The set of CORE tasks were selected due to their ability to provide a low variance signal of learning, even at small scales. We include a diverse range of tasks aimed at assessing a variety of model capabilities.

**CORE tasks.**

- The AGI Eval LSAT-AR dataset [200] (3-shot, 230 examples) tests for model knowledge in the legal domain and evaluates analytical reasoning capabilities.

- The ARC easy (2376 examples) and ARC challenge (1,172 examples) datasets [43] (10-shot) contain four-way multiple choice questions taken from grade 3-9 science exams, where questions in the easy dataset require knowledge of basic science, and the challenge questions require some procedural reasoning.

- We use a series of 6 datasets from Big-Bench [18] (all 10-shot): (1) QA Wikidata (20,321 examples) which requires models to complete factual statements with the correct answer, (2) Dyck languages (1,000 examples) where the model needs to complete a partially balanced expression consisting of parentheses and braces, (3) Operators (210 examples) where the model is given some newly defined operators and asked to compute the output from some expression using those operators, (4) Repeat Copy Logic (32 examples) which requires the model to differentiate instructions from text-to-copy and to perform a sequence of operations, (5) CS Algorithms (1,320 examples) which requires the model to execute algorithms such as recursion and dynamic programming, and (6) Language Identification (10,000 examples) where the model is expected to identify the language of a sequence of natural language text.

- BoolQ [41] (10-shot, 3,270 examples) is a binary question answering dataset where the model is expected to answer questions about relevant passages.

- CommonsenseQA [159] (10-shot, 1,221 examples) is a 5-way multiple choice question answering dataset which evaluates the models ability to understand and apply commonsense knowledge on everyday scenarios.

- COPA [142] (0-shot, 100 examples) consists of causal reasoning questions where the model is given two possible outcomes to a scenario and must use commonsense to select the outcome that is more likely.

- CoQA [140] (0-shot, 6,304 examples) is a conversational question answering dataset where the model is given a passage and conversation between two participants and then expected to extract an answer from the passage to a question from one of the participants.

- HellaSwag [192] (0-shot and 10-shot, 10,042 examples) is a 4-way multiple choice commonsense reasoning dataset, where the model is required to understand implicit context and common knowledge in order to correctly select the continuation to a context.

- Jeopardy [89] (10-shot, 2,117 examples) is a dataset of questions posed in the format of the "Jeopardy!" quiz show, covering a wide variety of topics.

- LAMBADA [122] (0-shot, 5,153 examples) is a collection of narratives where a human is able to guess the final word of the narrative, but is not able to if they are only given the final sentence. To perform well on this task requires the model to attend to context from the full narrative and cannot simply rely on the local context.

- OpenBookQA [114] (0-shot, 500 examples) is a 4-way multiple choice question answering dataset that requires the model to use multi-step reasoning and commonsense knowledge.

- PIQA [23] (10-shot, 1,838 examples) is a binary multiple choice question answering dataset that requires the model to use physical commonsense reasoning to answer correctly.

- SQuAD [139] (10-shot, 10,570 examples) is a question answering dataset where the model is given a question and a passage containing the answer to that question.
- The Winograd Schema Challenge [95] (0-shot, 273 examples) is binary multiple choice pronoun resolution task where the model is given a context and asked to determine which entity a pronoun refers to, requiring the model to exhibit commonsense knowledge and contextual understanding.
- The Winogrande [146] (0-shot, 1,267 examples) dataset extends the Winograd Schema Challenge dataset by expanding the dataset to a wider variety of domains.

**EXTENDED tasks.**

- We use a series of 4 additional tasks from the AGI Eval suite of datasets [200] (all 3-shot): (1) LSAT-LR (510 examples) and (2) LSAT-RC (268 examples) test for model knowledge in the legal domain and evaluate logical reasoning and reading comprehension, respectively, (3) SAT-En (206 examples) evaluates the model's capabilities in English, and (4) SAT-Math (220 examples) evaluates the model's capability in math using chain-of-thought prompting.
- AQuA [99] (3-shot, 245 examples) is a 4-way multiple choice question answering dataset that evaluates the model on algebra questions using chain-of-thought prompting.
- BBQ [123] (3-shot, 55,006 examples) is a multiple choice question answering dataset designed to detect model's biases along nine social dimensions.
- We use a series of 9 additional datasets from Big-Bench [18] (all 10-shot): (1) Conceptual Combinations (103 examples) which evaluates the model's capability to parse conceptual combinations by selecting sentences where these combinations are used correctly, (2) Conlang Translation (164 examples) where the model is expected to deduce a new translation from English to an obscure constructed language based on a limited number of translation examples, (3) Elementary Math QA (34,313 examples) which is a multiple choice question answering dataset of simple quantitative reasoning problems, (4) Logical Deduction (1,500 examples) which requires a model to parse, understand, and apply information about objects and relationships between objects to infer new information, (5) Misconceptions (219 examples) evaluates whether a model can discern popular misconceptions from truth, (6) Novel Concepts (32 examples) measures the models ability to creatively construct a necessary abstraction that is unlikely to have existed in training data, (7) Strange Stories (174 examples) measures a model's capacity for Theory of Mind, (8) Strategy QA (2,289 examples) is a test that requires a model to answer questions requiring multi-step implicit reasoning, (9) Understanding Fables (189 examples) which evaluates the model's capability to understand the moral of a short story.
- Enterprise PII classification [126] (10-shot, 3,395 examples) is a binary classification task that evaluates whether a model can detect PII (e.g. usernames, emails) within text.
- GPQA-main (448 examples) and GPQA-diamond (198 examples) [141] (5-shot) are 4-way multiple choice question answering datasets written by domain experts in biology, physics, and chemistry, which are intended to be very difficult for non-experts to answer (even with access to the web). The diamond set is a high-quality subset including only questions where two experts answer correctly, but most non-experts answer incorrectly.
- GSM8K [44] (3-shot, 1,319 examples) is a dataset of grade school math word problems that requires between 2 to 8 steps to solve, where the model uses chain-of-thought prompting.
- LogiQA [100] (10-shot, 651 examples) is a 4-way multiple choice question answering dataset that evaluates logical reasoning.
- Math QA [11] (10-shot, 2,983 examples) is a 5-way multiple choice question answering dataset that evaluates math word problem solving capabilities, built on top of AQuA.
- MMLU [78] (0-shot and 5-shot, 14,042 examples) is a 4-way multiple choice question answering dataset that covers 57 different domains and tasks, evaluating both world knowledge and problem solving capabilities.
- PubMedQA [84] (10-shot, 1,000 examples) is a 3-way multiple choice question answering dataset which evaluates the model's ability to answer biomedical research questions given context from a relevant research article.

- Simple arithmetic with spaces and without spaces [116] (10-shot, 1,000 examples) are datasets consisting of simple arithmetic problems with up to 3 operations using numbers with up to 3 digits, evaluating a model's ability to follow the correct order of operations and perform arithmetic.
- Social Interaction QA [147] (10-shot, 1,954 examples) is a binary multiple choice question answering dataset that evaluates a model's social commonsense intelligence.
- SVAMP [125] (3-shot, 300 examples) is a set of challenging elementary-level math word problems that uses chain-of-thought prompting.
- Trivia QA [86] (3-shot, 11,313 examples) is an open-ended question answering dataset that evaluates the world knowledge of a model.
- The Winogender male and Winogender female datasets [144] (10-shot, 60 examples) are variants of the winograd schemas method that creates a minimal pair of sentences that differ only by the gender of one pronoun, designed to evaluate a model's gender bias.

### G.2 LightEval

Given the multiple ways of evaluating accuracy [57], we conducted a miniature study using the LightEval evaluation framework [60]. Notably, under this framework, we are able to achieve scores above random (25%) for 0-shot MMLU for 1B models by considering the log-probabilities of entire answer passages as opposed to single letters. The 1B Hugging Face model trained on FineWeb-Edu [106] has shown to work well on this, so we wanted to more closely examine how LightEval evaluation scores correlate with evaluation scores from LLM Foundry. We present our findings in Figure 5.

The key difference between LightEval and LLM Foundry for multiple choice tasks like MMLU is that LightEval considers the log probabilities of entire answer sequences, whereas LLM Foundry only considers log probabilities of single letters. Nonetheless, Figure 5 shows a positive correlation between the two evaluation frameworks on MMLU 0-shot accuracy.

In Figure 5 we were able to reproduce the MMLU scores reported in the FineWeb-Edu blog [106]. Notably, we found that LightEval indeed gave MMLU scores above random for the 1B scales, whereas in LLM Foundry, all the 1B models have accuracies around 0.25. At larger scales, however, the LightEval scores for the models become quite cramped together, which may make it more difficult to compare models and may make the comparisons more susceptible to noise. For example, the models Gemma-7B, Llama3-8B, and Mistral-7B all have scores between 0.43 and 0.44 in LightEval, while their scores range from 0.56 to 0.62 for LLM Foundry. We also see that FineWeb-Edu 7B-2x and DCLM 7B-2x perform quite similarly in LightEval, but DCLM-7B is better by close to 10 points in LLM Foundry. In conclusion, we believe that LightEval can be potentially a good choice when evaluating smaller models, but other frameworks like LLM Foundry could give clearer signals when comparing larger models.

One limitation of this study is that we took MMLU as a representative task, and we did not evaluate on other tasks. In the future, it would be interesting to compare with additional tasks, as well as additional frameworks like Eleuther LLM Harness.

## H    Hyperparameter study

A potential concern is that differences in the training recipe can change conclusions about which dataset is optimal, due to interaction between training hyperparameters and dataset distributions. To address this confounder, we show in Table 12 that orderings between datasets are preserved for various combinations of weight decay and learning rate. Moreover we find that performance gains from optimal hyper-parameter choice and dataset design tend to be orthogonal and complement each other. We illustrate this effect in Table 13.

## I    Architecture ablations

Similar to our hyperparameter study in Appendix H, here we explore whether our results also generalize across different architectures. We train models using two alternative architectures and

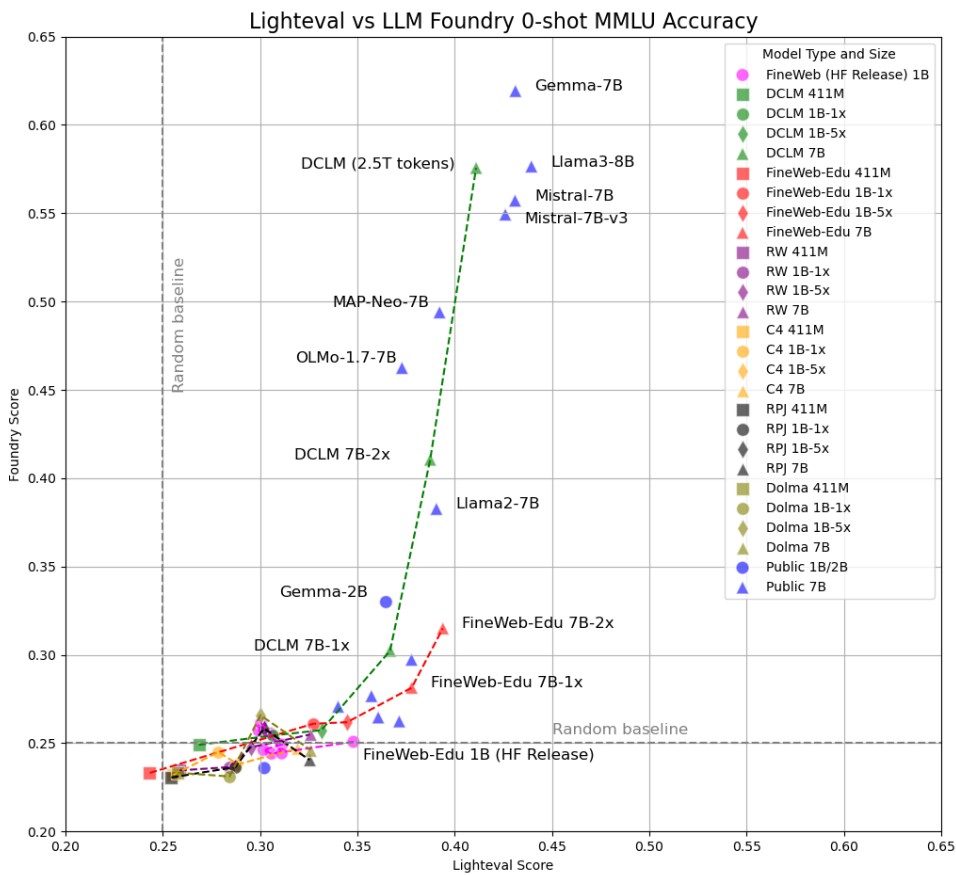

Figure 5: **Comparisons Between LightEval MMLU scores (x-axis) and LLM Foundry MMLU scores (y-axis).** LightEval is able to provide signal (i.e. score above random baseline) earlier for weaker models, but the LightEval scores at larger scales appear to be capped at a much lower threshold and are more closely clumped together.

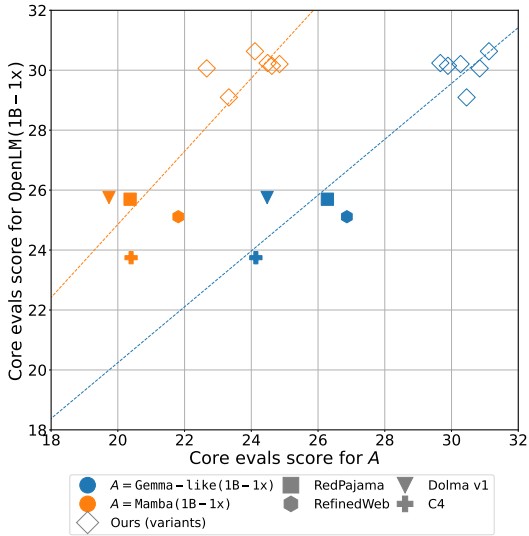

Figure 6: **Performance of datasets in architecture variants versus our original architecture.** We see that there is high correlation between the performance of a model trained with either a Gemma-like architecture or a Mamba-like architecture and the performance of the corresponding OpenLM one. This means that dataset improvements are consistent across model changes.

Table 12: **Rankings are stable across hyperparameters** (1B-1x scale). We train models on 3 datasets with 5 hyperparameter settings, varying learning rate and weight decay settings. Across the hyperparameter settings, the dataset ranking remains largely stable, with DCLM-BASELINE outperforming RedPajama, which in turns outperforms C4. With improved hyperparameters, the gaps between the datasets grows: e.g., at 'Default' (the best hyperparameter setting), DCLM-BASELINE outperforms RedPajama by 4.5 points and RedPajama outperforms C4 by 2 points, while at '0.1x Learning Rate' (the lowest performing setting), the gaps reduce to 3.3 points and 0.9 points respectively. Note: When changing learning rate, we also update weight decay so the product of the two remains the same.

| Hyperparameters | Dataset | Learning Rate (LR) | Weight Decay (WD) | CORE | EXTENDED |
|---|---|---|---|---|---|
| 0.1x Learning Rate | C4 | 3.0e-04 | 3.3e-01 | 22.1 | 11.6 |
| | RedPajama | 3.0e-04 | 3.3e-01 | 23.0 | 11.8 |
| | DCLM-BASELINE | 3.0e-04 | 3.3e-01 | **26.3** | **14.4** |
| 0.1x Weight Decay | C4 | 3.0e-03 | 3.3e-03 | 21.8 | 11.6 |
| | RedPajama | 3.0e-03 | 3.3e-03 | 23.0 | 11.8 |
| | DCLM-BASELINE | 3.0e-03 | 3.3e-03 | **28.3** | **15.1** |
| Default | C4 | 3.0e-03 | 3.3e-02 | 23.7 | 12.5 |
| | RedPajama | 3.0e-03 | 3.3e-02 | 25.7 | 13.5 |
| | DCLM-BASELINE | 3.0e-03 | 3.3e-02 | **30.2** | **15.4** |
| 10x Weight Decay | C4 | 3.0e-03 | 3.3e-01 | 21.8 | 12.0 |
| | RedPajama | 3.0e-03 | 3.3e-01 | 22.5 | 11.9 |
| | DCLM-BASELINE | 3.0e-03 | 3.3e-01 | **27.1** | **14.0** |
| 10x Learning Rate | C4 | 3.0e-02 | 3.3e-03 | 22.7 | 12.3 |
| | RedPajama | 3.0e-02 | 3.3e-03 | 26.0 | 13.2 |
| | DCLM-BASELINE | 3.0e-02 | 3.3e-03 | **29.0** | **15.0** |

Table 13: **Improvements from better hyperparameters stack with better datasets.** (7B-1x scale). We evaluate the impact of the most influential step in our dataset design, model based filtering ('fastText filtering'), stacked with a better hyperparameter setting. We see that for both MMLU and CORE benchmarks, the two inteventions (better dataset and better hyperparmaeters) seem to be orthogonal and stack on top of each other.

| Hyperparameters | fastText filtering | LR | WD | MMLU | CORE |
|---|---|---|---|---|---|
| Low LR | ✗ | 3.0e-04 | 0.33 | 25.4 | 38.3 |
| | ✓ | 3.0e-04 | 0.33 | **29.2** | **41.0** |
| High LR | ✗ | 1.0e-03 | 0.1 | 25.5 | 39.7 |
| | ✓ | 1.0e-03 | 0.1 | **38.3** | **44.1** |

examine the correlation of the results with the performance on the OpenLM architecture used in the rest of the paper. Specifically, we try the following:

- First, we implement a Gemma [160] inspired variant of our base architecture, where we change the activation to GeGLU and include an RMS normalization layer. Here, we aim to test the effect of small changes to the architecture, while keeping the fundamental design the same (i.e., this variant is still a decoder-only, transformer based model).

- Our second choice is the Mamba [74] architecture. Here, we aim to see how our datasets perform when the workings of the model change drastically (from a regular decoder-only model to a state-space model).

As shown in Figure 6, there is high correlation between the performance of a dataset when using the above two architecture variants and its performance for the original OpenLM architecture. This implies that the dataset improvements generalize across architectures.

# J Model-based quality filters

We presented results for several different model-based quality filters in Section 4.4. In this section, we describe their implementations in further detail, focusing especially for `fastText` classifiers which were our method of choice for DCLM-BASELINE.

## J.1 `fastText` classifiers

**Training**   We use the supervised `fastText` package from Joulin et al. [87] to train models to classify between chosen "high-quality" *reference data* which are given positive labels, and *web-crawled data* which are given negative labels. We then apply these classifiers to score each document from the pool we wish to filter, taking the predicted probability of the positive label as the score and computing a percentile-based threshold. In terms of training hyperparameters, we mostly used the default choices from the `fastText` package; the only hyperparameter change that we tried was to expand the feature space from unigrams only to both unigrams and bigrams (via setting the `wordNgrams` argument to be 2 instead of the default 1). This helped improve the quality of downstream filtered datasets, as shown in Table 14 (which extends Table 5 from Section 4.4).

Table 14: `fastText` **feature-space ablation** (7B-1x scale). Adding bigrams to the feature space helps over the default setting of unigrams only.

| Dataset | Threshold | Features | CORE | MMLU | EXTENDED |
|---|---|---|---|---|---|
| OH-2.5 + ELI5 | 10% | Unigrams + Bigrams | **41.0** | **29.2** | 21.4 |
| OH-2.5 + ELI5 | 10% | Unigrams | 40.0 | 28.3 | **22.1** |

**Data preparation.**   The bulk of our experimentation for training `fastText` models focused on constructing their underlying training sets, specifically the positively labeled reference data. For each experiment, we fixed the size of the training set to be 400K examples (i.e., 200K positive, 200K negative). The negatively labeled examples were sampled randomly from a set of documents that came from an earlier (smaller) version of our `RefinedWeb` reproduction. This version used `trafilatura` as the extractor instead of `resiliparse`, which we hypothesize might actually help for training the filtering model; as shown in Appendix K, `trafilatura` more aggressively removes boilerplate content that may appear in many pages (especiall from the same website). This type of content, if left in, may lead to the `fastText` models over-relying on these "spurious" features instead of the main contents of the page. For the positively labeled reference data, we tried several different sources, some of which involved further pre-processing:

- *Wikipedia.* We use the processed version from `RedPajama` [166] and apply English filtering by only keeping pages from the `en.wikipedia.org` domain. To encourage the classifier to rely on the core content of each page, we remove occurrences of the section titles `"See Also"` and `"References"`, at least one of which occurs in 90% of articles.

- *OpenWebText2.* We use this dataset as is, taken from the version in The Pile [63].

- *GPT-3 Approx.* We mix together Wikipedia and OpenWebText2 along with the books source from `RedPajama` [166]. Given the long length of individual books, we instead define examples by extracting chunks of text that are at most 2048 tokens long.

- OH-2.5 + *ELI5.* Our goal for this mix was to source instruction and question-answer formatted data that is both high-quality and covers a wide range of potential topics. We sample 100K examples from OH-2.5, which we do not further pre-process. For ELI5, each raw page from the `r/ExplainLikeImFive` subreddit contains a *post* asking a specific question and then some number of *comments* aiming to answer said question. We curate examples for training `fastText` models by taking a post and combining it with the top-scoring answer (using the karma score derived from community up/down-votes). If there

are ties, the longest answer is chosen. We also filter these examples by keeping only those where the post has score $\geq 0$, the best comment has score $\geq 5$, and there are at least 3 comments total.

## J.2 Other quality filtering baselines

We also examined other quality filters, though found none as effective as the `fastText` methods described above, as shown in Table 4. We now provide further details for some of these baselines.

**PageRank.** An intuitively promising, but ultimately unfruitful approach was to consider page centrality metrics such as PageRank and Harmonic centrality metrics, with the idea that more "central" web text would yield higher quality data. We collected PageRank metrics from Common Crawl's host level webgraph dataset[3] and omitted any hosts that did not appear in the crawl. Next we partitioned our `RefinedWeb` reproduction into quintiles based on their PageRank score and trained several models at the `1B-1x` scale. These results are collated in Table 15, but unfortunately no quintile performed better than a pool sampled from the union of all quintiles.

Table 15: **PageRank-based filtering** (`1B-1x` scale). Using PageRank score to select data is not helpful for improving upon our `RefinedWeb` reproduction. Using any quintile based on this score performs worse than a random sample from the same initial pool.

| Quintile | All | 1 | 2 | 3 | 4 | 5 |
|---|---|---|---|---|---|---|
| CORE | **27.8** | 26.1 | 27.3 | 26.6 | 26.3 | 27.1 |

**AskLLM.** A recent line of work studies using instruction-tuned models as annotators to determine the potential usefulness of a document. Sachdeva et al. [145] proposed AskLLM, in which the authors prompted Flan-T5 models [40] to evaluate whether the given document *". . . contain[s] informative signal for pretraining a large-language model? An informative data point should be well-formatted, contain some usable knowledge of the world, and strictly NOT have any harmful, racist, sexist, etc. content."*. We implemented this method ourselves, testing several models as annotators, different settings for maximal sequence length, and several prompts on a small scale. We found that the best configuration was using `Mistral-7B-Instruct-v0.2` [83], clipping the document at 1024 tokens, and taking the cumulative probabilities of *Yes* and *yes* tokens as the model score. We used the following prompt template:

```
Evaluate the following paragraph for LLM pretraining suitability:
- Is it well-structured or contains useful examples to natural
texts?
- Does it offer insights, useful facts or relevant information?
- Does it teach how to comply with open-ended tasks such as writing
letters, poems, emails etc.?
- Is it free from harmful content?

If most criteria are met based on the content below, indicate 'yes'
for suitable. Otherwise, indicate 'no' for unsuitable.

### <input> ###

Is it suitable for LLM pretraining? OPTIONS:
- yes
- no
```

where `<input>` is replaced with the document tokens, clipped if too long, and appended with "`...[The rest of the paragraph is omitted]`" in such cases. While this method worked

---

[3]https://commoncrawl.org/web-graphs

slightly better than random sampling from our pool (see Table 4), it significantly underperformed compared to our `fastText` experiments. Considering the high costs associated with applying it at scale, we did not perform this experiment on a larger scale.

**Semantic deduplication.** Following the success of the different deduplication methods we used (Appendix L), we studied the effect of Semantic deduplication as proposed by Abbas et al. [1]. In this approach, the authors propose embedding the documents using pre-trained language models, clustering them using k-means, and removing all but one document from each group of closely related documents to encourage diversity in the dataset. We began by embedding each document in a pool of approximately 100 million documents (following Abbas et al. [1]'s best practices) with BGE-base [182]. We then used `faiss-GPU` [85] to perform spherical k-means clustering, with 20 iterations and $K = 11000$. We sampled documents after discarding 25% of the data. As seen in Table 4, this intervention only negatively impacted the trained model. We hypothesize that the model used for embedding has a significant impact on the outcomes of this method. However, due to the large computational overhead when scaled, making it infeasible, we opted to rely on the deduplication methods outlined in Appendix L and leave this line of research for future work.

# K Text extraction comparison

Here, we share more detailed quantitative and qualitative comparisons between our chosen extractor, `resiliparse` [20], and the two alternatives previously used by other datasets: WET files, `trafilatura` [17].

## K.1 Profiling

We compute basic summary statistics for each extractor based on a sample of 10 WARC files (corresponding to 900K individual pages), presenting the results in Table 16. Notably, both `resiliparse` and `trafilatura` result in at least 2x shorter documents on average compared to WET files. As shown in the examples in Appendix K.2, WET files indeed contain many additional lines with seemingly little value for pretraining (e.g. navigation bars, boilerplate notices, copyright statements). `trafilatura` and `resiliparse` trim most of these lines out, with the former being more strict about doing so. Between the two, `resiliparse` still keeps in about 10% more text; some of this additional text may provide useful content such as section titles and dates for articles. In terms of runtime, the two are much farther apart, with `resiliparse` being roughly 8x faster.

Table 16: **Text extractor profiling.** Characters and tokens are averaged over the number of resulting output pages (note that this may differ for each extractor due to due to the possibility of extraction failures). Throughput is measured in MBs of input WARCs processed per second for each CPU core.

| Extractor | Avg. Chars | Avg. Tokens | Throughput (MB / sec / core) |
|---|---|---|---|
| `resiliparse` | 3,227 | 1,329 | 4.55 |
| `trafilatura` | 2,901 | 1,179 | 0.56 |
| WET | 6,580 | 2,824 | – |

## K.2 Extraction examples

### K.2.1 Example set 1

---

**[Trafilatura]**

HERE is a sampling of some of the better antiques and flea markets around the United States.
Two or Three Times a Year
BRIMFIELD Route 20, Brimfield, Mass. 01010; 413-245-3436. Second weekend of May and July, and the second weekend after Labor Day.
RENNINGER'S OUTDOOR EXTRAVAGANZA Noble Street, Kutztown, Pa.; 717-385-0104. Thursday, Friday and Saturday of the last weekend of April, June, September.
FARMINGTON ANTIQUES WEEKEND Farmington Polo Grounds, Town Farm Road, Farmington, Conn. 06032; 508-839-9735. Starting Wednesday before shows open; 203-677-7862. June 9-10 and Sept. 1-2.
Monthly
ANN ARBOR ANTIQUES MARKET, P.O. Box 1512, Ann Arbor, Mich. 48106; 313-662-9453. May through October, third Sunday.

---

Continue reading the main storyKANE COUNTY FLEA MARKET, Kane County Fairgrounds, P.O. Box 549, St. Charles, Ill. 60174; 708-377-2252. Year-round, first weekend.
THE METROLINA EXPO, 7100 Statesville Road, Charlotte, N.C. 28213; 704-596-4643. Year-round, first weekend of every month.
SPRINGFIELD ANTIQUE SHOW AND FLEA MARKET, Clark County Fairgrounds, Route 41, Springfield, Ohio, 45501; 513-325-0053. Year-round, third weekend.
Weekly
BAKERSFIELD SWAP-O-RAMA, 4501 Wible Road, Bakersfield, Calif. 93313; 805-831-9342. Saturday and Sunday.
LAMBERTVILLE ANTIQUE MARKET, Route 29, Lambertville, N.J. 08530. Weekend number: 609-397-0456. Weekday: 215-752-4485, between 5 and 7 P.M. Market on Saturday and Sunday.
ATLANTA FLEA MARKET AND ANTIQUE CENTER, 5360 Peachtree Industrial Boulevard, Chamblee, Ga. 30341; 404-458-0456. Friday, Saturday and Sunday.
Continue reading the main story

---

**[Resiliparse]**

This is a digitized version of an article from The Times's print archive, before the start of online publication in 1996. To preserve these articles as they originally appeared, The Times does not alter, edit or update them.

Occasionally the digitization process introduces transcription errors or other problems. Please send reports of such problems to archive_feedback@nytimes.com.

May 10, 1990, Page 00006 The New York Times Archives

HERE is a sampling of some of the better antiques and flea markets around the United States.

Two or Three Times a Year

BRIMFIELD Route 20, Brimfield, Mass. 01010; 413-245-3436. Second weekend of May and July, and the second weekend after Labor Day.

RENNINGER'S OUTDOOR EXTRAVAGANZA Noble Street, Kutztown, Pa.; 717-385-0104. Thursday, Friday and Saturday of the last weekend of April, June, September.

FARMINGTON ANTIQUES WEEKEND Farmington Polo Grounds, Town Farm Road, Farmington, Conn. 06032; 508-839-9735. Starting Wednesday before shows open; 203-677-7862. June 9-10 and Sept. 1-2.

Monthly

ANN ARBOR ANTIQUES MARKET, P.O. Box 1512, Ann Arbor, Mich. 48106; 313-662-9453. May through October, third Sunday.

Continue reading the main story

KANE COUNTY FLEA MARKET, Kane County Fairgrounds, P.O. Box 549, St. Charles, Ill. 60174; 708-377-2252. Year-round, first weekend.

THE METROLINA EXPO, 7100 Statesville Road, Charlotte, N.C. 28213; 704-596-4643. Year-round, first weekend of every month.

SPRINGFIELD ANTIQUE SHOW AND FLEA MARKET, Clark County Fairgrounds, Route 41, Springfield, Ohio, 45501; 513-325-0053. Year-round, third weekend.

Weekly

BAKERSFIELD SWAP-O-RAMA, 4501 Wible Road, Bakersfield, Calif. 93313; 805-831-9342. Saturday and Sunday.

LAMBERTVILLE ANTIQUE MARKET, Route 29, Lambertville, N.J. 08530. Weekend number: 609-397-0456. Weekday: 215-752-4485, between 5 and 7 P.M. Market on Saturday and Sunday.

ATLANTA FLEA MARKET AND ANTIQUE CENTER, 5360 Peachtree Industrial Boulevard, Chamblee, Ga. 30341; 404-458-0456. Friday, Saturday and Sunday.

Continue reading the main story

---

**[WET file]**

A Guide To Markets - The New York Times
NYTimes.com no longer supports Internet Explorer 9 or earlier. Please upgrade your browser. LEARN MORE ›
Sections
Home
Search
Skip to content Skip to navigation View mobile version
The New York Times
Archives|A Guide To Markets
Search
Subscribe Now
Log In
0
Settings
Close search
Site Search Navigation
Search NYTimes.com
Clear this text input
Go
https://nyti.ms/29nVV3Q
Loading...

Archives | 1990

A Guide To Markets

MAY 10, 1990

About the Archive

May 10, 1990, Page 00006 The New York Times Archives

HERE is a sampling of some of the better antiques and flea markets around the United States.

Two or Three Times a Year

BRIMFIELD Route 20, Brimfield, Mass. 01010; 413-245-3436. Second weekend of May and July, and the second weekend after Labor Day.

RENNINGER'S OUTDOOR EXTRAVAGANZA Noble Street, Kutztown, Pa.; 717-385-0104. Thursday, Friday and Saturday of the last weekend of April, June, September.

FARMINGTON ANTIQUES WEEKEND Farmington Polo Grounds, Town Farm Road, Farmington, Conn. 06032; 508-839-9735. Starting Wednesday before shows open; 203-677-7862. June 9-10 and Sept. 1-2.

Monthly

ANN ARBOR ANTIQUES MARKET, P.O. Box 1512, Ann Arbor, Mich. 48106; 313-662-9453. May through October, third Sunday.

Continue reading the main story
Advertisement
Continue reading the main story

KANE COUNTY FLEA MARKET, Kane County Fairgrounds, P.O. Box 549, St. Charles, Ill. 60174; 708-377-2252. Year-round, first weekend.

THE METROLINA EXPO, 7100 Statesville Road, Charlotte, N.C. 28213; 704-596-4643. Year-round, first weekend of every month.

SPRINGFIELD ANTIQUE SHOW AND FLEA MARKET, Clark County Fairgrounds, Route 41, Springfield, Ohio, 45501; 513-325-0053. Year-round, third weekend.

Weekly

BAKERSFIELD SWAP-O-RAMA, 4501 Wible Road, Bakersfield, Calif. 93313; 805-831-9342. Saturday and Sunday.

LAMBERTVILLE ANTIQUE MARKET, Route 29, Lambertville, N.J. 08530. Weekend number: 609-397-0456. Weekday: 215-752-4485, between 5 and 7 P.M. Market on Saturday and Sunday.

ATLANTA FLEA MARKET AND ANTIQUE CENTER, 5360 Peachtree Industrial Boulevard, Chamblee, Ga. 30341; 404-458-0456. Friday, Saturday and Sunday.

A version of this list appears in print on May 10, 1990, on Page C00006 of the National edition with the headline: A Guide To Markets. Order Reprints| Today's Paper|Subscribe

Continue reading the main story
What's Next
Loading...
Go to Home Page »

Site Index The New York Times
Site Index Navigation
News
World
U.S.
Politics
N.Y.
Business
Tech
Science
Health
Sports
Education
Obituaries
Today's Paper
Corrections
Opinion
Today's Opinion
Op-Ed Columnists
Editorials
Op-Ed Contributors
Letters
Sunday Review
Video: Opinion
Arts
Today's Arts
Art & Design
Books
Dance
Movies
Music
N.Y.C. Events Guide
Television
Theater
Video: Arts
Living
Automobiles
Crossword
Food
Education
Fashion & Style
Health
Jobs

```
Magazine
N.Y.C. Events Guide
Real Estate
T Magazine
Travel
Weddings & Celebrations
Listings & More
Reader Center
Classifieds
Tools & Services
N.Y.C. Events Guide
Multimedia
Photography
Video
NYT Store
Times Journeys
Subscribe
Manage My Account
NYTCo
Subscribe
Subscribe
Home Delivery
Digital Subscriptions
Crossword
Email Newsletters
Gift Subscriptions
Group Subscriptions
Education Rate
Mobile Applications
Replica Edition
Site Information Navigation
© 2019 The New York Times Company
Home
Search
Accessibility concerns? Email us at accessibility@nytimes.com. We would love to hear from you.
Contact Us
Work With Us
Advertise
Your Ad Choices
Privacy
Terms of Service
Terms of Sale
Site Information Navigation
Site Map
Help
Site Feedback
Subscriptions
```

## K.2.2 Example set 2

### [Trafilatura]

```
Possible Duplicate:
When should I use an em-dash, an en-dash, and a hyphen?
When do I put a - in a sentence? Is it a more powerful comma? With a bigger pause?
Possible Duplicate:
When should I use an em-dash, an en-dash, and a hyphen?
When do I put a - in a sentence? Is it a more powerful comma? With a bigger pause?
This question has been asked before and already has an answer. If those answers do not fully address your question,
please ask a new question.
The dashes you described are known respectively as the en-dash and the em-dash. To describe the difference between
their origins, Mental Floss writes:
An en dash (-) is bigger than a hyphen but shorter than an em dash (-). Th e names come from an obscure typographical
measurement system, but the dashes have now taken on a life of their own in grammar. The em dash is the spork of
English grammar: It ain't particularly pretty, but you can use it for most anything. Em dashes can replace colons or
sets of parentheses, or represent a sudden change in thought or tone.
So when do you use an en-dash? Again from Mental Floss:
To show numerical ranges, signifying "up to and including"-of dates, ages, pages, etc. (Example: "I read pages 7-22
last night.")
The storied "compound adjective hyphen," an event so rare in the English language that proofreaders shiver with
excitement whenever they come across it. Basically "pro-American" gets a regular hyphen because "American" is only one
word, whereas "pro-Falkland Islands" gets an en dash because "Falkland Islands" is two words. So, too phrases like
"Civil War-era."
What about an em-dash? From here:
Similar to an extended hyphen (-), an em dash is used to show a break in thought or a shift of tone.
If you'd like to read more about the differences between a hyphen (-), en-dash (-), and em-dash (-), see the blog post
here which summarizes the above.
```

### [Resiliparse]

```
1

Possible Duplicate:
When should I use an em-dash, an en-dash, and a hyphen?
```

When do I put a - in a sentence? Is it a more powerful comma? With a bigger pause?

marked as duplicate by waiwai933, MrHen, user2683, Robusto, Thursagen Jul 13 '11 at 0:32

This question has been asked before and already has an answer. If those answers do not fully address your question, please ask a new question.

0

The dashes you described are known respectively as the en-dash and the em-dash. To describe the difference between their origins, Mental Floss writes:

An en dash (-) is bigger than a hyphen but shorter than an em dash (–). Th e names come from an obscure typographical measurement system, but the dashes have now taken on a life of their own in grammar. The em dash is the spork of English grammar: It ain't particularly pretty, but you can use it for most anything. Em dashes can replace colons or sets of parentheses, or represent a sudden change in thought or tone.

So when do you use an en-dash? Again from Mental Floss:

  1. To show numerical ranges, signifying "up to and including"-of dates, ages, pages, etc. (Example: "I read pages 7-22 last night.")

  2. The storied "compound adjective hyphen," an event so rare in the English language that proofreaders shiver with excitement whenever they come across it. Basically "pro-American" gets a regular hyphen because "American" is only one word, whereas "pro-Falkland Islands" gets an en dash because "Falkland Islands" is two words. So, too phrases like "Civil War-era."

What about an em-dash? From here:

Similar to an extended hyphen (-), an em dash is used to show a break in thought or a shift of tone.

If you'd like to read more about the differences between a hyphen (-), en-dash (–), and em-dash (—), see the blog post here which summarizes the above.

Not the answer you're looking for? Browse other questions tagged or ask your own question.

---

**[WET file]**

syntax - What's the difference between - and -- in a phrase? - English Language & Usage Stack Exchange
Stack Exchange Network
Stack Exchange network consists of 175 Q&A communities including Stack Overflow, the largest, most trusted online community for developers to learn, share their knowledge, and build their careers.
Visit Stack Exchange
Log In Sign Up
current community
English Language & Usage
help chat
English Language & Usage Meta
your communities
Sign up or log in to customize your list.
more stack exchange communities
company blog
Tour Start here for a quick overview of the site
Help Center Detailed answers to any questions you might have
Meta Discuss the workings and policies of this site
About Us Learn more about Stack Overflow the company
Business Learn more about hiring developers or posting ads with us
By using our site, you acknowledge that you have read and understand our Cookie Policy, Privacy Policy, and our Terms of Service.
English Language & Usage Stack Exchange is a question and answer site for linguists, etymologists, and serious English language enthusiasts. Join them; it only takes a minute:
Sign up
Here's how it works:
Anybody can ask a question
Anybody can answer
The best answers are voted up and rise to the top
Home
Questions
Tags
Users
Unanswered
What's the difference between - and - in a phrase? [duplicate]
Ask Question
1
Possible Duplicate:
When should I use an em-dash, an en-dash, and a hyphen?
When do I put a - in a sentence? Is it a more powerful comma? With a bigger pause?
syntax dashes symbols
share|improve this question
edited Apr 13 '17 at 12:38
Community
1
asked Jul 13 '11 at 0:10
curiouscurious
123115
marked as duplicate by waiwai933, MrHen, user2683, Robusto, Thursagen Jul 13 '11 at 0:32
This question has been asked before and already has an answer. If those answers do not fully address your question, please ask a new question.
add a comment |

1 Answer 1
active oldest votes
0
The dashes you described are known respectively as the en-dash and the em-dash. To describe the difference between their origins, Mental Floss writes:

An en dash (-) is bigger than a hyphen but shorter than an em dash (-). Th e names come from an obscure typographical measurement system, but the dashes have now taken on a life of their own in grammar. The em dash is the spork of English grammar: It ain't particularly pretty, but you can use it for most anything. Em dashes can replace colons or sets of parentheses, or represent a sudden change in thought or tone.

So when do you use an en-dash? Again from Mental Floss:

To show numerical ranges, signifying ''up to and including''-of dates, ages, pages, etc. (Example: ''I read pages 7-22 last night.'')

The storied ''compound adjective hyphen,'' an event so rare in the English language that proofreaders shiver with excitement whenever they come across it. Basically ''pro-American'' gets a regular hyphen because ''American'' is only one word, whereas ''pro-Falkland Islands'' gets an en dash because ''Falkland Islands'' is two words. So, too phrases like ''Civil War-era.''

What about an em-dash? From here:

Similar to an extended hyphen (-), an em dash is used to show a break in thought or a shift of tone.

If you'd like to read more about the differences between a hyphen (-), en-dash (-), and em-dash (-), see the blog post here which summarizes the above.

share|improve this answer
edited Aug 4 '15 at 17:00
zwol
2,51911424
answered Jul 13 '11 at 0:16
simchonasimchona
30.9k5112139
add a comment |
Not the answer you're looking for? Browse other questions tagged syntax dashes symbols or ask your own question.
asked
7 years, 9 months ago
viewed
2,336 times
active
3 years, 8 months ago

Featured on Meta
Announcing the arrival of Valued Associate #679: Cesar Manara
Planned maintenance scheduled April 17/18, 2019 at 00:00UTC (8:00pm US/Eastern)

Linked
279
When should I use an em-dash, an en-dash, and a hyphen?

Related
279
When should I use an em-dash, an en-dash, and a hyphen?
4
When to use -, - and -?
6
What is the difference between `-` and `--`
5
Does ''cost-benefit ratio'' use a hyphen or an en-dash?
0
What kind of dash character should I use at the end of a famous saying to mark of the author?
-1
dash non-restrictive element in the middle of a sentence
1
em dash followed by a comma
0
what's the difference between a hyphen, a dash and a minus sign?
0
Using comma to delimit the name of a group and its constituents?
1
I tend to overuse the hyphen as a pause, and would appreciate some feedback on this

Hot Network Questions
How could we fake a moon landing now?
Using audio cues to encourage good posture
Chinese Seal on silk painting - what does it mean?
How does the math work when buying airline miles?
What causes the direction of lightning flashes?
Maximum summed subsequences with non-adjacent items
Most bit efficient text communication method?
Significance of Cersei's obsession with elephants?
Is it a good idea to use CNN to classify 1D signal?
What font is "z" in "z-score"?
How can I use the Python library networkx from Mathematica?
Fundamental Solution of the Pell Equation
Dating a Former Employee
Do I really need recursive chmod to restrict access to a folder?
Generate an RGB colour grid
How to Make a Beautiful Stacked 3D Plot
How to find all the available tools in mac terminal?
Has negative voting ever been officially implemented in elections, or seriously proposed, or even studied?
What is the meaning of the simile ''quick as silk''?
Circuit to "zoom in" on mV fluctuations of a DC signal?
What are the out-of-universe reasons for the references to Toby Maguire-era Spider-Man in Into the Spider-Verse?
What does this Jacques Hadamard quote mean?
Can a new player join a group only when a new campaign starts?
Why wasn't DOSKEY integrated with COMMAND.COM?
more hot questions
English Language & Usage
Tour
Help
Chat

Contact
Feedback
Mobile
Company
Stack Overflow
Stack Overflow Business
Developer Jobs
About
Press
Legal
Privacy Policy
Stack Exchange
Network
Technology
Life / Arts
Culture / Recreation
Science
Other
Stack Overflow
Server Fault
Super User
Web Applications
Ask Ubuntu
Webmasters
Game Development
TeX - LaTeX
Software Engineering
Unix & Linux
Ask Different (Apple)
WordPress Development
Geographic Information Systems
Electrical Engineering
Android Enthusiasts
Information Security
Database Administrators
Drupal Answers
SharePoint
User Experience
Mathematica
Salesforce
ExpressionEngine® Answers
Stack Overflow em Português
Blender
Network Engineering
Cryptography
Code Review
Magento
Software Recommendations
Signal Processing
Emacs
Raspberry Pi
Stack Overflow на русском
Programming Puzzles & Code Golf
Stack Overflow en español
Ethereum
Data Science
Arduino
Bitcoin
more (31)
Photography
Science Fiction & Fantasy
Graphic Design
Movies & TV
Music: Practice & Theory
Worldbuilding
Seasoned Advice (cooking)
Home Improvement
Personal Finance & Money
Academia
Law
more (15)
English Language & Usage
Skeptics
Mi Yodeya (Judaism)
Travel
Christianity
English Language Learners
Japanese Language
Arqade (gaming)
Bicycles
Role-playing Games
Anime & Manga
Puzzling
Motor Vehicle Maintenance & Repair
more (33)
MathOverflow
Mathematics
Cross Validated (stats)
Theoretical Computer Science
Physics
Chemistry
Biology

```
Computer Science
Philosophy
more (10)
Meta Stack Exchange
Stack Apps
API
Data
Blog
Facebook
Twitter
LinkedIn
site design / logo © 2019 Stack Exchange Inc; user contributions licensed under cc by-sa 3.0 with attribution required.
rev 2019.4.18.33353
English Language & Usage Stack Exchange works best with JavaScript enabled
```

## L  Deduplication

We perform extensive ablations and experimentation on various deduplication pipelines. This section is organized by first describing the deduplication methods considered and then outlining the ablations that lead us to the choice of deduplication pipeline used in generating DCLM-BASELINE (and other DCLM scales).

### L.1  Deduplication methods

Prior work such as Lee et al. [94], Penedo et al. [127] use a two-stage deduplication pipeline where near duplicates are first removed at a inter-document level by identifying and removing near-duplicates using the MinHash algorithm, and then at an intra-document level where any substring of a predetermined length that occurs more than once in the entire corpus is removed. Intuitively, this strategy makes sense as the notion of a "duplicate" is poorly defined and can include documents such as: (i) exact copies of entire documents (targeted at the document-level); (ii) documents where the majority of the text is a duplicate, but there are unique differences in just the header or footer (targeted at the document-level); or (iii) documents where there are significant sections of unique text, but also massively repeated boilerplate text (targeted at the intra-document level). Performing multiple resolutions of deduplication can target all such cases, and further, a deduplication pipeline that can target near-duplicates, often referred to as "fuzzy deduplication" can identify documents that humans would intuitively refer to as duplicates.

While we ultimately rely on a Bloom filter based method of deduplication for our datasets, we describe the other pipelines considered:

**MinHash.**   MinHash is a locality-sensitive hashing technique used to group sets into collections based on their Jaccard similarity [29]. In the context of deduplicating text datasets, MinHash was first employed in Lee et al. [94] and then used in numerous other projects [46, 127]. We point readers to the main text of Lee et al. [94] and Appendix G.3.1 of Penedo et al. [127] for more details. The primary hyperparameters of note are the n-gram-size, and the number of permutations used. Following Lee et al. [94], Penedo et al. [127], we use an n-gram-size of 5 tokens and target a Jaccard similarity of 0.8. Departing from prior work, however, we modify the number of MinHash permutations used. Both Lee et al. [94] and Penedo et al. [127] use a total of 9,000 permutations, split into 450 buckets of 20 hashes each. We found this to be overly expensive and notice that similar Jaccard similarity plots can be attained with a much smaller number of permutations. For all of our ablations, we instead use a total of 1,395 permutations, split into 93 buckets of size 15. These hyperparameters were chose programmatically to mimic the Jaccard similarity plots as closely as possible, in an $\ell_2$ sense, with a fixed hash budget. See Figure 7 for more details.

**Suffix arrays.**   Suffix arrays, first introduced in Manber & Myers [111], enable efficient identification and removal of substrings of a large corpus of text. This is done by first concatenating all text in the corpus together and then sorting each suffix. By scanning this sorted list, substrings with a common prefix can by identified by scanning the prefces of neighboring elements in the sorted list. This latter step can be done in an embarrassingly parallel fashion, but the implementation we employed, borrowed from the codebase provided in Lee et al. [94] is not done in a multi-node fashion and requires loading the entire corpus into RAM. We directly employ the hyperparameters used in Lee et al. [94] and remove all repeated substrings that are at least 50 tokens long.

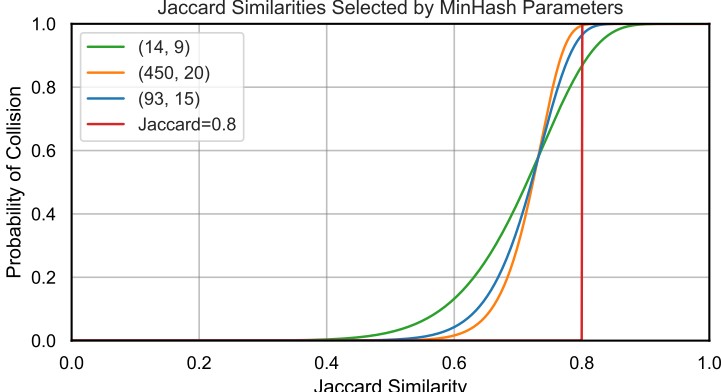

Figure 7: Probability of two documents with Jaccard similarity (x-axis) being marked as duplicates (y-axis) with varying (number of buckets, bucket size) parameters. (450, 20) corresponds to Lee et al. [94], Penedo et al. [127], our experiments used (93, 15), chosen to be a cheaper alternative emulating the same performance as (450, 20). The parameters (14,9) were used by Penedo et al. [128].

**Bloom filters.** Bloom filters are a data structure that enable space-efficient set membership queries [26]. Explicitly, in sublinear space, a Bloom filter maintains a sketch of a set, that supports an `insert` operation, and a probabilistic `membership_query` operation, where the latter will never return any false negatives (i.e., return False for an element in the set), but will occasionally return a false positive (i.e., return True for an element not in the set). These were first used in the context of exact-duplicate removal in Soldaini et al. [156], but have since been extended to perform near-duplicate document and paragraph removal in a tool known as BFF (Big Friendly Filter) [72], and we further modify BFF to perform deduplication at the document and paragraph level simultaneously. We found that this technique is vastly more efficient than a MinhHash and SuffixArray pipeline. However there is one important caveat in that MinHash performs document-level deduplication at a document vs. document level, whereas BFF performs document-level deduplication at a document vs. corpus level.

**Paragraph + document BFF.** Here we outline our modified Bloom filter based deduplication algorithm. Upon initialization, we require an estimate of the number of tokens in our entire corpus as well as a desired false-positive rate, to initialize a Bloom filter with a fixed size and number of hashers. The optimal number of hashers, $k$, is given by the formula

$$k = -\frac{\ln \epsilon}{\ln 2},$$

where $\epsilon$ is the desired false positive rate. The optimal size $m$ for $k$ hashers and $n$ tokens can then be computed by solving for $m$ in the following formula:

$$\epsilon = \left(1 - e^{\frac{-kn}{m}}\right)^k.$$

While this does not admit an easy analytical solution, it is trivial to solve for $m$ by using a binary search algorithm.

Once we have established a Bloom filter, we proceed through each document in our corpus and perform the following steps. First we tokenize the document using the UniSeg tokenizer [48], and then further break the document into paragraphs by splitting on the newline character \n. For each document, we maintain counters `total_ngrams`, and `contained_ngrams`. Each paragraph is then handled in turn, according to hyperparameters denoting `min_ngram_size`, `max_ngram_size`, and `threshold`:

- If the paragraph is fewer than `min_ngram_size` tokens long, it is left as is.
- If the paragraph is in between `min_ngram_size` and `max_ngram_size` (inclusive) then `total_ngrams` is incremented and this n-gram's membership is checked in the Bloom filter. If it is present in the Bloom filter, it is removed from the paragraph and `contained_ngrams` is incremented. Otherwise, it is added to the Bloom filter.

- If the paragraph is at longer than `max_ngram_size` tokens, then each n-gram of size `max_ngram_size` increments the total counter and is checked against the Bloom filter. If present, the `contained_ngrams` counter is incremented. If greater than `threshold` fraction of the n-grams in this paragraph are contained in the Bloom filter, then the entire paragraph is removed from the document. Otherwise, every non-contained n-gram is added to the Bloom filter.

Once all paragraphs have been processed, if the ratio between the counters `contained_ngrams` and `total_ngrams` is greater than `threshold`, then the entire document is removed from the corpus.

To finalize our discussion on the Bloom filter based deduplication, we offer brief explanations on the hyperparameter choices made.

- **False Positive Rate:** The two parameters that dictate the memory footprint required by BFF are the number of tokens and the false positive rate. However we only can control the false positive rate, and we notice that the Bloom filter size scales linearly with the negative log of the false positive rate. In particular, for a corpus of 1T tokens, occupying roughly 2TB of disk space, ensuring no false positives, i.e. setting the false positive rate to $1/1T$, would require 6.5TB of RAM. Here we argue analytically that a false positive rate of even as low as 0.01 suffices, which we support with experimentation in the next section.

  In choosing a false positive rate for the n-gram-based Bloom filter, it's important to recognize that removal of a paragraph or document is dictated by having greater than a `threshold` fraction of the n-grams contained in the set. As an example, suppose we are given a paragraph of $N$ n-grams, where $S$ of them are already contained in the Bloom filter and we set `threshold` to $T$. Because Bloom filters do not allow false negatives, every one of the $S$ n-grams are marked (correctly) as contained, and $N - S$ of them could potentially be marked as a false positive. Indeed, of the $N - S$ of these n-grams, at least $TN - S$ of them would need to be marked as a false positive, each of which occurs independently with probability $\epsilon$. This is equivalent to $N - S$ Bernoulli random variables with parameter $\epsilon$, and can be bounded by a crude Hoeffding bound. In this particular case, the probability that a document or paragraph is falsely marked as a duplicate is bounded by:

$$\exp\left(\frac{-2 \cdot \left(TN - S - \epsilon \cdot (N - S)\right)^2}{N - S}\right)$$

  To put things concretely, in a document with 100 n-grams and a threshold of 0.8 and a false positive rate of 0.01, if 60 of the n-grams have been seen before, the probability of the document being marked as a duplicate is less than $10^{-8}$. Unless otherwise specified, we always use a false positive rate of 0.01.

- `min_ngram_size:` In choosing a size for minimum n-grams, we recognize that many documents contain paragraphs that are itemized lists and are quite short; for example, recipes often include bullet-pointed ingredients lists, and MMLU multiple choice questions may often be quite short. While we originally noticed improved CORE scores by setting a minimum size to 5 tokens, we noticed that this caused a worse performance on MMLU. After manual inspection, we settled on a min and max n-gram size of 13 tokens.

- `threshold:` Ablations did not show a noticable difference in deduplication performance.

## L.2 Deduplication experiments

### L.2.1 Deduplication ablations: pipeline at `1B-1x` scale

We first perform ablations regarding the full pipeline choice for deduplication at the `1B-1x` scale. We start with a pool of 76B tokens subsampled from Common Crawl with the preprocessing steps from Penedo et al. [127] applied. Then we apply a combination of deduplication steps, and subsample the pool further to the 28B tokens required for the `1B-1x` scale. Finally we train and evaluate the CORE score and the percentage of tokens that were removed by deduplication. The main questions we seek to answer from this round of ablations are:

- For multi-step deduplication pipelines, how much of a contribution does each step provide?
- Which deduplication pipeline is worth scaling up to larger pool sizes?

Results are contained in Table 17. The main conclusions we can arrive at from this table are as follows: i) Suffix Array deduplication seems to help more than MinHash deduplication, thereby giving some signal to the source of the gains procured by a MinHash+SuffixArray pipeline; ii) BFF provides comparable performance to a full Exact+MinHash+SuffixArray pipeline, giving strong evidence that the multiresolution BFF could be an easily scalable alternative to the relatively more expensive MinHash+SuffixArray pipeline of prior works. Interestingly, it appears that a SuffixArray pipeline seems to outperform MinHash alone, though this falls within the range of variance for the CORE score due to the nondeterminism in subsampling the dataset and training a model.

Table 17: **Deduplication ablations** (1B-1x scale). Starting from a pool of 76B tokens acquired from Common Crawl with the RefinedWeb Penedo et al. [127] pipeline applied, we evaluate the removal rate and CORE score on different combinations of deduplication methods. Our Bloom filter method performs as well as a combination of exact deduplication, MinHash and Suffix Array based techniques.

| Exact Dedup | MinHash | Suffix Array | Bloom Filter | Tokens | Removal Rate | CORE | Δ from Baseline |
|:---:|:---:|:---:|:---:|:---:|:---:|:---:|:---:|
| ✗ | ✗ | ✗ | ✗ | 76B | 00% | 24.7 | +0.0 |
| ✓ | ✗ | ✗ | ✗ | 66B | 13% | 26.0 | +1.3 |
| ✗ | ✓ | ✗ | ✗ | 62B | 18% | 25.6 | +0.9 |
| ✗ | ✗ | ✓ | ✗ | 51B | 33% | 26.6 | +1.9 |
| ✗ | ✗ | ✗ | ✓ | **56B** | **26%** | **26.8** | **+2.1** |
| ✓ | ✓ | ✗ | ✗ | 58B | 24% | 25.0 | +0.3 |
| ✓ | ✗ | ✓ | ✗ | 49B | 36% | 26.2 | +1.5 |
| ✗ | ✓ | ✓ | ✗ | 48B | 37% | 26.3 | +1.6 |
| ✓ | ✓ | ✓ | ✗ | **45B** | **41%** | **26.8** | **+2.1** |

### L.2.2 Deduplication ablations: pipeline at `7B-1x` / `7B-2x` scale

To further check the effects of BFF versus the more classical MinHash+SuffixArray we ran several experiments at the `7B-1x` scale. Here we also introduce another hyperparameter, which we refer to as **shards**. By "sharding," we mean we break a dataset into chunks of roughly equal size and run the deduplication pipeline on each one of them independently. This is primarily done for engineering purposes, in that sharding is an easy way to further parallelize deduplication and convert single-node algorithms to multi-node algorithms. However, there are the side benefits of sharding for deduplication in that more shards yields a larger token pool: there are fewer documents to compare against and many documents which are repeated only a small number of times can survive such a process. Additionally there is some recent evidence that sharding seems to improve evaluation performance [128]. We also note that RefinedWeb [127] performs their deduplications on a 100-way sharding of the Common Crawl pool.

For this round of ablations, we start with a pool sourced from one tenth of Common Crawl and run the preprocessing steps from Penedo et al. [127] and apply various deduplication pipelines. Then we subsample down to 138B tokens and train and evaluate models at the `7B-1x` scale. The main questions we seek to answer from this round of ablations are:

- Is BFF still competitive with a MinHash+SuffixArray pipeline at larger scales?
- Which BFF hyperparameters yield the highest CORE and MMLU performance at this scale?

Results are contained in Table 18. The first point to note is that BFF with a `min_ngram_size` at 13 and 20 yields CORE scores and MMLU scores that are comparable to the scores attained by a MinHash+SuffixArray deduplicated pool at the same scale. The second point to note regards the BFF `min_ngram_size` and sharding: interestingly a lower `min_ngram_size` yields higher CORE scores, but lower MMLU scores. We also see that fewer shards decreases the token yield, but has variable effect on the CORE score. We examine the hyperparameters for BFF more fully in the next subsection.

Encouraged by these results, next we examine the top candidates for a scalable deduplication pipeline at the `7B-2x` scale. Again we start with a pool obtained from one tenth of Common Crawl and generate several deduplicated pools. The questions of interest are the same as above and we summarize the results in Table 19. The key takeaways from this round of ablations is that at the

Table 18: **Deduplication Ablations** (7B-1x scale). Starting with a pool from Common Crawl and the `RW-Filter` pipeline processing applied, we compared several BFF hyperparameters against the MinHash and Suffix Array pipeline of [94, 127]. Our best BFF run and the prior works are bolded.

| Method | min-ngram | max-ngram | Shards | MMLU | CORE | Token Yield |
|---|---|---|---|---|---|---|
| Bloom Filter | 5 | 13 | 32 | 25.0 | 40.6 | 4T |
| Bloom Filter | 5 | 13 | 1 | 27.1 | 41.7 | 1.3T |
| **Bloom Filter** | **13** | **13** | **32** | **28.7** | **40.5** | **3.8T** |
| Bloom Filter | 20 | 20 | 32 | 27.7 | 40.0 | 4T |
| Bloom Filter | 20 | 20 | 10 | 28.4 | 39.7 | 3T |
| **MinHash+SA** | **N/A** | **N/A** | **16** | **29.1** | **40.8** | **3.2T** |

7B-2x scale, BFF with a `min_ngram_size` of 13 and 10 shards attains nearly identical performance to a MinHash+SuffixArray pipeline, whereas BFF with a `min_ngram_size` of 20 and 32 shards starts to lag behind, and that a `min_ngram_size` of 5 yields competitive CORE scores, but falters in MMLU evaluations. While these experiments also vary the sharding choice, we view sharding primarily as a choice made to trade-off scalability with token yield. Larger shards are more expensive and less parallelizable and can decrease the token yield. For this round of ablations, the primary interest is to gain signal about how BFF compares to MinHash and Suffix Arrays at scale, and which are the correct hyperparameters for BFF. On this latter point, we chose to move forward with a `min_ngram_size` of 13 for generating DCLM-BASELINE.

Table 19: **Deduplication Ablations** (7B-2x scale). From the same pools as in Table 18, we trained and evaluated models at the 7B-2x scale. Notice that a `min_ngram_size` of 5 yields competitive CORE results but drastically reduces MMLU scores.

| Method | min_ngram | max_ngram | Shards | MMLU | CORE | Token Yield |
|---|---|---|---|---|---|---|
| BFF | 20 | 20 | 10 | 43.6 | 45.8 | 3T |
| **BFF** | **13** | **13** | **10** | **44.3** | **45.3** | **3T** |
| BFF | 5 | 13 | 32 | 32.0 | 44.5 | 4T |
| **MinHash+SA** | **N/A** | **N/A** | **16** | **44.4** | **45.5** | **3.2T** |

### L.2.3   BFF hyperparameter ablations

While the above ablations largely focused on the CORE score and MMLU as performance metrics, these are expensive and not suited for large swaths of ablations. Here we instead explore statistics of datasets deduplicated by BFF as we toggle the `ngram_size` hyperparameters, false positive rate, and input dataset size. We run separate experiments for each hyperparameter and finish each paragraph with the choice of hyperparameter we use for all larger scale runs.

**False positive rate**   Here we start with the 75B token data pool as in Appendix L.2.1 and focus on a paragraph-only level BFF. In other words, we run BFF as described above, except omit the full-document removal step. We use the default hyperparameters for n-gram sizes as in Groeneveld [72], of 5 and 13 for `min_ngram_size` and `max_ngram_size` and a threshold of 0.8. We specifically look at the effect of changing the false positive rate and compute the removal rate (in bytes) of the output. From Table 20, we can see that a false positive rate of 0.1 suffices for a reasonably small pool such as this one. For larger pools, to be safe, we always set the false positive rate to 0.01.

Table 20: **False positive rate ablations.** Starting with a pool of 75B tokens from the RW-Filter pipeline, we ran BFF with default hyperparameters, varying the false-positive rate to indicate that this does not have a large bearing on output pool size.

| False Positive Rate | Removal Rate (Bytes) |
|---|---|
| 0.1 | 20.47% |
| 0.01 | 20.47% |
| 0.001 | 20.47% |

Table 21: **BFF hyperparameter ablations.** Starting with a pool of 341B tokens taken from Common Crawl with the `RW-Filter` pipeline applied, we run our Bloom filter deduplication with various hyperparameters noting how the document length and pool size change after deduplication. The input pool statistics are noted in the first row.

| Min | Max | Threshold | Avg Tokens/Doc | Median Tokens/Doc | Total Documents | Total Tokens |
|-----|-----|-----------|----------------|-------------------|-----------------|--------------|
| N/A | N/A | N/A | 883 | 451 | 386M | 341.0B |
| 1 | 13 | 0.8 | 778 | 403 | 246M | 191B |
| 5 | 13 | 0.8 | 802 | 426 | 250M | 201B |
| 13 | 13 | 0.8 | 836 | 456 | 246M | 205B |
| 13 | 25 | 0.8 | 839 | 458 | 248M | 208B |
| 13 | 50 | 0.8 | 833 | 453 | 253M | 211B |
| 13 | 13 | 0.75 | 842 | 460 | 241M | 203B |
| 13 | 13 | 0.8 | 836 | 456 | 246M | 205B |
| 13 | 13 | 0.9 | 822 | 446 | 256M | 211B |
| 13 | 13 | 0.99 | 797 | 427 | 275M | 218B |

**Min n-gram size.** From Table 18 and Table 19, we saw that altering the `ngram_size` hyperparameters can affect both token yield and evaluation metrics. In particular, we seek to examine how surviving documents are altered by deduplication. As a proxy for this, we focus on the document lengths and removal rates. Results for this paragraph and the two following paragraphs are collated in Table 21. One key observation is that as the `min_ngram_size` parameter is reduced, the mean and median document lengths become shorter. This indicates that too-low a `min_ngram_size` parameter can dramatically affect the language statistics of the dataset and should be avoided. This tracks with intuitive sense where many documents include linebreak separated lists where each list element is short and possibly repeated: e.g., many webpages include recipes that might call for "1 stick of butter", which would get removed with a `min_ngram_size` of 5 but would injuriously damage the source document.

**Max n-gram size.** Next we examine increasing the `max_ngram_size` hyperparameter. Starting with the chosen `min_ngram_size` parameter of 13, decided in the previous paragraph, we consider `max_ngram_size` parameters of 13, 25, and 50. Contrary to the `min_ngram_size`, we do not see a dramatic alteration of language statistics as this parameter becomes increased. For simplicity, we choose to use a `max_ngram_size` of 13 for large-scale pools.

**Threshold.** The `threshold` hyperparameter dictates how close a document must be to previously seen n-grams before it is considered a duplicate. We ablate this choice from 0.75 to 0.99, examining how this affects document length statistics and removal rates. Interestingly, as the threshold increases, documents get shorter, mirroring the statistics seen for reducing the `min_ngram_size`. As expected, higher thresholds yield lower removal rates. Following the Jaccard similarity choice used in MinHash deduplication and noting that 0.8 yields median tokens/doc closest to the baseline, we use a threshold of 0.8 going forward.

**Shards.** Finally we simulate how shards affect the statistics of the deduplicated datasets. As above, the key statistics we focus on here are the removal rate and the average and medium document lengths. This is mostly to get a sense for how these features change as the dataset scales, with the prevailing thought that dramatically altering document statistics might adversely effect downstream evaluations. For larger pools, we can always shard them as heavily as desired, so we treat sharding as a hyperparameter that controls removal rate and document statistics. Results are collated in Table 22 and Figure 8. The key takeaways here are that removal rates increase monotonically with dataset size as expected, but do so in a concave fashion. This provides some signal for how heavily to shard an input pool if a desired token yield is specified. The next point of interest is to consider the document lengths as the dataset scales. These decrease monotonically as the pool increases in size.

In building DCLM-BASELINE, at the point of deduplication, the dataset is approximately 70TB in size. Since Table 19 shows that BFF had the best performance at a 10-way shard with a roughly 7TB input size, we adhere to a 100-way sharding for DCLM-BASELINE, where each shard is roughly 700GB in size.

Table 22: **Deduplication shard size.** We run a single-shard BFF with the `ngram_size` set to 13, false positive rate 0.01, threshold of 0.80 on pools of varying size. As the pool size scales the deduplication rate increases, documents get shorter and the removal rate increases.

| Input Tokens | Input Documents | Avg Tokens/Doc | Median Tokens/Doc | Token Removal Rate |
|---|---|---|---|---|
| 114B | 129M | 466 | 866 | 29% |
| 227B | 257M | 460 | 848 | 35% |
| 341B | 386M | 456 | 836 | 40% |
| 455B | 516M | 453 | 826 | 43% |
| 569B | 643M | 450 | 918 | 46% |

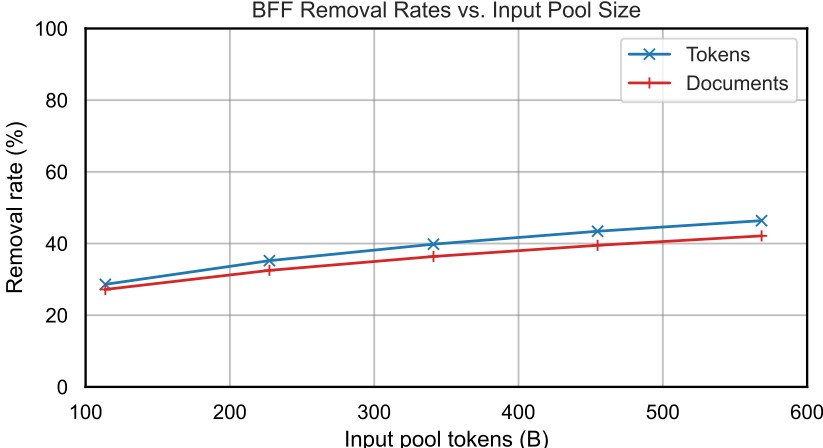

Figure 8: **Deduplication shard size.** We run a single-shard BFF with the `ngram_size` set to 13, false positive rate 0.01, threshold of 0.80 on pools of varying size. Larger pools have a larger removal rate, but this scales in a concave fashion. The removal rates for tokens and documents begin to diverge at larger scales.

### L.2.4 Global MinHash on Open Datasets

Finally, to get a sense for the duplicates remaining in a dataset after a full processing pipeline has been applied, we run a global (i.e., one shard) MinHash on several open datasets. These results are collated in table table 23. We evaluate our DCLM-BASELINE, the official RefinedWeb dataset from HuggingFace, our emulation of the RefinedWeb pipeline, and Dolma V1. MinHash is performed using 14 buckets and a bucket size of 9, corresponding to the green curve in fig. 7.

We note several observations here. First, we note that pools deduplicated with a Bloom Filter still have large numbers of "fuzzy duplicates" in the MinHash/Jaccard Similarity sense. This indicates that what the Bloom Filter considers a duplicate and what MinHash considers duplicates are not identical concepts. Second, we see that while MinHash is a roughly idempotent procedure, deduplication over shards fails to remove a large portion of the duplicates. Third, we see that our 100-shard Bloom filter deduplication applied to DCLM-BASELINE still leaves many duplicates in the dataset, yet does not seem to adversely effect downstream performance. This calls into question the general prevailing thought that the presence of any duplicates hinders downstream performance: we instead conjecture that either i) only large amounts of duplicates are detrimental to downstream performance, or ii) aggressive single-sharded deduplication eliminates many high quality documents. We leave such experimentation for future work.

## M   Mixing sources

In Section 4.5, we showed that mixing our dataset with commonly used sources did not improve its general performance, and hypothesized that this is due to the more stringent filtering performed by DCLM-BASELINE on Common Crawl. Based on this hypothesis, one could argue that improved

Table 23: **Global MinHash on Open Datasets** We perform a global MinHash on several open datasets and evaluate the number of duplicates that would be removed. We denote the deduplication applied to generate each pool and the number of shards used (* implies inferred sharding). DolmaV1 contained approximately 600M documents containing only the empty string, so we report numbers with and without the empty strings in the dataset.

| Dataset | Num Documents | Deduplication Applied | Shards | MinHash Removal Rate |
|---|---|---|---|---|
| DCLM-BASELINE | 3.2B | (Fuzzy) Bloom Filter | 100 | 85% |
| RefinedWeb (official) | 968M | MinHash+SA | 1* | 0% |
| RefinedWeb (ours) | 2.0B | MinHash+SA | 16 | 45% |
| Dolma V1 (w/ empty) | 5.2B | (Exact) Bloom Filter | 1 | 43% |
| Dolma V1 (w/o empty) | 4.6B | (Exact) Bloom Filter | 1 | 36% |

filtering of the other sources before mixing them with DCLM-BASELINE could also lead to improvements in performance. As one such attempt, we perform an experiment where we apply the same `fastText` classifier for filtering the other sources as we do for our DCLM-BASELINE.

We take several sources from `RedPajama` [166], and individually filter them with the `fastText` classifier applied in our DCLM-BASELINE, while keeping only the highest scored ones. We then add the resulting data to our pretraining dataset, and train models at the `1B-1x` scale. The results of this can be seen in Table 24. We see that despite the more uniform handling of mixing across various sources, the additional sources still decrease performance.

We leave further analysis on potential filtering and mixing of non-Common Crawl sources for future work, hoping the mixing track can be used by participants to explore such directions.

Table 24: **Mixing with filtered data.** We evaluate our models on mixtures of data, where we combine our DCLM-BASELINE with filtered data from other sources of `RedPajama` [166]. We find that the case where we use only DCLM-BASELINE performs the best in our experiments. Evaluation is done at the `1B-1x` scale.

| Dataset Mixture | CORE | MMLU | EXTENDED |
|---|---|---|---|
| DCLM-BASELINE only | **31.7** | **26.5** | **16.6** |
| DCLM-BASELINE + Filtered Wiki | 31.0 | 24.9 | 16.3 |
| DCLM-BASELINE + Filtered Books | 30.6 | 25.8 | 15.5 |
| DCLM-BASELINE + Filtered Arxiv | 31.5 | 26.0 | 15.6 |
| DCLM-BASELINE + Filtered Github | 30.0 | 24.4 | 15.2 |

## N  Human judgment

Prior work suggests that using human annotators may introduce undesired bias or noise into the data due to under-training of the annotators for the task, lack of skill or motivation, or unintended leakage of subjective bias [42, 66, 68]. However, human annotators are still widely considered the gold standard for annotating data with a clear task at hand. A natural hypothesis is that if human annotators could manually filter the large pool of raw data, we would end up with a particularly high-quality dataset. To test this, we ask 16 English-speaking AI graduate students and professors to annotate approximately 500 randomly selected documents from a pool of data without a quality filter. We obtain three annotations per document and use the majority vote in each as the gold label. The average inter-annotator agreement is 71%. We further extract the subset of 281 samples where all three annotators are in agreement, naming the full data MAJORITY and the subset AGREEMENT.

We then evaluate various quality filters from Section 4.4 on this data to search for correlation between dataset quality (as measured by CORE accuracy) and filter agreement with human labels. Figure 9 (a) depicts the CORE scores of models trained on datasets filtered with the respective quality filter

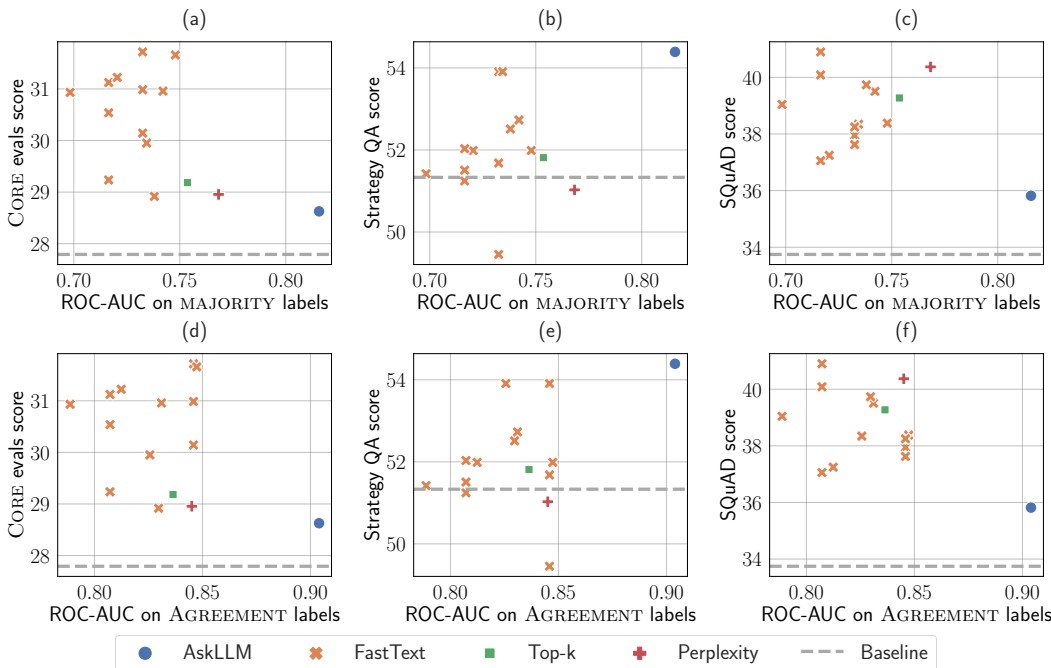

Figure 9: Accuracy measurements against ROC-AUC of different quality filters on subsets of our human annotated samples. Top: MAJORITY, bottom: AGREEMENT. Left: CORE score, middle: StrategyQA, and right: SQuAD. All models share the same scale (1B-1x) and training hyperparameters and are based on the same pre-filtered pool, using similar filtering-ratios for different classifiers (keeping top $\sim 15\%$ of the pool). The horizontal line marks the baseline score of a model trained on random subset of the unfiltered pool. While it may seem there is some positive correlation for StrategyQA, the opposite is true for SQuAD and in both cases the $R^2 < 0.3$. Similar to what seen in CORE score, for almost all other tasks, there is no apparent relationship.

against ROC-AUC[4] of our quality-filters on the MAJORITY data. Notably, both the best and worst fastText-based filters score about the same on the MAJORITY data ($\sim 73\%$ ROC-AUC), while the AskLLM filter that is highly correlated with human annotations ($\sim 82\%$ ROC-AUC) performs much worse as a quality filter ($\sim 28.5\%$ CORE compared to $> 31\%$ in several fastText classifiers).

We continue this study by inspecting correlations to specific downstream tasks and comparing them to the ROC-AUC on the AGREEMENT data, where all three annotators agreed on the label. Figure 9 depicts the scores on a few tasks against the ROC-AUC on the annotated data of the representative set of quality filters. While some positive correlation may be observed for StrategyQA [67], the opposite is true for other QA datasets such as SQuAD [139], and in both cases the $R^2 < 0.3$. In most other downstream tasks, the results are similar to Figure 9 (a), where no correlation can be observed. This suggests that human intuition may not reliably identify the most useful documents for language model training purposes. We hypothesize that human curators may create datasets that lack sufficient diversity and leave further investigation of these hypotheses to future research.

**Collecting the data.** We sampled 499 random documents from our pool, after going through the rule-based quality filters and deduplication. Figure 10 shows a histogram of the length of the documents in words. We asked 16 English speak AI graduate and professors to annotate each example as a good candidate to be included in an LM pretraining corpus (see instructions and some examples

---

[4]ROC-AUC measures a classifier's ability to distinguish between classes by summarizing the trade-off between true positive and false positive rates across all thresholds, making it a robust and common metric for model performance. See https://en.wikipedia.org/wiki/Receiver_operating_characteristic for further details.

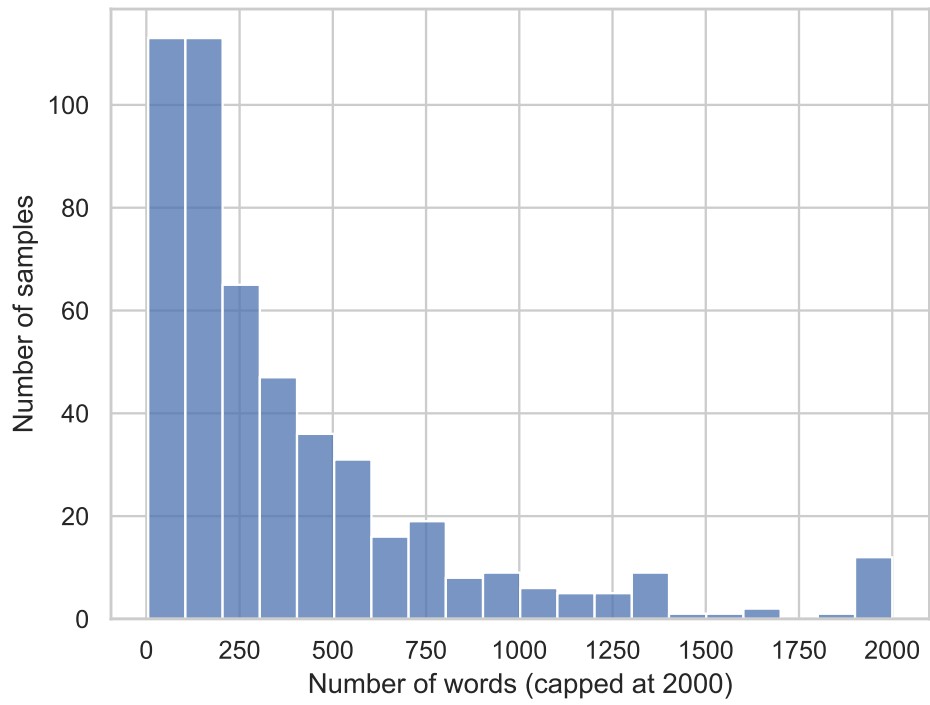

Figure 10: Histogram of length in words for samples in our human-annotated data (capped at 2,000).

given to annotators below).[5] Out of the 499 samples, in 281 samples there was full agreement between all three annotators. We release both datasets in our codebase.[6]

## N.1 Instructions given to annotators

```
Your task is to assess the quality of the documents (0-bad, 1-good), sampled randomly from
↪  RefinedWebV2, in terms of their usefulness for LLM training.

How do you judge the usefulness of a document?

While this is a subjective task, there are a few things to keep in mind:
    1. We encourage you to review the details of the evaluation tasks that the LLMs are expected to
    ↪  excel in [link to spreadsheet with the tasks was provided]. Broadly, these tasks include
    ↪  reading comprehension, language understanding, world knowledge, and commonsense reasoning.
    2. Check whether a particular document will be useful for answering any open-ended imperative
    ↪  tasks or questions. For example: writing a letter to a friend, writing a poem, replying to an
    ↪  email, cooking a dish, etc.
    3. Check whether the data contains harmful content that you would not want an LLM to generate.
    ↪  This can include personal information, vulgar language, etc.
    4. See the examples below to build some intuition.
```

---

[5]All annotation efforts were done voluntarily and were not paid for.
[6]https://github.com/mlfoundations/dclm/tree/main/data

## N.2 Examples

### Doc1 (Bad)

```
Welcome to the Southern California Connection
Prayer Line Ministry
Sponsored by the Central Filipino Church - Women's Ministry Department
Gwen Shorter - Women's Ministry Director
Joanne Williams - Prayer Leader
Dial: (712) 432-0075
Enter Participant Code: 624255
To hear the most recent call, please dial (712) 432-1085
Enter Participant Code: 624255
Sunday through Friday - at 7 am and 7 pm
Psalm 91 (King James Version)Prayer Line Moderators
|Sunday, Monday and Friday
|Joanne Williams, Marsha Harold
|Tuesday, Wed., and Thursday
|Gwen Shorter, Gloria Duckett
Note: Moderators schedule may vary from time to time.
Webmasters: Michael Wong, Will Fults
Hard Copies - For CD's of programs ($3.00 each) Please call Gloria @ 541-476-0038
Click on the links below for some detailed information:
Prayer Line Speakers - Biographys and Photographs
Prayer Line Recordings by Date
View Prayer Line Testimonials - Submit Prayer Line Testimonial
Download, Print and Share:
|free web hit counter
As of December 9, 2010.
```

### Doc2 (Bad)

```
Host: Zander
Program Category: Music
Frequency: Weekly
Length: 2 Hours
Terms: Barter
Delivery Method: Internet
|''Zander's knowledge of music and his straight-forward approach has struck a huge interest among our
↪  listeners. The Rockin' 80's is EXACTLY what we've been looking for!"" - Terry West, WQLA
The Rockin' 80's is the only 80's show with a mix of the best rock from the decade of excess plus ''oh
↪  wow'' tracks that add spice to the weekly line up. The two hour version of the show features
↪  rarities from the 80's ''Lost and Found'', an 80's ''Two-Fer,'' spotlighting two contrasting songs
↪  from one band played back to back and much more. Featured core musical artists include Guns N'
↪  Roses, Motley Crue, Van Halen, Rush and AC/DC.
|3733 Park East Drive ● Room 222 ● Cleveland, Ohio 44122 P: 216-831-3761 ● F: 216-514-4699 ● Email us
|©2014 Envision Networks. All rights reserved.
Site design by Single Source Marketing
```

### Doc3 (Good)

```
|Chinese Five-Spice Noodles with Shitake Mushrooms
|Recipes - Chinese Five Spice
Chinese Five-Spice Noodles with Shitake Mushrooms
Cook the noodles until tender according to the package directions. Drain and rinse under cold running
↪  water, and drain well again.
Bring 2 1/2 cups water to a boil in a small saucepan. Add the dried mushrooms, cover, and simmer for 3
↪  minutes. Strain the liquid through a paper coffee filter into a bowl to remove any grit, then
↪  squeeze the mushrooms over the bowl. Roughly chop the mushrooms, then set aside with the liquid.
Heat the butter and oil in a wok or large saucepan over medium heat. Add the Chinese five-spice powder
↪  and shallots and saute, stirring frequently, until tender, about 2 minutes. Add the garlic and
↪  saute 2 minutes more. Increase the heat to medium-high and add the fresh and reconstituted-dried
↪  shitakes with 1/2 cup of the reserved mushroom water. Cook, stirring frequently, until the
↪  mushrooms are tender, about 3 minutes. Add the soy sauce and rice wine. Increase the heat to high
↪  and cook, stirring, for 1 minute. Add the remaining mushroom-soaking water.
Add the noodles to the wok and stir until heated through and coated with the sauce, about 1 minute.
↪  Garnish with the green onions and sesame seeds, if using, and serve at once.
Makes 4 servings
```

**Doc4 (Good)**

```
"Is it necessary to purchase a travel book or is it realistic that we can get similar information from
↪  other resources? Usually, most individuals have a major question on buying a travel book. So here
↪  are the pros and cons of purchasing one such book.
Advantages of a Travel Book
A travel book, which may be a paperback or e-book, comes in handy while traveling. Glancing through a
↪  travel book enables you to understand the custom and culture of a particular place in the world.
↪  So you can adapt yourself to that particular environment and stay there comfortably for longer
↪  periods.
- They Come In Handy - The travel guide comes in various forms such as, e-books, paperbacks and the
↪  file formats. You can have easy access to these books, which would assist you with all details
↪  compatible to the region you are traveling to.
- They Provide Enormous Information - Electronic or traditional travel guides provide you with answers
↪  to all types of questions such as how to learn some sayings that can be used in the place where
↪  you are traveling to? How to get data on where to reside, what to see and where to eat? How to get
↪  a clear knowledge about the history of a specific region or the atmosphere that it has?
- They Suit To Your Requirements - To access full information about a specific country or a region,
↪  both types of general and specific travel books are made available. The e-book may easily fit into
↪  your e-book reader whereas the paperback can fit into your backpack.
Disadvantages of Travel Book
- The Price - The e-book and paperback travel guides are very expensive compared to the information
↪  obtained from travel websites or from those who have moved or traveled to that region.
- Qualitative Images In Travel Books - Most travel books are in black and white. Only a few e-books
↪  consist of colored photos. Hence make a thorough revision before purchasing a travel guide or an
↪  e-book.
- Travel Books Make The Trip Less Natural - Traveling can be made more spontaneous by acquiring
↪  suggestions from locals than from travel books."
```

# O   Decontamination

Table 25: **MMLU overlap removal comparison.** We remove overlaps detected with MMLU, in cases where a question and one of its options are detected in the text. For `Dolma-V1.7` [156], we sample 1/10th of the dataset for this analysis (roughly 230B tokens). For FineWeb-Edu [106], we use the 10B token subset released by the authors. Note that because our flagging rule prioritizes recall over precision, these numbers are likely to be overestimates of the true contamination rates.

| Dataset | Percentage of samples flagged |
|---|---|
| DCLM-BASELINE | 0.007% |
| Dolma-V1.7 | 0.001% |
| FineWeb-Edu | 0.009% |

In Section 4.6, we examined whether contamination explains the strong results of DCLM-BASELINE on MMLU and Hellaswag. Here, we present some analysis with respect to other open-source datasets and evaluation tasks. First, we show in Table 25 that the percentage of documents in DCLM-BASELINE that are flagged by our MMLU-specific decontamination is roughly similar to that for other high-performing contemporary datasets.

To expand our analysis to more evaluation tasks, instead of the decontamination that we performed in Section 4.6, here we follow a more general approach based on token overlaps. Overall, a generally applicable decontamination rule is difficult to specify, given the potentially subjective nature of what constitutes as contamination in text data as well as the diversity in formats across tasks. Following Touvron et al. [168], we search for contaminated tokens that exist in overlapping 10-grams (or longer) between DCLM-BASELINE and our downstream tasks. We measure the percentage of samples in each evaluation set where more than 80% of the tokens are contaminated (such samples are considered "dirty" per Touvron et al. [168]), as well as the percentage where less than 20% of the tokens are contaminated (considered "clean" by the same criterion).

We examine the difference of performance of the same 7B-2x model trained on DCLM-BASELINE, between the full evaluation set and the evaluation samples that are marked as "not dirty" per the criterion in Touvron et al. [168] (less than 80 % of the tokens in the sample are marked as contaminated), and between the full evaluation set and samples marked as "clean" using the same criterion (less than 20 % of the tokens are marked). Results can be seen in Figure 11, where we see

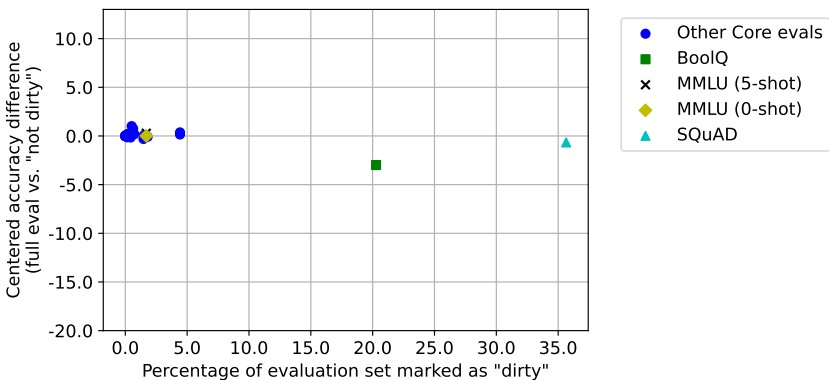

Figure 11: **Analysis of performance on the "not dirty" subset**. The x-axis is the percentage of samples from each evaluation task where more than 80% of the tokens are contaminated (such samples are considered "dirty" per Touvron et al. [168]). The y-axis is the performance of our 7B-2x model trained on DCLM-BASELINE over the full training set, minus the performance on the "not dirty". Each point is an evaluation task in our CORE subset, as well as MMLU. There is no clear correlation with changes in performance over the full and the "not dirty" evaluation subsets and contamination.

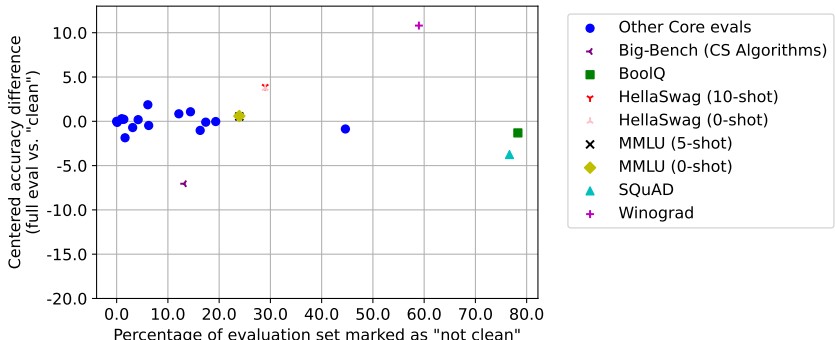

Figure 12: **Analysis of performance on the "clean" subset**. The x-axis is the percentage of samples from each evaluation task where more than 20% of the tokens are contaminated (such samples are considered "not clean" per Touvron et al. [168]). The y-axis is the performance of our 7B-2x model trained on DCLM-BASELINE over the full training set, minus the performance on the "clean" subset (less than 20% of the tokens contaminated). Each point is an evaluation task in our CORE subset, as well as MMLU. Most evaluation tasks (including MMLU) have similar performance in the full eval and in the "clean" subset.

that the difference in performance over the full dataset and the "not dirty" samples is minimal. In fact, for BoolQ and SQuAD, which are marked as highly contaminated, our model performs slightly better on the "not dirty" subset. Moreover, in Figure 12 we see that the difference in performance between the full evaluation set and the "clean" subset is similarly small for most datasets. While Hellaswag appears to be a notable exception, we showed in Section 4.6 that removing contaminated Hellaswag examples from our training set does not decrease performance. We note here that it's difficult to identify a correct threshold for what counts as a contaminated sample (as 20 % token overlap might lead to many false positives, but at the same time 80 % might be too high to detect all contaminated samples).

## P   Instruction tuning

Instruction tuning has emerged as a critical step to allow users to interact with pretrained language models [117, 120, 172, 174, 189]. To investigate whether models trained on DCLM-BASELINE can have strong instruction-following capabilities, we instruction-tune the DCLM-BASELINE (7B)

Table 26: **Instruction tuning results on AlpacaEval2.0.** We see that DCLM-BASELINE w/ OH-2.5 performs similarly to Mistral-7B finetuned also on OH-2.5, indicating similar behavior during instruction tuning. Also, with better data, we see DCLM-IT can be even better and can beat many existing models of similar scales.

| Model | AlpacaEval2.0 LC Win-rate (%) |
|---|---|
| *Our runs* | |
| DCLM-IT | 16.6 |
| Mistral-7B w/ OH-2.5 | 15.4 |
| DCLM-BASELINE w/ OH-2.5 | 13.8 |
| *Reported from the leaderboard* | |
| Llama-3-Instruct-8B | **22.9** |
| Mistral-v0.2-7B | 17.1 |
| Mistral-7B w/ OH-2.5 | 16.2 |
| Zephyr-Beta-7B | 13.2 |
| Vicuna-v1.3-13B | 10.8 |
| Gemma-Instruct-7B | 10.4 |
| Nous-Hermes-13B | 9.7 |
| DaVinci001 | 9.0 |
| LLaMA-2-Chat-13B | 8.4 |
| Alpaca-7B | 5.9 |

with the OpenHermes-2.5 (OH-2.5) dataset [163]. Specifically, we train our model to predict the response given the instruction from the instruction-tuning dataset. We train DCLM-BASELINE using the Adam [91] optimizer for 10 epochs with a 10% warmup ratio and a cosine learning rate schedule. We perform a hyperparameter search over the three learning rates $\{1e-7, 5e-6, 2e-5\}$ and report the best-performing numbers on the evaluation metrics i.e., AlpacaEval 2.0 length-controlled win-rate [54]. Post-training, we generate the responses for the instructions from AlpacaEval using a sampling temperature of 0.3 and maximum response length of 500. We benchmark our model with relevant baselines (e.g., LlaMA-2-Chat [168], Zephyr-7B [169]) taken directly from the AlpacaEval leaderboard as well as Mistral-7B [83] finetuned in the same manner as DCLM-BASELINE.

We present the results in Table 26. We find that DCLM-BASELINE finetuned with OH-2.5 outperforms various instruct models such as Zephyr-Beta-7B and Gemma-Instruct-7B. This indicates that we can elicit high-quality responses from the pretrained DCLM-BASELINE model with instruction-tuning. In addition, we observe that the DCLM-BASELINE slightly lags behind Mistral-7B-OH-2.5 meaning DCLM-BASELINE is competitive with other existing models of the same scale for finetuning. The small difference in performance *might* be attributed to the DCLM-BASELINE w/ OH-2.5 having longer generations on average than Mistral-7B w/ OH-2.5 or the lesser number of tokens seen during DCLM-BASELINE pretraining in comparison to Mistral-7B.

A follow-up question is whether DCLM-BASELINE can be finetuned to be even more competitive with models of similar scale. To further improve the instruction-following capabilities of DCLM-BASELINE, we curate a custom dataset, DCLM-IT, by combining some of the best instruction-tuning datasets including UltraFeedback [49], Tulu-v2 SFT [82], CodeFeedback [198], OH-2.5, Nectar [201], NoRobots [138], WildChat [196], WebInstruct [190], and StarCoder2-Self-OSS-Instruct [175]. There are roughly 4 million instances and 8 billion tokens in this dataset. Subsequently, we perform instruction-tuning and response generation of DCLM-BASELINE on this dataset with the training recipe mentioned above. We present the results in Table 26. We find that DCLM-IT outperforms DCLM-BASELINE w/ OH-2.5 by 2.8 percentage points. Our results highlight that there is room to enhance the instruction-following capabilities of DCLM-BASELINE with better datasets such as DCLM-IT. We further clarify that the current instruction-tuned models do not undergo any alignment procedures such as PPO [148], DPO [135] or others [16, 33, 56, 113]. We leave the development of aligned versions of DCLM-BASELINE for future research.

Table 27: Hyper-parameters for large scale run. Note the LR schedule uses a training length of 4.4T, but we do not train for the full length as we stop early and cooldown.

| Hyper-parameter/Config | Value |
|---|---|
| Training Tokens | 4,409,222,758,400 |
| Warmup Steps | 10,000 |
| Initial Learning Rate | $2 \times 10^{-3}$ |
| Weight Decay | 0.05 |
| Final Learning Rate | $3 \times 10^{-5}$ |
| Global Batch Size | 2048 |
| Accumulation Steps | 2 |
| Query-Key Normalization | True |
| Z Loss | $5 \times 10^{-6}$ |

Table 28: Results for 2.5T run, first row was run for 2T + 200B (cooldown), second row was run for 2T + 270B (cooldown), third is evaluation of average of weights of first two rows (0.2*CoolDown #1 + 0.8*CoolDown #2)

| Model | MMLU | CORE | EXTENDED |
|---|---|---|---|
| CoolDown #1 (200B Tokens) | 62.7 | 55.9 | 43.8 |
| CoolDown #2 (270B Tokens) | 63.4 | 55.9 | **44.3** |
| Final Model Soup | **63.9** | **56.0** | 43.7 |

## Q  Scaling up

The final run 2.5T shown in Figure 1 and Table 8 uses a two stage training procedure as followed in Groeneveld et al. [73], Hu et al. [80] and Team et al. [160]. For stage 1 we use the hyperparameters from Table 27.

After 2T tokens, we cooldown on a re-weighted pretraining distribution. For the cooldown distribution we use a mix of 70% DCLM-BASELINE with a tighter `fastText` threshold (top 7% rather than top 10%) and 30% ProofPile. We keep all the hyperparameters the same as Table 27, so we cooldown to the same final learning rate, just over a smaller number of tokens. Before the cool-down MMLU performance was approximately 52%, and the LR was approximately $1 \times 10^{-3}$.

We performed 2 independent cooldowns, one for 270B tokens and another for 200B tokens, and created a "model soup" [179] with a weight of 0.8 on the 270B cooldown and a weight of 0.2 on the 200B cooldown. Thus the total number of tokens seen by this model is 2.5T. We present results of each individual cooldown and the model soup in Table 28. The model in Figure 1 and Table 8 uses the final model soup after long-context training for 100B tokens as described in Appendix Q.2.

### Q.1  Instruction tuning the scaled up model

In Appendix P we show how instruction tuning the above "model soup" for 80B additional tokens leads to strong performance on instruction tuning benchmarks and out-performs instruction tuned variants of similar 7B models such as Gemma-7B. In addition to the IT benchmarks covered in Appendix P, In Table 29 we show that a small amount of instruction tuning provides large improvements in "Extended" evals at the cost of a small degradation in "Core" and "MMLU" evals. Notably we note that our GSM8k performance goes 2.5% to 52.5% which is comparable to other similar language models that mixed IT data into pretraining such as Gemma-7B.

### Q.2  Continual learning to extend context length

In this section, we present continual learning results for adapting the above DCLM-BASELINE 7B model (with an original context length of 2048) to a context length of 8192, similar to [185]. We follow the continual learning recipe described in [81], loading the DCLM-BASELINE 7B checkpoint

Table 29: **Effect of Instruction Tuning.** We compare our final model with its instruction-tuned variant, both trained on a 4k context length. Including instruction tuning maintains performance on language tasks such as MMLU and results in considerable gains on 5-shot GSM8K with chain-of-thought, demonstrating the effectiveness of this training in performing complex reasoning.

| Model | Params | Tokens | CORE | EXTENDED | MMLU | GSM8K |
|---|---|---|---|---|---|---|
| DCLM-BASELINE | 7B | 2.5T | **56.0** | 43.7 | **63.9** | 2.1 |
| DCLM-BASELINE-IT | 7B | 2.58T | 55.0 | **46.5** | 62.9 | **52.5** |

Table 30: Regular and long-context evaluations for DCLM-Baseline 7B model, and DCLM-8k 7B model that is adapted to 8192 context length through continual learning for additional $\sim 120$B tokens.

| Model | Params | Tokens | Context length | Regular Evaluations | | | Multi-Document Evaluations | | | |
|---|---|---|---|---|---|---|---|---|---|---|
| | | | | Core | MMLU | Extended | 1-Doc | 10-Docs | 20-Docs | 30-Docs |
| Llama-2 | 7B | 2T | 4096 | 49.2 | 45.8 | 34.1 | 48.5 | 27.3 | 25.6 | NA |
| DCLM | 7B | 2.5T | 2048 | 56.0 | **63.9** | 43.7 | 72.0 | 43.4 | NA | NA |
| DCLM-8k | 7B | 2.6T | 8192 | **57.1** | 63.7 | **45.4** | **76.9** | **49.8** | **46.1** | **38.8** |

and warming up to a maximum learning rate of $10^{-4}$ over 2000 steps, then annealing with a cosine schedule to $10^{-5}$. All other hyper-parameters remain the same as original pretraining. The global batch size remains $2^{22}$ tokens per optimization step. We employ a variable sequence length curriculum as in [132], including batches of sequences ranging from 64 to 8192 in length. For this continual learning stage, we train with a total of $\sim 120$B tokens randomly sampled from the main dataset and distributed as follows among different sequence lengths: $64 : 2^{33}, 128 : 2^{33}, 256 : 2^{33}, 512 : 2^{33}, 1024 : 2^{33}, 2048 : 2^{33}, 4096 : 2^{35}, 8192 : 2^{35}$. We use the Grow-Linear curriculum (from short to long sequences) with 4 cycles as described in [132]. As proposed by [131] and similar to [185] for long-context continual learning, we increase the RoPE [158] base frequency from 10,000 to 100,000 during the continual learning stage for long context adaptation. The average context length for 20-Docs and 30-Docs is $\sim 4k$ and $\sim 6k$, respectively. Hence, the original DCLM with context length of 2048 model has poor performance for these benchmarks.

We show that the above strategy results in similar performance on regular evaluations as the starting checkpoint and significantly improves on the multi-document question-answering evaluation. We use the evaluation setup described in [101]: the context is filled with $k$ documents followed by a question. We ensure that one of the $k$ documents includes the answer to the question (a.k.a., golden document). We use $k = 1, 10, 20, 30$, and for each case, we run the evaluation multiple times by changing the position of the golden document in the context and report the average. Results are reported in table 30. We demonstrate that long context adaptation results in a checkpoint (DCLM-8k) that matches the original model on regular evaluations and significantly improves multi-document QA showing its long-context capabilities.

## Q.3 Scaling up the 1B model

In addition to scaling up our 7B model, we also train our 1.4B parameter model for 4.3T tokens to show the utility of DCLM-BASELINE for smaller-scale models. Similar to the 7B model, we train for 4.3T tokens on DCLM-BASELINE combined with the StarCoder and ProofPile2 datasets, with the hyper-parameters from Table 31. Unlike the 7B model, we train for the full 4.3T tokens; we do not perform a separate cooldown or model souping phase. We present results for this model in Table 32.

## Q.4 Full results from Figure 1

In Table 33, we reproduce the results from Figure 1 in table format to ease future comparisons.

Table 31: Hyper-parameters for the 1B large-scale run.

| Hyper-parameter/Config | Value |
|---|---|
| Training Tokens | 4,319,385,600,000 |
| Warmup Steps | 5000 |
| Initial Learning Rate | $1 \times 10^{-2}$ |
| Weight Decay | 0.01 |
| Final Learning Rate | $3 \times 10^{-5}$ |
| Global Batch Size | 2048 |
| Accumulation Steps | 2 |
| Query-Key Normalization | True |
| Z Loss | $1 \times 10^{-6}$ |

Table 32: Evaluation for our 1.4B model compared to state-of-the-art small models.

| Model | Params | Tokens | Open dataset? | CORE | MMLU | EXTENDED |
|---|---|---|---|---|---|---|
| **Open weights, closed datasets** | | | | | | |
| Phi-1.5 | 1.3B | 0.15T | ✗ | 39.8 | 43.4 | 25.7 |
| Qwen2-1.5B | 1.5B | 7T | ✗ | 42.1 | **56.4** | **32.4** |
| Gemma-2B | 2.5B | 3T | ✗ | **43.3** | 40.8 | 26.6 |
| **Open weights, open datasets** | | | | | | |
| OLMo-1B | 1.2B | 3T | ✓ | 29.7 | 26.0 | 16.1 |
| SmolLM | 1.7B | 1T | ✓ | 36.3 | 30.0 | 21.2 |
| DCLM-BASELINE | 1.4B | 4.3T | ✓ | **45.2** | 47.5 | 28.1 |

## R    Account of compute costs

We note the compute cost of training runs for each competition scale in Table 1. In total, we estimate that our runs for DCLM sum up to approximately 1.2M H100 hours. Our precise estimate from our experimental test bed is 859K H100 hours for training, but this is likely an underestimate due to additional compute that was not tracked, such as due to training failures. Of the 859K H100 hours, we spent 1,700 on 411M parameter models, 140,000 on 1B models, 4,800 on 3B models, and 713,216 hours on 7B models.

## S    Bias and Toxicity Evaluations

We examine the toxicity and bias of DCLM-BASELINE and compare to other popular base models, running evaluations on several safety-based tasks including CivilComments [27], Copyright [34], BBQ [123], WinoGender [144], and RealToxicityPrompts [65].

We briefly review these benchmarks for those unfamiliar: CivilComments (higher is better) studies how accurately a model can identify toxic content, and we report Exact Match as a metric. Copyright (lower is better) measures the capability of models to reiterate copyrighted content, and we report LCS as a metric. Real Toxicity Prompts (lower is better) measures how easily a user can prompt a model to generate toxic content, and we report Toxic Fraction. BBQ (higher is better) is a question-answering dataset that measures how likely a model's biases affect its choices, and we report Exact Match. Winogender (higher is better) measures how likely a model is to reinforce a gender-based stereotype when infilling a gendered pronoun.

The results of this analysis can be seen in Table 34. Out datasets lead to models that score comparably to existing ones, trained on preexisting datasets. The model trained on DCLM-Baseline is similar to other popular base models, such as Llama and Mistral, in terms of generating toxic content, as demonstrated by the Real Toxicity Prompts scores. The BBQ and Winogender metrics also

Table 33: We reproduce the results from Figure 1 (left) in tabular format for ease of reference below.

| Model | Params | Tokens | Open dataset? | CORE | MMLU | EXTENDED |
|---|---|---|---|---|---|---|
| C4 | 412M | 8B | ✓ | 13.1 | 24.9 | 8.0 |
| Dolma v1 | 412M | 8B | ✓ | 13.6 | 25.2 | 9.0 |
| RefinedWeb | 412M | 8B | ✓ | 15.1 | 25.5 | 8.9 |
| RedPajama | 412M | 8B | ✓ | 15.4 | 24.8 | 7.6 |
| DCLM-Baseline (ours) | 412M | 8B | ✓ | 18.0 | 25.5 | 9.8 |
| FineWeb-Edu | 412M | 8B | ✓ | 18.3 | 25.5 | 10.1 |
| C4 | 1B | 29B | ✓ | 23.7 | 25.4 | 12.5 |
| RefinedWeb | 1B | 29B | ✓ | 25.1 | 24.0 | 12.9 |
| RedPajama | 1B | 29B | ✓ | 25.7 | 24.3 | 13.5 |
| Dolma v1 | 1B | 29B | ✓ | 25.8 | 24.3 | 13.5 |
| FineWeb-Edu | 1B | 29B | ✓ | 26.6 | 26.3 | 13.5 |
| OLMo | 1B | 2T | ✓ | 29.7 | 26.0 | 16.1 |
| DCLM-Baseline (ours) | 1B | 29B | ✓ | 30.2 | 23.8 | 15.4 |
| C4 | 3B | 56B | ✓ | 30.5 | 26.3 | 16.0 |
| Dolma v1 | 3B | 56B | ✓ | 31.5 | 27.6 | 16.7 |
| RedPajama | 3B | 56B | ✓ | 31.6 | 25.3 | 16.6 |
| RefinedWeb | 3B | 56B | ✓ | 33.2 | 25.2 | 17.3 |
| C4 | 7B | 138B | ✓ | 37.1 | 25.5 | 19.8 |
| DCLM-Baseline (ours) | 3B | 56B | ✓ | 37.1 | 25.2 | 20.0 |
| RedPajama | 7B | 138B | ✓ | 37.8 | 25.1 | 20.6 |
| Dolma v1 | 7B | 138B | ✓ | 38.2 | 25.9 | 19.6 |
| FineWeb-Edu | 7B | 138B | ✓ | 38.7 | 26.3 | 22.1 |
| Together-RPJ | 7B | 1T | ✓ | 39.4 | 25.2 | 21.8 |
| RefinedWeb | 7B | 138B | ✓ | 39.7 | 26.5 | 21.7 |
| LLM360/Amber | 7B | 1T | ✓ | 39.8 | 27.9 | 22.3 |
| FineWeb-Edu | 7B | 276B | ✓ | 41.9 | 37.3 | 24.5 |
| OLMo | 7B | 2T | ✓ | 42.7 | 28.4 | 24.6 |
| Gemma | 2B | 3T | ✗ | 43.3 | 40.8 | 26.6 |
| Falcon | 7B | 1T | ✓ | 44.1 | 27.4 | 25.1 |
| MPT | 7B | 1T | ✓ | 44.3 | 28.8 | 25.7 |
| DCLM-Baseline (ours) | 7B | 138B | ✓ | 44.8 | 42.2 | 0.0 |
| Llama 1 | 7B | 1T | ✗ | 46.3 | 34.1 | 28.7 |
| OLMo-1.7 | 7B | 2T | ✓ | 47.0 | 54.0 | 34.2 |
| LLM360/CrystalCoder | 7B | 1T | ✓ | 48.1 | 48.2 | 33.2 |
| DCLM-Baseline (ours) | 7B | 276B | ✓ | 48.7 | 51.9 | 32.5 |
| Llama 2 | 7B | 2T | ✗ | 49.2 | 45.8 | 34.1 |
| MAP-Neo | 7B | 4T | ✓ | 50.2 | 57.1 | 40.4 |
| DeepSeek | 7B | 2T | ✗ | 50.7 | 48.5 | 35.3 |
| DCLM-Baseline (ours) | 7B | 3T | ✓ | 57.1 | 63.7 | 45.4 |
| Llama 3 | 8B | 15T | ✗ | 57.6 | **66.2** | **46.3** |
| Gemma | 8B | 6T | ✗ | **57.8** | 64.3 | 44.6 |

| Model | CivilComments ↑ | Copyright ↓ | Real Toxicity Prompts ↓ | BBQ ↑ | WinoGender ↑ |
|---|---|---|---|---|---|
| DCLM-BASELINE | 0.53 | 0.01 | **0.07** | 0.65 | **0.62** |
| Llama-2-7B | 0.56 | 0.01 | 0.09 | 0.58 | **0.62** |
| Llama-3-8B | **0.74** | 0.01 | 0.09 | 0.67 | 0.57 |
| Mistral-v0.3-7B | 0.67 | 0.01 | 0.09 | **0.71** | 0.48 |

Table 34: **Bias and toxicity comparisons.** We compare the model trained on DCLM-BASELINE with existing similarly sized base models. We see that, with the exception of CivilComments (toxic comment identification), our model performs similarly to existing ones

demonstrate that our model's generations tend to contain biases similar to those of other base models. However, our model does not identify toxic comments as accurately as other base models, according to CivilComments. These metrics demonstrate that DCLM-Baseline has relatively similar amounts of bias and toxicity to the datasets that other pretrained models were trained on.

# T    Existing assets used

In this section, we describe the assets we use in our benchmark and their associated licenses.

## T.1    Evaluation data

Appendix G discusses all downstream tasks we use for our evaluation. Below we mention them again, and specify their licenses.

- The AGI Eval LSAT-AR dataset [200] is distributed under the MIT license as indicated in `https://github.com/zhongwanjun/AR-LSAT`.
- The ARC easy and ARC challenge datasets [43] are distributed under the Creative Commons Attribution-Sharealike 4.0 International license as indicated in `https://allenai.org/data/arc`.
- We use a series of 6 datasets from Big-Bench [18] (1) QA Wikidata, (2) Dyck languages, (3) Operators, (4) Repeat Copy Logic, (5) CS Algorithms, and (6) Language Identification. They are distributed under the Apache 2.0 license as indicated in `https://github.com/google/BIG-bench/blob/main/LICENSE`.
- BoolQ [41] is distributed under the Creative Commons Share-Alike 3.0 license as indicated in `https://huggingface.co/datasets/google/boolq`.
- CommonsenseQA [159] is available through the official website `https://www.tau-nlp.org/commonsenseqa` with no specific license attached.
- COPA [142] is distributed under the BSD-2 clause license as indicated in `https://shorturl.at/t7I4k`, though we note the original distribution website is no longer available.
- CoQA [140] contains several parts, each of which is distributed under its own license, indicated here `https://stanfordnlp.github.io/coqa/`. Namely, the authors mention that CoQA contains passages from seven domains and make five of these public under the following licenses:
  - Literature and Wikipedia passages are shared under CC BY-SA 4.0 license.
  - Children's stories are collected from MCTest which comes with MSR-LA license.
  - Middle/High school exam passages are collected from RACE which comes with its own license.
  - News passages are collected from the DeepMind CNN dataset which comes with Apache license.
- HellaSwag [192] is distributed under the MIT license as indicated in `https://github.com/rowanz/hellaswag/blob/master/LICENSE`.

- Jeopardy [89] is available through `https://www.kaggle.com/datasets/tunguz/200000-jeopardy-questions`, with no specific license attached.

- LAMBADA [122] is distributed under the Creative Commons Attribution 4.0 International license as indicated in `https://zenodo.org/records/2630551`.

- OpenBookQA [114] is distributed under the Apache 2.0 license as indicated in `https://github.com/allenai/OpenBookQA/blob/main/LICENSE`.

- PIQA [23] is distributed under the (Academic Free License v. 3.0 as indicated in `https://github.com/ybisk/ybisk.github.io/tree/master/piqa`.

- SQuAD [139] is distributed under the CC-BY-SA-4.0 license as indicated in `https://huggingface.co/datasets/choosealicense/licenses/blob/main/markdown/cc-by-sa-4.0.md`.

- The Winograd Schema Challenge [95] is distributed under the Creative Commons Attribution 4.0 International License license as indicated in `https://cs.nyu.edu/~davise/papers/WinogradSchemas/WS.html`.

- The Winogrande [146] is distributed under the Apache 2.0 license as indicated in `https://github.com/allenai/winogrande/blob/master/LICENSE`.

- We use a series of 4 additional tasks from the AGI Eval suite of datasets [200] (1) LSAT-LR, (2) LSAT-RC, (3) SAT-En, and (4) SAT-Math. These suite is distributed under the MIT license as indicated in `https://github.com/ruixiangcui/AGIEval/blob/main/LICENSE`.

- AQuA [99] is distributed under the Apache 2.0 license as indicated in `https://github.com/google-deepmind/AQuA/blob/master/LICENSE`.

- BBQ [123] is distributed under the CC-By-4 license as indicated in `https://github.com/nyu-mll/BBQ/blob/main/LICENSE`.

- We use a series of 9 additional datasets from Big-Bench [18]: (1) Conceptual Combinations, (2) Conlang Translation, (3) Elementary Math QA, (4) Logical Deduction, (5) Misconceptions, (6) Novel Concepts, (7) Strange Stories, (8) Strategy QA, and (9) Understanding Fables. They are distributed under the Apache 2.0 license as indicated in `https://github.com/google/BIG-bench/blob/main/LICENSE`.

- Enterprise PII classification [126] is distributed via `https://github.com/mosaicml/llm-foundry` as indicated in `https://www.patronus.ai/announcements/patronus-ai-launches-enterprisepii-the-industrys-first-llm-dataset-for-detecting-business-sensitive-information`. LLM-foundry itself is released under the Apache-2.0 license.

- GPQA-main and GPQA-diamond [141] are distributed under the MIT license as indicated in `https://github.com/idavidrein/gpqa/blob/main/LICENSE`.

- GSM8K [44] is distributed under the MIT license as indicated in `https://github.com/openai/grade-school-math/blob/master/LICENSE`.

- LogiQA [100] is distributed in through the official public repository at `https://github.com/lgw863/LogiQA-dataset` with no specific license attached.

- Math QA [11] is distributed under the Apache 2.0 license as indicated in `https://huggingface.co/datasets/choosealicense/licenses/blob/main/markdown/apache-2.0.md`.

- MMLU [78] is distributed under the MIT license as indicated in `https://github.com/hendrycks/test/blob/master/LICENSE`.

- PubMedQA [84] is distributed under the MIT license as indicated in `https://github.com/pubmedqa/pubmedqa/blob/master/LICENSE`.

- Simple arithmetic with spaces and without spaces [116] is distributed under the Apache-2.0 through `https://github.com/mosaicml/llm-foundry`.

- Social Interaction QA [147] is distributed by AllenAI under the CC-BY-4.0 license as indicated in `https://allenai.org/data/socialiqa`.

- SVAMP [125] is distributed under the MIT license as indicated in `https://github.com/arkilpatel/SVAMP/blob/main/LICENSE`.

- Trivia QA [86] is distributed under the Apache 2.0 license as indicated in `https://github.com/mandarjoshi90/triviaqa/blob/master/LICENSE`.

- The Winogender male and Winogender female datasets [144] are distributed under the MIT license as indicated in `https://github.com/rudinger/winogender-schemas/blob/master/LICENSE`.

## T.2  Raw sources

Our main external asset used in constructing DCLM-POOL and its filtered version DCLM-BASELINE is Common Crawl [45]. In their **Terms of Use** (`https://commoncrawl.org/terms-of-use`), they grant a limited, non-transferable license to access and use their service, primarily for innovation, education, and research, with several restrictions on usage. While being relatively permissive, it does not conform to any specific common licenses and emphasize that the usage must comply with local and international laws, and users must respect third-party copyrights. We urge the user's discretion in verifying their use abide by these terms-of-use.

In addition to the above, as described in Sections 4.4, 4.5 and 5 and Appendices M and P we make use of the following datasets:

1. OpenHermes2.5 [163] for instruction finetuning and to train some of our quality filters. While the authors do not provide a specific license and refer users to determine the license by following the links for the subsets they use[7], we note that the dataset is based in part on outputs from OpenAI models, and thus cannot be used for training new models for commercial purposes.

2. StarCoder [96] and StarCoder2 [107] are used for some of our ablations (Section 4). While constructed from permissive data by extracting datasets that mention permissive licenses (e.g. MIT, Apache 2.0), they involve various licenses, and as described in the Terms of Use[8], require the user to follow all terms-of-use and licenses of the different datasets it comprises of.

3. ProofPile2 [14] is used to scale up the dataset to the trillion tokens scale (Section 5). The authors do not alter the licenses of underlying datasets and ask users to follow guidelines and licenses as described in these datasets.

4. GSM8k [44] was used in some of the ablations in Section 4 and follows the MIT license.

5. RedPajama [166] is used for ablations in Section 4.5 and Appendix M. Note that **we do not release models or datasets that include this data**. RedPajama filters the datasets it uses keeping only permissive licenses, and refers the user to adhere to underlying licenses where appropriate, as described in `https://huggingface.co/datasets/togethercomputer/RedPajama-Data-1T`.

6. UltraFeedback [49] is used for instruction tuning and is under the MIT License.

7. Tulu V2 SFT mixture [82] is used for instruction tuning and is under the Open Data Commons License Attribution family.

8. CodeFeedback [198] is used for instruction tuning and is under the Apache 2.0 License.

9. Nectar [201] is used for instruction tuning and is under the Apache 2.0 License.

10. NoRobots [138] is used for instruction tuning and is under the Creative Commons Attribution Non-Commercial 4.0.

11. WildChat [196] is used for instruction tuning and is under the AI2 ImpACT License - Low Risk Artifacts ("LR Agreement").

12. WebInstruct [190] is used for instruction tuning and is under the Apache 2.0 License.

13. StarCoder2-Self-OSS-Instruct [175] is used for instruction tuning and is under the Open Data Commons License Attribution family.

---

[7]`https://huggingface.co/datasets/teknium/OpenHermes-2.5/discussions/9#65f2a0254ab77537428cc000`

[8]`https://huggingface.co/datasets/bigcode/the-stack#terms-of-use-for-the-stack`

### T.3 Libraries

The main libraries used in our benchmark pipeline are:

1. `transformers` uses the Apache 2.0 License.[9]
2. `PyTorch` uses a similar license to the 3-caluse BSD, and is defined in https://github.com/pytorch/pytorch/blob/main/LICENSE.
3. `OpenLM` [76] which is provided with MIT license.[10]
4. `llm-foundry` uses the Apache 2.0 License.[11]
5. `ChatNoir Resiliparse` uses the Apache 2.0 License.[12]
6. `BFF` uses the Apache 2.0 License.[13]
7. `Ray` uses the Apache 2.0 License.[14]
8. `slurm` is accessible under the GPL license.[15]
9. `fastText` [87] uses the MIT License.[16]
10. `nltk` uses the Apache 2.0 License.[17]
11. `langdetect` uses the Apache 2.0 License.[18]

In addition, the installation may include common ML and web development packages, and we urge commercial users to verify their endowment to refrain from license violations.

---

[9] https://github.com/huggingface/transformers/blob/main/LICENSE
[10] https://github.com/mlfoundations/open_lm/blob/main/LICENSE
[11] https://github.com/mosaicml/llm-foundry/blob/main/LICENSE
[12] https://github.com/chatnoir-eu/chatnoir-resiliparse/blob/develop/LICENSE
[13] https://github.com/allenai/bff/blob/main/LICENSE
[14] https://github.com/ray-project/ray/blob/master/LICENSE
[15] https://github.com/SchedMD/slurm/tree/master?tab=License-1-ov-file
[16] https://github.com/facebookresearch/fastText/blob/main/LICENSE
[17] https://github.com/nltk/nltk/blob/develop/LICENSE.txt
[18] https://github.com/Mimino666/langdetect/blob/master/LICENSE

# U  Datasheet

## U.1  Motivation

Q1 **For what purpose was the dataset created?** Was there a specific task in mind? Was there a specific gap that needed to be filled? Please provide a description.

- The purpose of DCLM and the associated DCLM-POOL and DCLM-BASELINE datasets is to enable the study of what makes a strong pretraining dataset for large language models. These models are transformative to society and act as the foundation of numerous applications, but they are often associated with steep costs. While prior work explores many curation techniques, these studies are often coupled with various architectural and training design choices and evaluated in different settings, making controlled comparison nearly impossible. This slows down progress and forces a lot of duplicate work between research teams. Prior work mainly focuses on data curation in the context of supervised datasets and smaller scales (see Section 2 and Appendix B). In our initial release of DCLM, we focus on 53 downstream language understanding tasks that also include reasoning abilities, math, and more. For details see Section 3.5 and Appendix G.

Q2 **Who created the dataset (e.g., which team, research group) and on behalf of which entity (e.g., company, institution, organization)?**

- DCLM-POOL and DCLM-BASELINE were created by a group of researchers with the following affiliations, listed in alphabetical order: Allen Institute for Artificial Intelligence, Apple, Carnegie Mellon University, Columbia University, Contextual AI, Cornell University, DatologyAI, Harvard University, Hebrew University, Juelich Supercomputing Center, Research Center Juelich, SambaNova Systems, Stanford University, SynthLabs, Tel Aviv University, Toyota Research Institute, TU Munich, University of California, Los Angeles, University of California, Santa Barbara, University of Southern California, The University of Texas at Austin, University of Washington.

Q3 **Who funded the creation of the dataset?** If there is an associated grant, please provide the name of the grantor and the grant name and number.

- Funding for this research was generously provided by the University of Washington, the University of Texas (Austin), the Institute for Foundations of Machine Learning (IFML), and Open Philanthropy.

Q4 **Any other comments?**

- We anticipate that DCLM benchmark, tooling and pools will drive data-centric research in ML and AI, fostering the development of the next generation of web-scale datasets, enhancing model abilities, lowering training costs and developing knowledge sharing across research teams.

## U.2  Composition

Q5 **What do the instances that comprise the dataset represent (e.g., documents, photos, people, countries)?** *Are there multiple types of instances (e.g., movies, users, and ratings; people and interactions between them; nodes and edges)? Please provide a description.*

- Each instance represented a web-crawled page (document). It contains the URL and the corresponding HTML content. Each sample is also tagged with metadata about its crawl time and additional information such as the detected language, for processed instances such as those in DCLM-BASELINE. Additional information can be found in Appendix E.

Q6 **How many instances are there in total (of each type, if appropriate)?**

- DCLM-POOL contains ∼200B documents, all of which are of the same instance, and comes from hundreds of millions of different sources. The subset DCLM-BASELINE contains approximately 3B documents.

Q7 **Does the dataset contain all possible instances or is it a sample (not necessarily random) of instances from a larger set?** *If the dataset is a sample, then what is the larger set? Is the sample representative of the larger set (e.g., geographic coverage)? If so, please describe how this representativeness was validated/verified. If it is not representative of the larger set, please describe why not (e.g., to cover a more diverse range of instances, because instances were withheld or unavailable).*

- DCLM-POOL is an unfiltered web-text corpus comprised of all Common Crawl data prior to 2023. As such, it represent the full breadth of possible instances from this source. However, we note that Common Crawl does not cover the entire web data, due to reach and compute limitations for instance. For our DCLM-BASELINE, we use various filtering and deduplication strategies as described in Section 4 in the explicit attempt to improve its quality for preatining, thus removing low-quality instances, and in doing so, becoming non-representative of the full set of instances. For a complete treatment and visualization of our data processing funnel, see Sections 4, 4.2 and 4.3 and Appendix E.

Q8 **What data does each instance consist of?** *"Raw" data (e.g., unprocessed text or images) or features? In either case, please provide a description.*

- Each sample contains a web-page url for and the extracted HTML content associated with. Additionally, each sample contains metadata fields shown in Table 9 (e.g., WARC-Type, WARC-date, Content-Type etc.).

Q9 **Is there a label or target associated with each instance?** *If so, please provide a description.*

- We do not provide any labels associated with the samples, as they are used to pretrain language models by performing self-supervised next-token prediction.

Q10 **Is any information missing from individual instances?** *If so, please provide a description, explaining why this information is missing (e.g., because it was unavailable). This does not include intentionally removed information, but might include, e.g., redacted text.*

- No, each sample is the full text as extracted from the HTML content, and the respective metadata.

Q11 **Are relationships between individual instances made explicit (e.g., users' movie ratings, social network links)?** *If so, please describe how these relationships are made explicit.*

- No, the dataset is released as it is with no explicit attempt to establish relationships between instances. Some links may be drawn based on metadata information such the as the source URL, but we do not deliberately form any such connections.

Q12 **Are there recommended data splits (e.g., training, development/validation, testing)?** *If so, please provide a description of these splits, explaining the rationale behind them.*

- No. The evaluation procedure is made of tasks as described in Section 3.5. We also attempt to prevent test set contamination in as described in Section 4.6 and Appendix O.

Q13 **Are there any errors, sources of noise, or redundancies in the dataset?** *If so, please provide a description.*

- DCLM-POOL is based on Common Crawl, which can be thought of as a snapshot of the internet at a given time. Hence, there can be considerable noise (e.g., placeholder text, broken links, failed extraction of HTML content, duplicate data, etc.)

Q14 **Is the dataset self-contained, or does it link to or otherwise rely on external resources (e.g., websites, tweets, other datasets)?** *If it links to or relies on external resources, a) are there guarantees that they will exist, and remain constant, over time; b) are there official archival versions of the complete dataset (i.e., including the external resources as they existed at the time the dataset was created); c) are there any restrictions (e.g., licenses, fees) associated with any of the external resources that might apply to a future user? Please provide descriptions of all external resources and any restrictions associated with them, as well as links or other access points, as appropriate.*

- Each sample is associated with a URL that links other external resources on the internet with no guarantee that the resources will exist in perpetuity or that that the resources will not change. However, the dataset itself contains already extracted HTML content and is thus self-contained for the purposes of this benchmark as described in Appendix C.

Q15 **Does the dataset contain data that might be considered confidential (e.g., data that is protected by legal privilege or by doctor–patient confidentiality, data that includes the content of individuals' non-public communications)?** *If so, please provide a description.*

- The dataset consists of data that was publicly accessible on the internet at the time of collection. However, it is possible that some of the data may include confidential information, such as private data that is unintentionally or maliciously made public.

Q16 **Does the dataset contain data that, if viewed directly, might be offensive, insulting, threatening, or might otherwise cause anxiety?** *If so, please describe why.*

- Given the diverse backgrounds of individuals worldwide, it is highly plausible that DCLM-POOL contains content that could be upsetting. Since our dataset consists of text scraped from the internet, it may include hateful, racist, sexist, and other offensive or toxic material. We consider the dataset a research artifact and hope future work will critically examine DCLM-POOL to develop improved safety filters. Our processed dataset, DCLM-BASELINE does apply a reproduction of the content-filtering from `RefinedWeb`. This involves url-based filtering using a domain banlist curated from Blacklists UT1[19] and a set of banned url-substrings curated from the LDNOOBW [20] list. While these banlists are extensive, they may still let in content that is harmful.

Q17 **Does the dataset relate to people?** *If not, you may skip the remaining questions in this section.*

- As a snapshot of the Internet, the dataset may include information about people which they shared intentionally or that was shared about them without permission.

Q18 **Does the dataset identify any subpopulations (e.g., by age, gender)?**

- Our DCLM-POOL does not explicitly identify subpopulations in its metadata, as it is unclear how one can define such division over raw text data from the web.

Q19 **Is it possible to identify individuals (i.e., one or more natural persons), either directly or indirectly (i.e., in combination with other data) from the dataset?** *If so, please describe how.*

- As names and other identifiers are frequent in web data, it is likely that some content can be linked back to specific individuals. However, in most public sites which Common Crawl scrape people publish such information willingly, knowing it will be visible and public.

Q20 **Does the dataset contain data that might be considered sensitive in any way (e.g., data that reveals racial or ethnic origins, sexual orientations, religious beliefs, political opinions or union memberships, or locations; financial or health data; biometric or genetic data; forms of government identification, such as social security numbers; criminal history)?** *If so, please provide a description.*

- Yes. DCLM-POOL is created from data that is available on the public internet. Since people often debate their political views, sexual preferences, religious beliefs and other such information, it is highly likely such information is contained in the dataset. While such information is often published willingly in the explicit intent that it will be publicly visible (see Q19), we do encourage additional research on filtering such data both to preserve privacy as well as to discard any potentially biased or toxic content from the training data of the models.

Q21 **Any other comments?**

- DCLM-POOL is a research artifact, and we aim for it to be useful for those studying ways to make internet-scale datasets safer.

---

[19]https://dsi.ut-capitole.fr/blacklists/index_en.php
[20]https://github.com/LDNOOBW/List-of-Dirty-Naughty-Obscene-and-Otherwise-Bad-Words/blob/master/en

### U.3 Collection process

**Q22 How was the data associated with each instance acquired?** *Was the data directly observable (e.g., raw text, movie ratings), reported by subjects (e.g., survey responses), or indirectly inferred/derived from other data (e.g., part-of-speech tags, model-based guesses for age or language)? If data was reported by subjects or indirectly inferred/derived from other data, was the data validated/verified? If so, please describe how.*

- Data is directly scraped from the public internet by Common Crawl.

**Q23 What mechanisms or procedures were used to collect the data (e.g., hardware apparatus or sensor, manual human curation, software program, software API)?** *How were these mechanisms or procedures validated?*

- We begin by downloading the entire Common Crawl data prior to 2023. We ran Python-based processing scripts to parse these archives, filtering low-quality or irrelevant content, deduplicate samples and in some cases decontaminate against downstream tests sets, and compute various model-based features. We ran processes on hundreds of AWS CPU nodes for Common Crawl parsing and data downloading. Model-based features were run on GPU clusters. For software links see Q37 and Appendix T or refer to https://datacomp.ai/dclm.

**Q24 If the dataset is a sample from a larger set, what was the sampling strategy (e.g., deterministic, probabilistic with specific sampling probabilities)?**

- DCLM-POOL is not a probabilistic sample. As described in Q7, DCLM-POOL contains all data from Common Crawl before 2023. Common Crawl is a sample of the Web, and we refer to Common Crawl documentation for details of their sampling process.

**Q25 Who was involved in the data collection process (e.g., students, crowdworkers, contractors) and how were they compensated (e.g., how much were crowdworkers paid)?**

- The authors participated in the data collection as part of an open-source effort. No researchers received specific compensation for their contributions to this project.

**Q26 Over what timeframe was the data collected? Does this timeframe match the creation timeframe of the data associated with the instances (e.g., recent crawl of old news articles)?** *If not, please describe the timeframe in which the data associated with the instances was created.*

- The data was downloaded between January 2023 and May 2023. The urls are collected from Common Crawl archives up to 2023. Common Crawl archives may include URLs from the early days of the internet. Hence, the download / collection timeframe does not match the creation timeframe. Additionally, future users of DCLM-POOL and its subsets will have to download data themselves using our tooling, though the snapshot should not be altered in any way.

**Q27 Were any ethical review processes conducted (e.g., by an institutional review board)?** *If so, please provide a description of these review processes, including the outcomes, as well as a link or other access point to any supporting documentation.*

- A formal ethics review / IRB has not been conducted to date because DCLM-POOL contains only data that is already publicly available as part of Common Crawl.

**Q28 Does the dataset relate to people?** *If not, you may skip the remaining questions in this section.*

- Yes. As described in Q17, people's data may appear as part of the data scraped.

**Q29 Did you collect the data from the individuals in question directly, or obtain it via third parties or other sources (e.g., websites)?**

- The data was gathered from data scattered across the web.

**Q30 Were the individuals in question notified about the data collection?** *If so, please describe (or show with screenshots or other information) how notice was provided, and provide a link or other access point to, or otherwise reproduce, the exact language of the notification itself.*

- Individuals were not notified about the data collection.

Q31 **Did the individuals in question consent to the collection and use of their data?** *If so, please describe (or show with screenshots or other information) how consent was requested and provided, and provide a link or other access point to, or otherwise reproduce, the exact language to which the individuals consented.*

- Following our usage of Common Crawl, we respect `robots.txt` files, which specify parts of websites that a crawler may access. It is, however, possible that some private content of people such as personal notes, medical information or private correspondence were uploaded to the internet without a person's consent or under the assumption the host site is private. To mitigate against such safety concerns we make an effort to exclude some malicious domains and filter such content as low quality.

Q32 **If consent was obtained, were the consenting individuals provided with a mechanism to revoke their consent in the future or for certain uses?** *If so, please provide a description, as well as a link or other access point to the mechanism (if appropriate).*

- While we have no control over the raw data scraped and hosted by Common Crawl, we will make an effort to provide user a mechanism to request exclusion of specific URLs, which can be filtered out of our DCLM-POOL and its derived datasets.

Q33 **Has an analysis of the potential impact of the dataset and its use on data subjects (e.g., a data protection impact analysis) been conducted?** *If so, please provide a description of this analysis, including the outcomes, as well as a link or other access point to any supporting documentation.*

- Bender et al. [19], Luccioni & Viviano [108] conducted such research that web-based datasets still contain substantial amounts of hate speech and sexually explicit content, even after filtering. Such content can propagate biases and harmful stereotypes when used to train language models, resulting in outputs that may be inappropriate or offensive in various contexts.

Q34 **Any other comments?**

- We anticipate and hope that future studies will leverage DCLM-POOL and DCLM-BASELINE to investigate techniques for building better web-scale datasets.

### U.4 Preprocessing, cleaning, and/or labeling

Q35 **Was any preprocessing/cleaning/labeling of the data done (e.g., discretization or bucketing, tokenization, part-of-speech tagging, SIFT feature extraction, removal of instances, processing of missing values)?** *If so, please provide a description. If not, you may skip the remainder of the questions in this section.*

- Yes. See Q7. For more details see Section 4 and Appendix E.

Q36 **Was the "raw" data saved in addition to the preprocessed/cleaned/labeled data (e.g., to support unanticipated future uses)?** *If so, please provide a link or other access point to the "raw" data.*

- The raw data is stored and accessible through Common Crawl. DCLM-POOL contains raw text data after HTML extraction using `resiliparse`.

Q37 **Is the software used to preprocess/clean/label the instances available?** *If so, please provide a link or other access point.*

- We use the following, open-source software to aid in data processing:
  - ChatNoir Resiliparse: https://github.com/chatnoir-eu/chatnoir-resiliparse
  - Ray: https://www.ray.io
  - BFF: https://github.com/allenai/bff
  - slurm: https://github.com/SchedMD/slurm
  - fastText: https://github.com/facebookresearch/fastText
  - nltk: https://github.com/nltk/nltk/blob/develop/LICENSE.txt
  - langdetect: https://github.com/Mimino666/langdetect

For a more complete list of software and associated licenses, please refer to Appendix T.

Q38 **Any other comments?**

- The creation of DCLM-POOL, DCLM-BASELINE, the DCLM tooling and our trained models relies heavily on tools developed by the open-source community and would not have been possible without it.

## U.5    Uses

Q39 **Has the dataset been used for any tasks already?** *If so, please provide a description.*

- The full dataset (and subsets) have been used to train hundreds of language models at various scales and compute budgets as presented in our main paper. We evaluate these models on our testbed of 53 zero- and few-shot downstream tasks. See Sections 3.5 and 4.

Q40 **Is there a repository that links to any or all papers or systems that use the dataset?** *If so, please provide a link or other access point.*

- No. There is, however, a leaderboard connected to DCLM. Those interested can review the submissions and examine publications that utilize our data. Refer to: https://datacomp.ai/dclm/leaderboard.

Q41 **What (other) tasks could the dataset be used for?**

- Large language models are now widespread and used for an incredibly large spectrum of tasks, ranging from spell-checking and translation to interactive agents. The dataset could provide the necessary data to pretrain such models. DCLM-POOL could also be used for sociological studies, such as examining biases and trends in human communication, as well as studying human behavior on the public internet.

Q42 **Is there anything about the composition of the dataset or the way it was collected and preprocessed/cleaned/labeled that might impact future uses?** *For example, is there anything that a future user might need to know to avoid uses that could result in unfair treatment of individuals or groups (e.g., stereotyping, quality of service issues) or other undesirable harms (e.g., financial harms, legal risks) If so, please provide a description. Is there anything a future user could do to mitigate these undesirable harms?*

- DCLM-POOL and its related datasets and models are not designed for use in production systems, particularly those involving sensitive areas such as race, gender identity or expression, ethnicity, sexual orientation, age, socioeconomic status, disability, religion, national origin, or creed. DCLM-POOL is unsuitable for applications that involve decision-making about individuals. Since DCLM-POOL is sourced from the internet, it inherently contains biases, unfairness, and stereotypes prevalent in society. It is intended solely as a research tool to examine language-modeling dataset curation on a large scale and to study the impact of various data curation methods on downstream models.

Q43 **Are there tasks for which the dataset should not be used?** *If so, please provide a description.*

- As mentioned in Q42, neither DCLM-POOL in its current state nor the subsets included in this paper should be used in decision-making software involving individuals. It is intended solely as a research tool for academic study.

Q44 **Any other comments?**

- Our aim with DCLM-POOL and DCLM was to establish a benchmark for the community to measure dataset progress across various dimensions (e.g., model performance on diverse tasks). We consider this essential for creating more effective and safer datasets, minimizing redundant efforts, promoting knowledge sharing, and making large language model research more accessible.

### U.6 Distribution

Q45 **Will the dataset be distributed to third parties outside of the entity (e.g., company, institution, organization) on behalf of which the dataset was created?** *If so, please provide a description.*

- Yes. We use HuggingFace datasets for public release.

Q46 **How will the dataset be distributed (e.g., tarball on website, API, GitHub)?** *Does the dataset have a digital object identifier (DOI)?*

- The dataset will be distributed via HuggingFace.

Q47 **When will the dataset be distributed?**

- DCLM-POOL and DCLM-BASELINE will be available starting June 2024.

Q48 **Will the dataset be distributed under a copyright or other intellectual property (IP) license, and/or under applicable terms of use (ToU)?** *If so, please describe this license and/or ToU, and provide a link or other access point to, or otherwise reproduce, any relevant licensing terms or ToU, as well as any fees associated with these restrictions.*

- We distribute our datasets in full, including extracted page content and associated metadata under a standard CC-BY-4.0 licence (see Appendix E). The code associated with DCLM is released under the MIT license. We also note that the use of this dataset is also subject to CommonCrawl's Terms of Use as described in `https://commoncrawl.org/terms-of-use`.

Q49 **Have any third parties imposed IP-based or other restrictions on the data associated with the instances?** *If so, please describe these restrictions, and provide a link or other access point to, or otherwise reproduce, any relevant licensing terms, as well as any fees associated with these restrictions.*

- We do not copyright samples in the dataset.

Q50 **Do any export controls or other regulatory restrictions apply to the dataset or to individual instances?** *If so, please describe these restrictions, and provide a link or other access point to, or otherwise reproduce, any supporting documentation.*

- No, the dataset is provided as individual samples with extracted content and associated metadata based on the content in Common Crawl hosted data.

Q51 **Any other comments?**

- We provide several subsets of DCLM-POOL in different sizes, along with extensive tooling to sample from it which makes it easy for any research entity to download and experiment with the data at scale suited for them.
  We release our code and dataset as open-source with permissive licenses as described in Q48.
  We, the authors, bear all responsibility for any violation of rights associated with this dataset. While we have made maximal efforts to respect all licenses of used assets and to mitigate any risks of causing harm, the responsibility for any misuse of the dataset by others does not rest with us. This dataset is intended solely for scientific research and not for use in production systems. We strongly encourage all users to adhere to local and national laws, respect privacy, and make every effort to avoid harming anyone when using this dataset.

### U.7 Maintenance

Q52 **Who will be supporting/hosting/maintaining the dataset?**

- HuggingFace currently hosts the datasets. The DCLM team will be responsible for maintaining the dataset.

Q53 **How can the owner/curator/manager of the dataset be contacted (e.g., email address)?**

- We can be contacted at `contact@datacomp.ai`.

Q54 **Is there an erratum?** *If so, please provide a link or other access point.*

- There are no errata at this time. If any issues arise, we will inform the public through our website at https://datacomp.ai/dclm.

Q55 **Will the dataset be updated (e.g., to correct labeling errors, add new instances, delete instances)?** *If so, please describe how often, by whom, and how updates will be communicated to users (e.g., mailing list, GitHub)?*

- Currently, there are no plans to update DCLM-POOL to maintain scientific integrity and comparability among participants in the DCLM competition. However, we will address user takedown requests (see Q56). DCLM-POOL is inherently noisy, and its release aims to encourage researchers to study dataset cleaning in the context of raw, web-crawled text samples.

Q56 **If the dataset relates to people, are there applicable limits on the retention of the data associated with the instances (e.g., were individuals in question told that their data would be retained for a fixed period of time and then deleted)?** *If so, please describe these limits and explain how they will be enforced.*

- Until we establish an automated method for takedown requests, users can contact us through contact@datacomp.ai with takedown requests and specify the offending URL.

Q57 **Will older versions of the dataset continue to be supported/hosted/maintained?** *If so, please describe how. If not, please describe how its obsolescence will be communicated to users.*

- This is the first version of DCLM-POOL and derivative DCLM-BASELINE dataset. We do not intend to maintain deprecated versions of DCLM-POOL. Any deprecation or modification will be announced on our website at https://datacomp.ai/dclm.

Q58 **If others want to extend/augment/build on/contribute to the dataset, is there a mechanism for them to do so?** *If so, please provide a description. Will these contributions be validated/verified? If so, please describe how. If not, why not? Is there a process for communicating/distributing these contributions to other users? If so, please provide a description.*

- Each proposed modification to the dataset will be addressed individually.

Q59 **Any other comments?**

- We encourage community members to contact us at contact@datacomp.ai with any suggestion or questions about dataset maintenance.

