# OpenReview forum: "DataComp-LM: In search of the next generation of training sets for language models"
_NeurIPS.cc/2024/Datasets_and_Benchmarks_Track — NeurIPS 2024 Track Datasets and Benchmarks Poster_

### Official Review · Reviewer_vx9q · 2024-07-19
**: The first benchmark for data curation DCLM**

**Rating:** 7
**Confidence:** 4
**Clarity:** The paper is well written and easy to…

**Review:**

DCLM is the first benchmark capable of measuring the performance of data curation. It can be used for fair comparisons between different methods in future data curation research. Models trained on DCLM-Baseline demonstrate superior performance on natural language understanding tasks compared to other models of the same scale. Through extensive experiments, the validity of the components that constitute DCLM and DCLM-Baseline is well demonstrated, providing valuable references for future related research.

However, DCLM proposes a unified experimental environment for evaluating data curation performance but does not introduce new methods for measuring data curation performance. Also, to demonstrate the superiority of the DCLM-Baseline datasets, it is important to show that training the same model architecture by using the datasets leads to more performance gain than the existing datasets. It is also recommended that the same effects across various model architectures be shown through experiments, which is not shown in this paper.

**Strengths:**

The DCLM framework facilitates a fair comparison of data curation methods, while the DCLM-Baseline proves valuable for language model training. Notably, the experimental results presented in this paper are valuable enough to be referenced in future related studies.

**Additional Feedback:**

-	In Table 2, it appears that "Extended" is incorrectly labeled as "Aggregated."

**Correctness:**

Although experiments are conducted only once due to the expensive training costs, this limitation is mitigated by performance measurement across multiple tasks. Additionally, their evaluation methodology aligns with widely recognized benchmarks, ensuring the reliability of their results.

**Documentation:**

The paper includes all the information required for the benchmark (DCLM) and dataset (DCLM-Baseline).

**Ethics:**

The paper is well within the ethical guidelines, and no ethical issues are suspected.

**Limitations:**

-	DCLM does not evaluate model’s bias and toxicity. In addition to performance, in natural language understanding, these factors are also crucial for model assessment. Since the bias in training datasets often translates to model bias, evaluation criteria related to these aspects are essential for an accurate assessment of data curation performance[1].
-	Performance of data curation may vary depending on the raw dataset. For example, appropriate filtering rules may differ based on the data source, whether it is news, blogs, or books. However, this is not discussed in the paper. Additional experiments and analysis on this topic are necessary.

[1]	Counterfactual data augmenta- tion for mitigating gender stereotypes in languages with rich morphology

**Opportunities For Improvement:**

Is there a specific reason for using the Open-LM architecture? Unlike Llama or Mistral, the superiority of this architecture has not been demonstrated, so explanation for the choice would be helpful. Additionally, including experiments on how data curation performance varies with different model architectures in future work may be beneficial.

**Relation To Prior Work:**

The differences from previous studies are clearly explained.

**Summary And Contributions:**

This paper proposes DCLM, a framework designed to evaluate data curation processes suitable for training Generative Language Models. DCLM provides a standardized environment that includes raw datasets, model architectures, hyperparameters, and evaluation tasks, enabling researchers to measure and compare the performance of data curation. Additionally, based on DCLM, they propose a new training dataset for LLMs, named DCLM-Baseline.

---

> ### Author Rebuttal · Authors · 2024-08-17
>
> We thank the reviewer for the positive feedback. We now address the three main points raised by the reviewer
> ## Model architecture
> > Is there a specific reason for using the Open-LM architecture? Unlike Llama or Mistral, the superiority of this architecture has not been demonstrated, so explanation for the choice would be helpful.
>
> First, to clarify: We primarily think of OpenLM as a training codebase, not necessarily a distinct model architecture. In fact, the specific model architecture we use for experiments in DCLM (implemented in OpenLM) is very similar to Llama and Mistral, and we view our model architecture as exchangeable with that of Llama and Mistral.
>
> The main reason for using the OpenLM training codebase was that neither Llama nor Mistral released training code for their models, and training models from scratch is a core part of the DCLM workflow. The OpenLM codebase allows us to train Llama-style models from scratch for model sizes up to 7B parameters. Moreover, the OpenLM code is computationally efficient and the resulting models achieve competitive evaluation scores.
>
> Of course, there are also other training frameworks (e.g., Megatron), as Reviewer 5qZv25 pointed out. We chose OpenLM because it achieved the best training performance for our model range of interest (up to 7B model parameters).
>
>
> > Additionally, including experiments on how data curation performance varies with different model architectures in future work may be beneficial.
>
> We agree with the reviewer that this is a very important question. Our submission already contained some results showing that the ordering of training sets is largely preserved under different model architectures. Specifically, Figure 3 in our submission shows that across three compute scales (400M-1x, 1B-1x, 7B-1x), there is a good correlation between performance at the three scales. Since the different compute scales also involve different architecture parameters (e.g., the number of layers or the size of the fully-connected layers), this experiment already shows that relative data curation performance is preserved under some architecture variations.
>
> To stress test this point more, we ran three additional sets of experiments for this rebuttal:
> We added an additional compute scale (3B model parameters, Chinchilla training factor 1x) and trained corresponding models on multiple training sets.
> We trained models with a Gemma-inspired architecture (GeGLU and RMSNorm) at the 1B parameter scale on multiple training sets.
> We trained models with a Mamba-style architecture at the 1B parameters scale on multiple training sets.
>
> Results for these can be seen in the Table below, where we list our Core evals for each of the three variants, for ten different training sets (with four of them being existing datasets, and the other six being variants of our DCLM-Baseline):
>
>
> | Training Dataset | 3B - 1x model | Gemma-inspired 1B-1x model | Mamba-inspired 1B-1x model |
> | ---------------------------- | ------------- | -------------------------- | --------------------------- |
> | C4 | 0.305 | 0.241 | 0.204 |
> | Dolma v1 | 0.315 | 0.245 | 0.197 |
> | RedPajama | 0.316 | 0.263 | 0.204 |
> | RefinedWeb | 0.332 | 0.269 | 0.218 |
> | DCLM-Baseline (variant 1) | 0.371 | 0.303 | 0.248 |
> | DCLM-Baseline (variant 2) | 0.375 | 0.311 | 0.241 |
> | DCLM-Baseline (variant 3) | 0.369 | 0.299 | 0.246 |
> | DCLM-Baseline (variant 4) | 0.366 | 0.308 | 0.227 |
> | DCLM-Baseline (variant 5) | 0.360 | 0.297 | 0.245 |
> | DCLM-Baseline (variant 6) | 0.369 | 0.305 | 0.233 |
>
>
> We can also see the above results as figures attached. Overall, we find that the 3B-1x models correlates strongly with 7B-1x models and the Mamba/Gemma-like architectures with our existing 1B-1x scale.

---

> > ### Author Rebuttal · Authors · 2024-08-17
> >
> > ## Evaluation of model bias and toxicity
> >
> > We agree that model bias and toxicity are important performance dimensions. Our evaluation suite at the time of submission already contained the BBQ dataset designed to detect
> > model’s biases along nine social dimensions (see Page 21 / Line Number 732 in the supplementary material PDF). For the purpose of this rebuttal, we augmented our model bias and toxicity evaluation by adding the HELM evaluation suite to our testbed.
> > | Model         | CivilComments (Exact Match) | Copyright (LCS) | Real Toxicity Prompts (Toxic Fraction) | BBQ (Exact Match) | Winogender |
> > |---------------|------------------|-----------------------|-----------------------------------|-----------------------------------------------|---------------------|
> > | DCLM Baseline      | 0.53             | 0.01                  | 0.07                              | 0.65                                          | 0.62                |
> > | Llama 2 7b       | 0.56             | 0.01                  | 0.09                              | 0.58                                          | 0.62                |
> > | Llama 3 8b  | 0.74             | 0.01                  | 0.09                              | 0.67                                          | 0.57                |
> > | Mistral 7b v0.3  | 0.67             | 0.01                  | 0.09                              | 0.71                                          | 0.48                |
> >
> > ​​We briefly review these benchmarks for those unfamiliar: CivilComments (higher is better) studies how accurately a model can identify toxic content. Copyright (lower is better) measures the capability of models to reiterate copyrighted content. Real Toxicity Prompts (lower is better) measures how easily a user can prompt a model to generate toxic content. BBQ (higher is better) is a question-answering dataset that measures how likely a model’s biases affect its choices. Winogender (higher is better) measures how likely a model is to reinforce a gender-based stereotype when infilling a gendered pronoun.
> >
> > Given these results, our datasets lead to models that score comparably to existing datasets and models. DCLM Baseline is similar to other popular base models, such as Llama and Mistral, in terms of generating toxic content, as demonstrated by the Real Toxicity Prompts scores. The BBQ and Winogender metrics also demonstrate that our model’s generations tend to contain biases similar to those of other base models. Our benchmarks also show that DCLM Baseline tends to reiterate copyrighted content less than similar base models. However, our model does not identify toxic comments as accurately as other base models, according to CivilComments. These metrics demonstrate that DCLM Baseline has relatively similar amounts of bias and toxicity to other pretrained checkpoints.
> >
> > ## Data curation on other raw datasets
> >
> > We agree with the reviewer that data curation methods may depend on the raw data source,
> > and in fact expect this to be the case (distinct data sources such as code would be a clear example). One of our experiments in Appendix L also suggests this, where we see that the classifier-based filtering we apply on Common Crawl, when added on top of already high quality datasets (such as Wikipedia and Arxiv), is not useful in improving the quality of the datasets when mixed in with already filtered CommonCrawl data.
> >
> > For the purpose of our submission though, we decided to focus on filtering Common Crawl as this is generally the largest component of current (open-source) pre-training datasets and broadly encompasses many kinds of data including news, blogs, and books. This investigation already led to a large submission and a strong-performing dataset, so we felt that filtering other sources (or specific slices of Common Crawl) with specialized methods is beyond the scope of our paper and a great direction for future work. Indeed, we hope others can explore this more in our mixing track!

---

> > > ### Comment · Reviewer_vx9q · 2024-08-30
> > > **Comment to the rebuttal.**
> > >
> > > The rebuttal clarifies some concerns of the reviewer. I will raise the score.

---

### Official Review · Reviewer_5qZv · 2024-07-25
**Nice initiative**

**Rating:** 7
**Confidence:** 4
**Correctness:** Yes, the claims made are reasonable.
**Clarity:** Yes

**Review:**

This is a well-motivated, solid, and timely effort that deals with important empirical issues in LLM research. I feel this is a good initiative to push forward LLM. The overall writing, structure and quality is good. I think this is nice work to be accepted by D&B track.

**Strengths:**

The strengths of this paper are clear. Firstly, the authors provide a comprehensive pipeline to control the experiment variables and to have a clean study for LM. Then, the authors provide comprehensive datasets for train corpus and for evaluation, making the results reliable. Lastly, the authors show with their tool set that we could do better token efficiency in model training, opening a door for community.

**Additional Feedback:**

NA

**Documentation:**

I think yes since this paper mainly works on data recepies, but I don't find a dedicated page explaining the data (preprocessing, usage, statistics, etc.). Thank the authors to point this to me.

**Limitations:**

I don't think there is notable limitations to be addressed in particular.

**Opportunities For Improvement:**

1. In table 1, I am a bit confused here. What are these scales doing? Does it concern certain splits in the 240T corpus (predetermined) or just a mode to the pipeline? Do we have the flexibility to choose whatever model scale and data size? Basically, I would like to ask about the flexibility in data source, model, evaluation, etc.
2. In L54, the authors mention 416 baseline experiments. What does these experiment include? I don't see any reference to this in later parts. And how these 416 experiments can be reused by participants?
3. I also have concerns on the usage of this framework. Can participants use whatever training framework and use whatever GPU cluster (not only AWS)?  For the latter, I am asking about the local / offline deployment of this framework. For the former, I think if we scale to larger models, other frameworks like megatron has better support.

**Relation To Prior Work:**

Yes

**Summary And Contributions:**

This paper presents DCLM (DataComp for Language Models) to benchmark on language training data recipes. This is important to have controlled study on data while keeping rest parts of LM like training, architecture, optimization, evaluation, consistent. The authors firstly prepare a tool set for conducting the above experiments. Then, the authors also provide a 240T CC-based corpus as data pool. Lastly, the authors experiment under these settings to train models from small to medium size to draw some conclusive observations. With their curated pipeline, the authors provide a strong baseline to have comparable results to Mistral 7B or Llama3 8B.

---

> ### Author Rebuttal · Authors · 2024-08-17
>
> Thank you for the positive review! We address the three opportunities for improvement below.
>
> > In table 1, I am a bit confused here. What are these scales doing? Does it concern certain splits in the 240T corpus (predetermined) or just a mode to the pipeline? Do we have the flexibility to choose whatever model scale and data size? Basically, I would like to ask about the flexibility in data source, model, evaluation, etc.
>
> The competition scales are indeed predetermined splits (subsets) of the 240T overall corpus. Since working with the full 240T corpus is computationally very demanding (approximately 1 million CPU hours for the RefinedWeb processing with our pipeline), we created smaller standardized data pools so that participants in DCLM can work with the pool sizes they can afford computationally. In addition, having multiple scales at which to evaluate data curation techniques enables us to study scaling trends for data curation.
>
> Participants indeed have the flexibility to choose the competition scales that best suit their available compute. In Table 1, we have paired specific model and pool sizes so that DCLM comes with a set of predetermined training setups, which makes it easier to compare different data curation techniques. These competition scales are the official DCLM scales we track in the leaderboard. Of course, researchers may also find the DCLM tooling useful for other experiments that do not exactly follow our prescribed setup in Table 1. We welcome creative uses of our DCLM framework for research that does not fit neatly into our official leaderboards.
>
> Lastly, since the reviewer also asked about flexibility with respect to evaluation: regardless of competition scale, we have a standardized evaluation testbed of 53 downstream tasks that applies to all scales. Some test sets such as MMLU only provide signal at the larger scales, but we generally still advise to run all evaluations for every scale so it is possible to spot new phenomena, e.g., when an improved training set shows non-trivial performance on a test set for which previous training sets at the same scale achieved only random performance. If the cost of running the evaluation suite is a concern, we suggest running only our Core evaluations, which show signal across all scales in DCLM (see Figure 1 left).
>
> > In L54, the authors mention 416 baseline experiments. What does these experiment include? I don't see any reference to this in later parts. And how these 416 experiments can be reused by participants?
>
> The 416 baseline experiments refer to the total of 416 individual model training runs (on various training sets) we conducted for DCLM. These experiments are the basis for all results in our paper. Not all 416 models appear in one of the main tables or figures because some of the experiments were exploratory and did not lead to relevant results (e.g., hyperparameter variations). To clarify what the 416 baseline experiments include, we will provide a CSV file with information about each experiment in our [git repository](https://github.com/mlfoundations/dclm), and point to the file in our paper. In addition, we will include a table with all experiments at the largest competition scale (7B-2x) in the paper.
>
> Not all of the 416 experiments are re-usable by participants, e.g., because some of them were conducted on earlier versions of our infrastructure or training data. Our code repository for DCLM provides code for participants to re-use the experiments in our paper that are most relevant for future work.
>
>
> > I also have concerns on the usage of this framework. Can participants use whatever training framework and use whatever GPU cluster (not only AWS)? For the latter, I am asking about the local / offline deployment of this framework. For the former, I think if we scale to larger models, other frameworks like megatron has better support.
>
> Participants in DCLM must use our provided training code to ensure that the resulting model performance numbers are comparable. Any GPU cluster should work as long as it is supported by PyTorch. We have already trained models with our framework on GPU clusters other than AWS (e.g., the academic clusters TACC and JSC).Thus, local / offline deployments of our framework are possible.
>
> Beyond directly participating in DCLM, our data tooling will hopefully also be useful to researchers independently of our training framework.
>
> Regarding larger models, it is important to remember that DCLM aims to enable rapid iteration of new datasets ideas. Hence our focus was specifically on quickly training small or medium-sized models with up to 7B parameters. In this regime, other frameworks such as megatron are less relevant. Before deciding on OpenLM as the training framework for DCLM, we compared OpenLM to Megatron and found that OpenLM was about 50% faster for the model sizes relevant for DCLM. Nonetheless, outside of participating in DCLM, you are more than welcome to use the data with a different training framework to train and study larger models.

---

> > ### Comment · Reviewer_5qZv · 2024-08-19
> >
> > Thank the authors for clarification. I am inclined to accept

---

### Official Review · Reviewer_JUUa · 2024-07-25
**DataComp-LM Review**

**Rating:** 8
**Confidence:** 4
**Clarity:** the paper is well written.

**Review:**

I would summarize the review in terms of pros and cons.

__pros__:
- comprehensive analysis of data curation techniques.
- extensive evaluation across a wide range of downstream tasks.
- detailed documentation of data collection and model training processes.

__cons__:
- requirement for a form to access data may not fully comply with accessibility guidelines.

**Strengths:**

- documented experiments on data curation strategies such as deduplication, filtering, and data mixing, with model scales ranging from 412M to 7B parameters.
- a full evaluation containing 53 downstream tasks suitable for base model evaluation.

**Additional Feedback:**

no extra feedback.

**Correctness:**

the paper is about a framework and dataset but i could not access the dataset to review.

**Documentation:**

The paper includes sufficient information on how the data was collected, by whom, and from where. The paper states that they have released the DCLM infrastructure, models, and training sets at https://github.com/datacomplm/DCLM. However, I found it difficult to obtain the data. For example, to access the candidate pool, we need to fill out a form: https://www.datacomp.ai/dclm/.

According to the NeurIPS guidelines:

"A key criterion is accessibility: datasets should be available and accessible, i.e. the data can be found and obtained __without__ a personal request to the PI."

Does requiring a form for access data contradict the principle of accessibility?

**Ethics:**

no ethical concerns, but i could not find if they did remove private information such as email address, etc.

**Limitations:**

no limitations.

**Opportunities For Improvement:**

no feedback. I was going to ask about experimenting (for improvement) with other concurrent work, FineWeb, which they have already addressed in their arXiv version.

**Relation To Prior Work:**

the paper discusses and directly compares with related works.

**Summary And Contributions:**

DataComp-LM (DCLM) is a comprehensive framework designed for building and training large language models (LLMs) with diverse datasets. They presents the DCLM dataset, a standardized corpus of 240T tokens extracted from Common Crawl. They also introduce DCLM benchmark which experiment data curation strategies such as deduplication, filtering, and data mixing at model scales ranging from 412M to 7B parameters. The resulting dataset, DCLM-BASELINE , enables training a 7B parameter language model from scratch to 63% 5-shot accuracy on MMLU with 2T training tokens.

---

> ### Author Rebuttal · Authors · 2024-08-17
>
> Thank you for your time! We are glad you appreciated the paper.
>
> ## Access Form
> We agree with this concern. Since the submission, we have addressed this by moving all of DCLM-Pool to a publicly accessible S3 bucket (provided by Common Crawl) and no longer require a form. This migration has been completed and the information has been updated on our website  https://www.datacomp.ai/dclm/.

---

> > ### Comment · Reviewer_JUUa · 2024-09-04
> > **Response by Reviewer**
> >
> > Thank you so much for providing the accessibility; my concern has been resolved. I did not take this concern into account as part of the scoring, and a value of 8 (clear accept) definitely shows the paper's great contribution.

---

### Official Review · Reviewer_gmJb · 2024-07-26
**Comprehensive benchmark with sufficient baseline experiments**

**Rating:** 7
**Confidence:** 5
**Clarity:** Yes.

**Review:**

Pros:
- This paper provide a comprehensive benchmark for language model training data curation. It helps researchers focus on data curation only to improve language models. This paper provides lots of valuable insights on how to extract/deduplicating/filtering/mixing to get a high-quality training dataset for language models.
- In this benchmark, a large-scale (240T tokens) training corpus is collected and open-source for participants to start their experiments of data curation.
- The authors conduct lots of experiments and ablation studies, including a SOTA experiment. These related experimental results can be a guide for later researchers on model scaling, data curation, and so on.

Cons:
- Considering the source (common crawl) and large scale of the dataset, implication on ethics, toxicity and other social perspectives of the proposed benchmark and dataset can be analyzed and discussed more in the paper.
- Missing related works in terms of data-centric benchmark for LLM, especially the Data-Juicer's serial challenges that also iterate data [1] (such as FT-Data Ranker, BetterMixture and BetterSynth)

---
[1] SIGMOD'24 "Data-Juicer: A One-Stop Data Processing System for Large Language Models"

**Strengths:**

Plz see pros above.

**Additional Feedback:**

None.

**Correctness:**

The claims in this paper are generally correct, and the proposed dataset is constructed in a sound way.

**Documentation:**

Yes.

**Ethics:**

The proposed datasets are collected from public web-crawled data, which involving many potential ethics problems. The authors claims that they consider this dataset as a research artifact and leave it to future works.

**Limitations:**

Discussed in Section 6.

**Opportunities For Improvement:**

Plz see cons above.

**Relation To Prior Work:**

Plz see point 2 of cons above.

**Summary And Contributions:**

This paper introduce a benchmark named data competition for improving language models, which includes a standardized corpus of 240T tokens, pretraining recipes in different scales, and a suite of 53 downstream evaluations. It allows participants to explore and experiment different dataset curation and processing strategies to produce truly high-quality dataset for language models. Besides, this paper provides lots of experiments and ablation studies to show some insights and key conclusions on data-centric perspective for later researchers, and a SOTA public training set for 7B language models.

---

> ### Author Rebuttal · Authors · 2024-08-17
>
> Thank you for your time and feedback! We hope to address your two main concerns below.
>
> ## Ethics and toxicity
> We agree that these considerations are important given the scale of our datasets and models. We note that our “Extended” evaluation does contain some safety-based tasks (e.g., Winogender and BBQ). Since the original submission, we have also expanded our evaluations to include additional safety-based tasks via HELM (CivilComments, Copyright, and RealToxicityPrompts) and have run them on our main models as well as several important baselines. Overall, the results are as follows:
>
>
> | Model         | CivilComments (Exact Match) | Copyright (LCS) | Real Toxicity Prompts (Toxic Fraction) | BBQ (Exact Match) | Winogender |
> |---------------|------------------|-----------------------|-----------------------------------|-----------------------------------------------|---------------------|
> | DCLM-Baseline      | 0.53             | 0.01                  | 0.07                              | 0.65                                          | 0.62                |
> | Llama 2 7b       | 0.56             | 0.01                  | 0.09                              | 0.58                                          | 0.62                |
> | Llama 3 8b  | 0.74             | 0.01                  | 0.09                              | 0.67                                          | 0.57                |
> | Mistral 7b v0.3  | 0.67             | 0.01                  | 0.09                              | 0.71                                          | 0.48                |
>
> We briefly review these benchmarks for those unfamiliar: CivilComments (higher is better) studies how accurately a model can identify toxic content. Copyright (lower is better) measures the capability of models to reiterate copyrighted content. Real Toxicity Prompts (lower is better) measures how easily a user can prompt a model to generate toxic content. BBQ (higher is better) is a question-answering dataset that measures how likely a model’s biases affect its choices. Winogender (higher is better) measures how likely a model is to reinforce a gender-based stereotype when infilling a gendered pronoun.
>
> Given these results, our datasets lead to models that score comparably to existing datasets and models. DCLM-Baseline is similar to other popular base models, such as Llama and Mistral, in terms of generating toxic content, as demonstrated by the Real Toxicity Prompts scores. The BBQ and Winogender metrics also demonstrate that our model’s generations tend to contain biases similar to those of other base models. Our benchmarks also show that DCLM-Baseline tends to reiterate copyrighted content less than similar base models. However, our model does not identify toxic comments as accurately as other base models, according to CivilComments. These metrics demonstrate that DCLM Baseline has relatively similar amounts of bias and toxicity to other pretrained checkpoints.
>
> In the final version, we will expand upon and highlight these results in the manuscript.
>
>
> ## Related work
> Thank you for pointing us to this related work. While we see DataComp-LM as having a different scope compared to the Data-Juicer competitions (i.e., focusing on LLM pre-training rather than fine-tuning or multi-modal image understanding), we agree that FT-DataRanker / BetterMixture / Bettersynth also share the core aim of more rigorous data-centric benchmarking. We will cite and discuss the relation to these works in the final version. We also welcome any additional pointers to related work the reviewer has.

---

### Comment · Area_Chair_M7Rt · 2024-08-29
**Please engage in the author-reviewer discussion**

Dear Reviewers,

Thank you for your hard work on the papers and reviews. Please note that the deadline for the author-reviewer discussion period is approaching (August 31, 2024). Some of you have engaged in discussions with authors-thank you! For reviewers who have not yet, please discuss with authors as this is a very important part of the reviewing process and authors are eager to have further feedback from you. If there are any changes to your scores, kindly provide explanations for these adjustments.

---

### Decision · Program_Chairs · 2024-09-26

**Decision:**

Accept (Poster)

**Comment:**

The paper presents a data competition benchmark DataComp-LM, including a standardized corpus of 240T tokens, pretraining recipes in different scales, and a suite of 53 downstream evaluations.

Pros:
1. The paper is well-motivated, providing timely resources.
2. The paper presents a comprehensive data curation pipeline.
3. The paper conducts lots of experiments and ablation studies.